# Efficient site-specific integration of large genes in mammalian cells via continuously evolved recombinases and prime editing

**Smriti Pandey**[1,2,3,7], **Xin D. Gao** [1,2,3,7], **Nicholas A. Krasnow** [1,2,3], **Amber McElroy**[4], **Y. Allen Tao**[1,2,3], **Jordyn E. Duby**[1,2,3], **Benjamin J. Steinbeck** [4], **Julia McCreary**[1,2,3], **Sarah E. Pierce**[1,2,3], **Jakub Tolar** [4], **Torsten B. Meissner** [5,6], **Elliot L. Chaikof**[5,6], **Mark J. Osborn**[4] & **David R. Liu** [1,2,3] ✉

Methods for the targeted integration of genes in mammalian genomes suffer from low programmability, low efficiencies or low specificities. Here we show that phage-assisted continuous evolution enhances prime-editing-assisted site-specific integrase gene editing (PASSIGE), which couples the programmability of prime editing with the ability of recombinases to precisely integrate large DNA cargoes exceeding 10 kilobases. Evolved and engineered Bxb1 recombinase variants (evoBxb1 and eeBxb1) mediated up to 60% donor integration (3.2-fold that of wild-type Bxb1) in human cell lines with pre-installed recombinase landing sites. In single-transfection experiments at safe-harbour and therapeutically relevant sites, PASSIGE with eeBxb1 led to an average targeted-gene-integration efficiencies of 23% (4.2-fold that of wild-type Bxb1). Notably, integration efficiencies exceeded 30% at multiple sites in primary human fibroblasts. PASSIGE with evoBxb1 or eeBxb1 outperformed PASTE (for 'programmable addition via site-specific targeting elements', a method that uses prime editors fused to recombinases) on average by 9.1-fold and 16-fold, respectively. PASSIGE with continuously evolved recombinases is an unusually efficient method for the targeted integration of genes in mammalian cells.

Mutations that contribute to human diseases range from single-nucleotide changes to large deletions, inversions, translocations and duplications[1–3]. Many genetic diseases are associated with a variety of loss-of-function mutations within a specific gene: for instance, over 500 *ABCA4*, 1,000 *PAH* and 2,000 *CFTR* gene variants have been reported in patients with Stargardt disease, phenylketonuria and cystic fibrosis, respectively[4–6]. In principle, integrating full-length healthy genes or complementary DNAs (cDNAs) into their endogenous loci could serve as a single therapeutic strategy for patients with different pathogenic alleles. Integration into the native locus could preserve physiological gene expression, evading gene overexpression associated with viral vector-mediated gene therapy that can induce pathology[7–9].

Motivated by this potential, the development of technologies that efficiently and precisely integrate large DNA sequences into the mammalian genome at specified target sites has been a long-standing goal[10]. Although programmable nucleases followed by either random

[1]Merkin Institute of Transformative Technologies in Healthcare, Broad Institute of MIT and Harvard, Cambridge, MA, USA. [2]Department of Chemistry and Chemical Biology, Harvard University, Cambridge, MA, USA. [3]Howard Hughes Medical Institute, Harvard University, Cambridge, MA, USA. [4]Department of Pediatrics, University of Minnesota Medical School, Minneapolis, MN, USA. [5]Department of Surgery, Beth Israel Deaconess Medical Center, Harvard Medical School, Boston, MA, USA. [6]Wyss Institute of Biologically Inspired Engineering, Harvard University, Boston, MA, USA. [7]These authors contributed equally: Smriti Pandey, Xin D. Gao. ✉e-mail: drliu@fas.harvard.edu

end-joining or homology-directed repair can perform targeted DNA integration, these approaches generate double-stranded breaks that can induce undesired consequences such as target locus deletion or chromosomal translocations, suffer from low integration efficiencies and typically generate a high frequency of uncontrolled indels, reversed-orientation cargo by-products, and multimeric insertions[11–19]. The recent discovery and characterization of clustered regularly interspaced short palindromic repeats (CRISPR)-associated transposase systems (CASTs) show promise for programmable integration but currently suffer from low efficiencies in mammalian cells (≤-1% genomic integration for Type-I CAST systems[20,21], and no reported mammalian genomic integration for Type-V-K CAST systems[22,23]).

We recently reported prime-editing-assisted site-specific integrase gene editing (PASSIGE), a technology that uses prime editing and site-specific large serine recombinases (LSRs) to integrate multi-kilobase DNA cargoes into targeted sites in the mammalian genome with up to 6.8% efficiency following a single transfection[24] (Fig. 1a). In PASSIGE, single-flap or dual-flap prime editing installs a site-specific recombinase landing site into a target genomic location[24,25]. The corresponding recombinase then catalyses cargo DNA insertion into the landing site, resulting in targeted integration. PASSIGE can be performed with a single transfection by simultaneously delivering all necessary components or using two successive transfections to perform the prime editing step and recombination at different times. A similar method, programmable addition via site-specific targeting elements (PASTE) that uses prime editors fused to site-specific recombinases, was later described by a separate study[26].

Current programmable, large gene-integration technologies in mammalian cells exhibit modest integration efficiencies. In PASSIGE, despite effective recombinase attachment site installation using dual-flap prime editing (PE) (typically >50%), overall integration efficiencies remain modest (2.6%–6.8%), indicating that the recombination step mediated by Bxb1 recombinase primarily constrains integration yields[24]. Indeed, multiple groups have reported that in mammalian cells with pre-installed genomic recombinase landing sites, treatment with Bxb1 and a donor DNA plasmid results in -10–20% integration[24,27–30]. These observations suggest the opportunity to improve PASSIGE efficiencies by evolving and engineering recombinase enzymes.

In this Article, we developed a phage-assisted continuous and non-continuous evolution (PACE and PANCE)[31,32] selection for recombinase activity and used it to evolve Bxb1 for higher PASSIGE efficiencies. Among dozens of Bxb1 variants with improved activity, one evolved variant, evoBxb1, achieved a 2.7-fold average improvement in genomic integration efficiencies in human cells at pre-installed recombinase attachment sites. We also combined evolved mutations to generate an even more active variant, eeBxb1.

We refer to the use of PASSIGE with evoBxb1 or eeBxb1 as evoPASSIGE or eePASSIGE. Across 12 mammalian genomic loci, evoPASSIGE and eePASSIGE demonstrate a 2.7-fold and 4.2-fold average improvement, respectively, in targeted large DNA integration efficiencies over PASSIGE, and outperform PASTE by an average of 9.1-fold and 16-fold. PASSIGE variants can achieve 20–46% integration of multi-kilobase gene-sized cargo at both safe-harbour and therapeutic loci following a single transfection. In primary human fibroblasts, eePASSIGE outperforms PASSIGE by 14-fold on average at two therapeutically relevant genomic sites, yielding integration efficiencies up to 30%. To our knowledge, these outcomes are among the highest RNA-programmed gene-sized genomic integration efficiencies that have been reported in mammalian cells thus far, and exceed efficiencies known to rescue a variety of loss-of-function genetic diseases[33–39].

## Results

### Development of a recombinase PACE circuit
PACE and PANCE[31,32] are methods for rapidly evolving proteins with diverse functions[40] (Fig. 1b). During PACE and PANCE, gene III, which

encodes pIII, a protein essential for phage replication is replaced with the protein being evolved in the M13 filamentous bacteriophage to generate the selection phage (SP). In a fixed-volume vessel ('lagoon'), the SP infects host *Escherichia coli* cells harbouring accessory plasmids that link the activity of the protein being evolved to the expression of gene III, as well as a mutagenesis plasmid (MP) that constantly mutagenizes the phage genome post infection. During PACE, the SP is continuously diluted with fresh host cells[31], while during PANCE, the SP is diluted in discrete steps[32]. PANCE is less stringent than PACE and can be helpful in the early phases of evolution when variants have low initial activity[32]. During evolution, only SP encoding active protein variants persist within the lagoon, while inactive SP are diluted out.

To link Bxb1-mediated recombination to gene III expression and subsequent phage propagation, we developed two selection circuits (Fig. 1c). In both circuits, the SP encodes Bxb1, and the host cells harbour a plasmid P1 with a promoter sequence and a plasmid P2 with a promoter-less gene III cassette. In circuit 1, the promoter and the sequence upstream of gene III are placed between two recombinase attachment sites. Upon Bxb1 expression, two recombination events place the promoter upstream of gene III, driving its expression. In contrast, circuit 2 has one recombinase attachment site present in each plasmid, and a single recombination event integrates P1 and P2, placing the promoter upstream of gene III. We anticipated that circuit 1 would be more stringent than circuit 2 as two recombination events are required for phage propagation.

To identify the best evolution strategy to evolve Bxb1, we established four subcircuits (1.1–1.4) for circuit 1 and two subcircuits (2.1–2.2) for circuit 2 (Extended Data Fig. 1). In subcircuits 1.1, 1.2, 2.1 and 2.2 we placed *attB*, one DNA landing site substrate for Bxb1, in P1, and *attP*, the partner DNA landing site substrate for Bxb1, in P2. In subcircuits 1.3, and 1.4 we instead placed *attP* in P1 and *attB* in P2. It is known that the central dinucleotide of the attachment site for the Bxb1 recombinase can either be GT or GA[27]. Subcircuits 1.2, 1.4 and 2.2 contained a GA instead of the canonical GT central dinucleotide.

### Evolution of the Bxb1 recombinase
We next performed PANCE of Bxb1 in all six subcircuits. Throughout the evolution, we increased stringency by reducing the time between serial dilution from 12 to 4 h and increasing dilution ratios between passages from 50:1 to 5,000:1. After six passages, phage across all six PANCE lagoons propagated from approximately onefold to >20,000-fold overnight, suggesting the emergence of Bxb1 variants with improved activity (Fig. 1d and Extended Data Fig. 1). Sequencing of individual phage revealed some mutational convergence, even across different circuits (Supplementary Table 1).

Since the evolutionary trajectory of subcircuit 1.3 suggested it to be the most stringent, we continued the evolution campaign using this circuit. We further increased selection stringency by decreasing gIII expression and continued PANCE for four additional passages (Fig. 2a and Extended Data Fig. 2a). We then evolved the resulting phage pools for an additional 132 hours using PACE (Extended Data Fig. 2b), increasing selection stringency by elevating the flow rate from 0.5 to 3.0 vol h⁻¹. Finally, we subjected the phage pools surviving PACE to six additional passages of PANCE on a more stringent circuit in which we increased the size of P1 from 3.2 kb to 6.5 kb (Extended Data Fig. 2c). Overall, SP encoding Bxb1 emerging after the entire evolution process survived an average total dilution of -10¹⁵⁰, and sequencing revealed additional mutations that could enhance recombinase activity (Supplementary Tables 2–4).

### Characterization of evolved variants in mammalian cells with pre-installed attachment sites
We cloned 40 unique evolved Bxb1 variants into mammalian expression vectors and tested them in human HEK293T cells homozygous for either *attP* at *AAVS1* or *attB* at *CCR5*. These evolved variants were tested alongside wild-type (WT) Bxb1 used in PASSIGE and a catalytically inactive

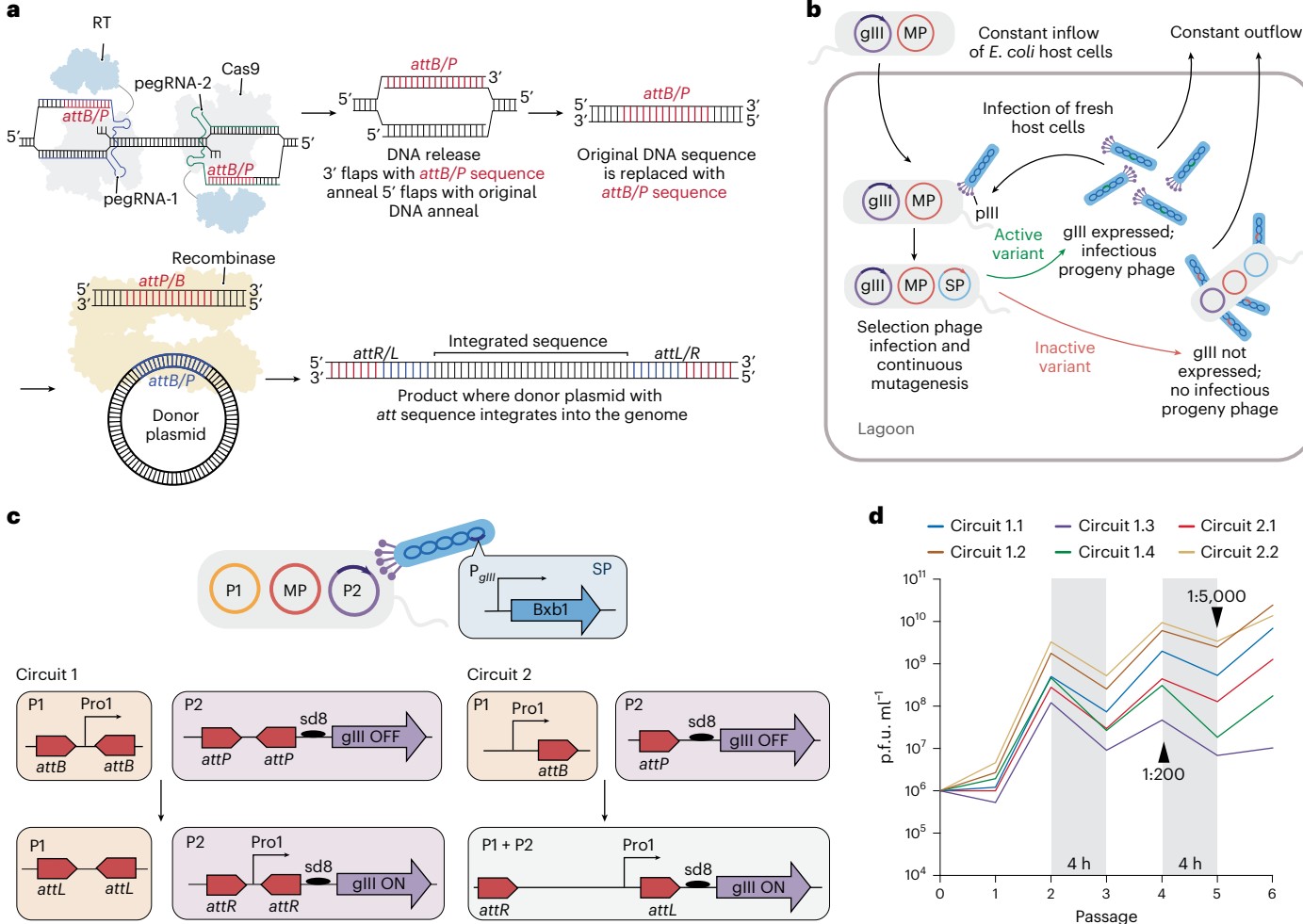

**Fig. 1 | Phage-assisted evolution of the Bxb1 recombinase for PASSIGE.**
**a**, An overview of PASSIGE. Prime editing (dual flap or single flap) precisely installs a large serine recombinase (LSR) attachment site (*attB* or *attP*) into a target locus in the genome. The LSR then recognizes the installed *att* motif and integrates donor DNA into this site. **b**, An overview of PACE. The selection phage (SP) encodes the protein being evolved. Host *E. coli* cells encode a mutagenesis plasmid (MP), as well as plasmids that link the activity of the evolving protein to expression of gIII, an essential phage gene. Only phages that encode active variants trigger gIII expression and propagate. A constant dilution of host cells and media washes out inactive phage variants that are unable to propagate faster than the dilution rate. **c**, A schematic of the recombinase-PACE selection circuit. Bxb1 recombinase is encoded on the SP. Host cells harbour plasmid

P1 that encodes promoter Pro1, and plasmid P2 that encodes a promoter-less gIII cassette. Bxb1-mediated recombination places Pro1 upstream of the gIII cassette, driving its expression. In circuit 1, two attachment sites are present in each plasmid resulting in two recombination events that exchanges sequences between P1 and P2. In circuit 2, one attachment site is present in each plasmid resulting in one recombination event that integrates P1 and P2. **d**, PANCE phage titre for the evolution of Bxb1 recombinase across six circuits (1.1–1.4 and 2.1–2.2). Each trace reflects the mean value of phage titres across four different lagoons. Individual traces for each lagoon are shown in Extended Data Fig. 1. Selection stringency was modulated by decreasing the selection time and increasing dilution factor. Unless otherwise indicated, each passage was performed overnight, and phage were diluted 1:50 after each passage.

Bxb1 variant (dead Bxb1, S10A and Y154C)[41]. Clonal HEK293T cells were transfected with the recombinase plasmid along with a 5.6-kb donor plasmid containing either an *attP* or *attB* landing site. After 72 h, integration efficiencies were assessed by droplet digital polymerase chain reaction (ddPCR) (Supplementary Note 1). Nearly all evolved variants (39/40) showed enhanced integration efficiencies over WT Bxb1, with the top 15 variants exceeding 2.4-fold improvements (Fig. 2b). The diverse set of unique solutions found by the Bxb1-encoding SP explains the absence of a single dominant genotype throughout evolution (Supplementary Tables 1–4). Notably, the V105I mutant supported 60% and 39% integration efficiencies at *AAVS1* and *CCR5*, respectively, compared with 18% and 12% with WT Bxb1, a 3.2-fold average improvement (Fig. 2c).

**Mapping beneficial mutations onto the AlphaFold-predicted structure of Bxb1**

To hypothesize potential roles of the evolved mutations that improve integration efficiencies, we mapped them onto an AlphaFold2 (ref. 42)

predicted structure of the Bxb1 recombinase. The N-terminal domain (NTD), C-terminal domain-a (CTD-a) and C-terminal domain-b (CTD-b), all aligned well with previously solved structures of serine recombinases (Protein Data Bank (PDB): 1ZR4 (ref. 43), 6DNW ref. 44 and 4KIS ref. 45) (Extended Data Fig. 3a). Despite being the smallest domain, the catalytic NTD harboured 30 unique mutations, more than any other domain. The DNA-binding CTD-a and CTD-b harboured 11 and 13 distinct mutations, respectively, and the linker connecting the two domains contained two mutations (Fig. 2d).

The 15 best-performing variants all contained a mutation in the NTD, and docking the DNA substrate of gammadelta resolvase tetramer[43] (PDB: 1ZR4) onto the predicted NTD structure suggested that majority of these mutations are present in flexible loops of the enzyme, near the active site and the DNA substrate (Extended Data Fig. 3b). All mutated residues in the flexible regions were also surface exposed (Extended Data Fig. 3c). Other conserved mutations that led to the highest improvements in integration efficiencies, including V105I,

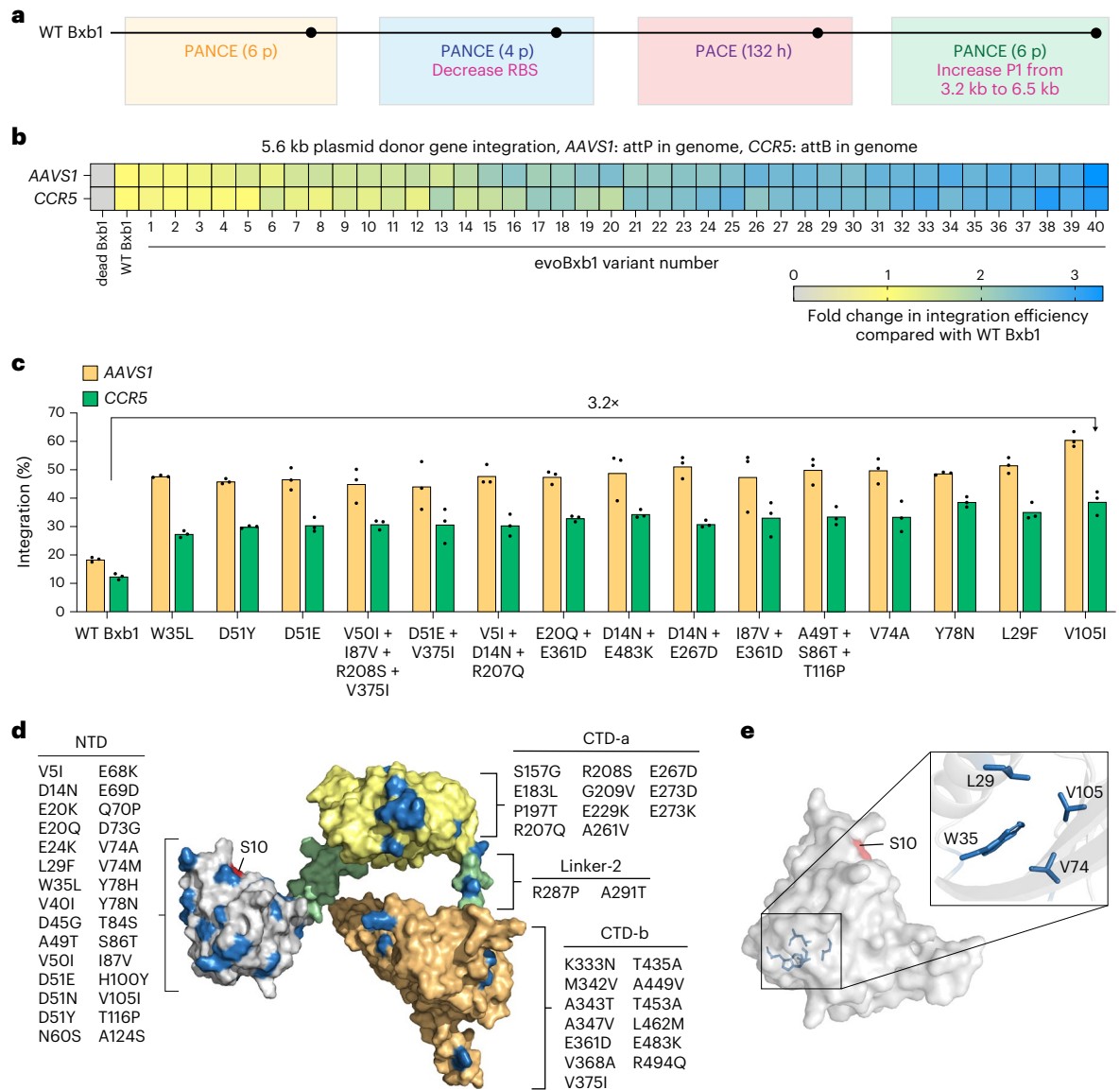

**Fig. 2 | Characterization of evolved Bxb1 variants in mammalian cells.**
**a**, Summary of the Bxb1 evolution campaign. p, PANCE passages. RBS, ribosome binding site. **b**, A heat map of fold change in integration efficiency compared to wild-type (WT) Bxb1 for evolved variants. A 5.6 kb donor plasmid along with either recombinase-dead Bxb1, WT Bxb1 or an evolved variant were transfected into HEK293T cells with either pre-installed *attP* in *AAVS1* or *attB* in *CCR5*. Each square reflects the mean value for three independent replicates. **c**, Absolute integration efficiencies for 15 evolved Bxb1 variants with the highest activity, and WT Bxb1 from **b**. The bars reflect the mean of three independent replicates and

dots show individual *n* = 3 replicate values. **d**, Alphafold2-predicted structure of the Bxb1 recombinase. The three distinct domains, NTD, CTD-a and CTD-b are in grey, yellow and orange, respectively. Linkers connecting the domains are in green. The catalytic residue, S10 is in red, and residues mutated during evolution are in blue. All mutated residues in each domain are listed. **e**, Predicted position of mutated residues that resulted in the highest integration efficiencies. Residues (blue) are mapped onto the AlphaFold2-predicted structure of the NTD of Bxb1 (grey). Integration efficiency (**b** and **c**) was assessed by ddPCR analysis as described in Supplementary Note 1.

L29F, V74A and W35L, were clustered at the protein core (Fig. 2e). The position of these mutations suggests that they probably stabilize the NTD core: for example, the change of Val to Ile at position 105 and Leu to Phe at position 29 may help stabilize the bulky, hydrophobic Trp residue at position 35 (Extended Data Fig. 3d). Collectively, these observations suggest that the evolved variants may enhance integration by optimizing active site conformation or improving protein stability.

## Characterization of evolved variants for PASSIGE

As reported previously[24], reducing the 3' flap overlap between the dual prime editing guide RNAs (pegRNAs) improves PASSIGE integration efficiencies by minimizing recombination between the donor DNA plasmid and pegRNA-encoding plasmids (Extended Data Fig. 4a).

To identify the ideal overlap length for installing either *attP* or *attB*, we co-transfected HEK293T cells with plasmids encoding PEmax and twinPE pegRNAs with varying overlap lengths. We found that the 3' flap overlap length could be truncated up to 28 bp for *attP* and 20 bp for *attB* installation without decreasing installation efficiencies or increasing indel frequencies (Extended Data Fig. 4b). Unless otherwise stated, all subsequent experiments used these overlap lengths, greatly reducing pegRNA design complexity across sites.

Next, we tested the ten most efficient Bxb1 variants from Fig. 2c in PASSIGE at the *AAVS1* and *CCR5* loci, where prime editing installed either an *attP* or *attB* sequence, respectively. HEK293T cells were co-transfected with a 5.6-kb donor DNA plasmid along with plasmids encoding either WT or an evolved Bxb1 variant, PEmax and dual

pegRNAs[24]. All ten variants showed improvements >2-fold compared to WT Bxb1, with the highest-performing variant, Bxb1-V74A, showing a 2.8-fold and 3.9-fold improvement in integration efficiency at the *AAVS1* and *CCR5* loci, respectively (Fig. 3a). Moving forward, we refer to the Bxb1-V74A variant as evoBxb1, and the use of evoBxb1 for PAS-SIGE as evoPASSIGE.

## Combining evolved mutations to further enhance integration efficiency

Since evoBxb1 as well as several other top-performing variants harbour only a single mutation, we next evaluated the integration efficiencies of Bxb1 variants with combined evolved NTD, CTD-a and CTD-b mutations. We generated 19 triple-mutant variants each harbouring one mutation in each domain, and tested them alongside WT Bxb1 in cell lines pre-installed with *attP* or *attB* at the *AAVS1* or *CCR5* loci, respectively. Interestingly, the best-performing variants all contained the E229K mutation and in both cell lines, combining evoBxb1 with E229K and V375I resulted in the highest integration efficiencies (Extended Data Fig. 4c). We refer to this evolved and engineered triple-mutant variant as eeBxb1, and the use of eeBxb1 for PASSIGE as eePASSIGE.

We compared the performance of single-transfection eePAS-SIGE, evoPASSIGE and PASSIGE side-by-side at *AAVS1* and *CCR5* in HEK293T cells and observed the highest integration efficiencies with eePASSIGE: 36% and 27% at the *AAVS1* and *CCR5* loci, respectively, compared with 27% and 22% with evoPASSIGE, and 13% and 10% with PAS-SIGE (Fig. 3b). In mouse N2a cells at the safe-harbour locus *Rosa26*, we observed an even more pronounced difference: eePASSIGE integrated a 5.6-kb donor DNA with 20% efficiency compared with 9.5% with evoPAS-SIGE and 3.2% with PASSIGE, corresponding to a 2.1-fold and 6.2-fold improvement in integration, respectively (Fig. 3b). For the *Rosa26* site, we used the PE6d prime editor variant, which we recently reported to outperform PEmax for *attB* installation at this site[46]. Improvements in attachment site installation led to modest enhancements in PASSIGE-mediated integration (Extended Data Fig. 4d), prompting us to evaluate PE6 variants for each target locus throughout the remainder of this study.

## Further characterization of PASSIGE variants

To determine whether the identity of the attachment site in the genome affects integration, we screened PE6 prime editor variants[46] to optimize installation efficiencies at the *AAVS1*, *CCR5*, *ACTB* and *Rosa26* loci (Supplementary Table 5) and evaluated donor integration when installing either *attP* or *attB*. When installing *attP*, we observed higher integration efficiencies at *AAVS1* and *ACTB* but not *CCR5* or *Rosa26*, implicating that the choice of attachment site that should be installed using prime editing is locus dependent (Fig. 3c).

Next, we compared PASSIGE variants side-by-side with PASTE, a similar technology reported to have improved integration efficiencies over PASSIGE. PASTE differs from PASSIGE by using (1) a pegRNA scaffold mutant previously described by Wu and co-workers[47] (atgRNAv2), (2) a different Cas9–reverse transcriptase (RT) linker, (3) addition of the L139P mutation that we previously characterized[25] to the engineered Moloney murine leukaemia virus (M-MLV) RT in PE2 and (4) a mutated *attP* sequence[26]. We systematically tested each of these optimizations in PASSIGE, evoPASSIGE and eePASSIGE systems but did not observe any consistent improvements in targeted integration across multiple genomic loci in HEK293T cells (Extended Data Fig. 5a–d). The atgRNAv2 scaffold slightly improved integration in some cases (Extended Data Fig. 5a), but the Cas9–RT linker, L139P mutation in the M-MLV RT, and *attP* mutant reduced integration efficiencies across all sites (Extended Data Fig. 5b–d).

In PASTE, fusing the recombinase to the prime editor protein was reported to substantially improve integration[26]. However, when we fused WT Bxb1, evoBxb1 or eeBxb1 to PEmax, integration efficiencies substantially decreased compared to unfused prime editor + recombinase (Extended Data Fig. 5e). This trend persisted when replacing the recombinase in PASTE with Bxb1 variants generated in this study (Extended Data Fig. 5f). These observations are consistent with the mechanism of prime editing and recombinase-mediated integration, in which prime editing and recombination cannot occur simultaneously and instead might interfere with each other when tethered. Indeed, PASSIGE, evoPASSIGE and eePASSIGE all outperformed PASTE by an average of 2.5-fold, 6.1-fold and 10-fold, respectively, across four genomic loci (Fig. 3d).

We assessed the influence of donor DNA size on integration efficiencies for all PASSIGE variants and PASTE at the *AAVS1* and *CCR5* loci. On average, PASSIGE and PASTE exhibited a 1.7-fold and 2.2-fold decrease in integration, respectively, when using a 10.5-kb plasmid compared with a 3.0-kb plasmid (Fig. 3e). In contrast, the impact of donor size on integration efficiency was less pronounced when using evo- and eePASSIGE, averaging 1.2- and 1.1-fold decreases, respectively (Fig. 3e). Notably, eePASSIGE achieved up to 35% targeted integration of the 10.5-kb donor plasmid, a 3.8-fold improvement compared to PASSIGE, which only achieved up to 9.4% integration.

## Characterization of PASSIGE variants at therapeutic sites

The above experiments evaluated the evolved Bxb1 variants at safe-harbour loci *AAVS1*, *CCR5* and *Rosa26* (ref. 48) and at the highly expressed essential gene *ACTB*[49]. Next, we tested the ability of PAS-SIGE to integrate gene-sized cargo into eight therapeutically relevant endogenous genomic sites in HEK293T and N2a cells, including (1) *ALB*, a highly expressed gene in the liver previously used to express clinically relevant protein levels for loss-of-function diseases[33,50], (2) *B2M* and *TRAC*, used to express chimeric antigen receptors for chimeric antigen receptor (CAR)-T cell therapy[51] and (3) *CFTR*, *GBA1*, *COL7A1*, *FANCA* and *Smn1*, implicated in cystic fibrosis[4], Gaucher disease[52], Parkinson's disease[53], dystrophic epidermolysis bullosa[54], Fanconi anaemia[55] and spinal muscular atrophy[56]. We designed pegRNAs and tested PE6 variants to install both *attB* and *attP* into each locus (Supplementary Table 5). For *B2M* and *TRAC*, we installed the attachment sites into the 5′ untranslated region (UTR) surrounding the start codon, as disrupting these genes increase therapeutic potency[51]. For *COL7A1*, we installed the attachment sites into intron 4, as most disease-causing mutations are located after exon 4 (ref. 54). For all other genes, we installed the attachment site into intron 1.

Next, we integrated a 5.6-kb donor plasmid into all eight therapeutically relevant loci using PASSIGE, evoPASSIGE, eePASSIGE and PASTE. Evo- and eePASSIGE showed substantial improvements in integration over PASSIGE at all eight target sites, and PASSIGE substantially outperformed PASTE at all sites tested (Fig. 4a). Across all eight target sites in human and mouse cells, eePASSIGE, evoPASSIGE, PASSIGE and PASTE mediated targeted donor DNA integration with an average efficiency of 22%, 17%, 7.8% and 3.8%, respectively.

We then analysed the fold change in integration efficiencies of evolved Bxb1 variants across all 12 genomic sites used in this study. Averaged across all 12 sites, evoPASSIGE and eePASSIGE outperformed PASSIGE by 2.7-fold and 4.2-fold, respectively, (Fig. 4b). PAS-SIGE, evoPASSIGE and eePASSIGE outperformed PASTE by an average of 3.3-fold, 9.1-fold and 16.2-fold, respectively (Fig. 4c). Notably, we observed >30% integration at *AAVS1*, *ACTB* and *FANCA*, and >20% integration at *B2M*, *GBA1*, *COL7A1*, *CFTR*, *Smn1*, *CCR5* and *ALB* when using eePASSIGE (Fig. 4d). Overall, these findings demonstrate that evoPAS-SIGE and eePASSIGE exceed targeted multi-kb donor DNA integration efficiencies over previously reported methods at multiple safe-harbour and therapeutic loci.

## Evaluating genomic outcomes after integration across multiple loci

To characterize alleles post-integration, we developed a high-throughput sequencing (HTS) assay (schematic shown in Extended

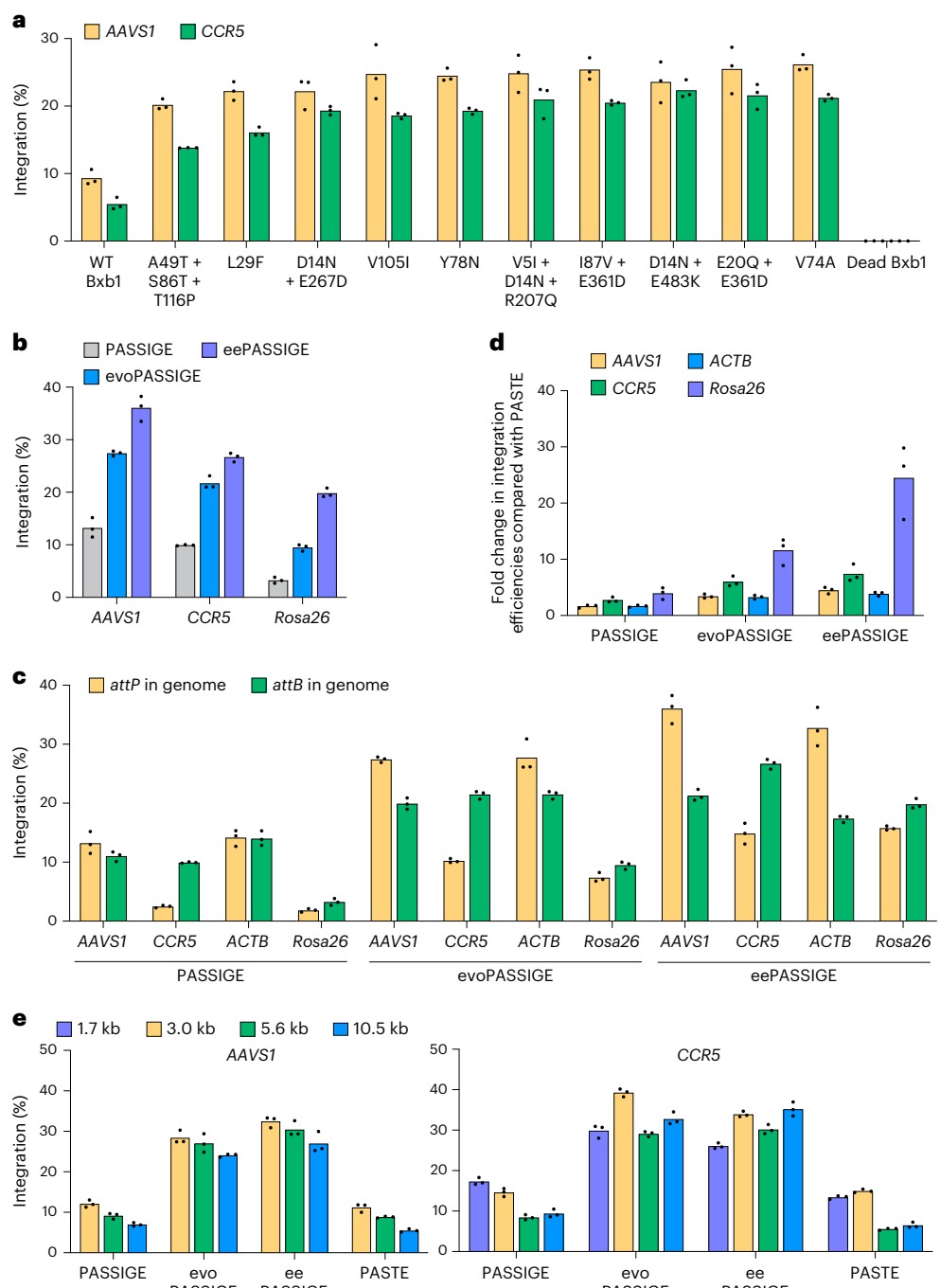

**Fig. 3 | Characterization of evolved Bxb1 variants for PASSIGE.**
**a**, Absolute integration efficiencies for ten evolved Bxb1 variants with the highest activity from Fig. 2b,c, and WT Bxb1 in the PASSIGE system. **b**, Absolute integration efficiencies for PASSIGE (WT Bxb1), evoPASSIGE (Bxb1-V74A) and eePASSIGE (Bxb1-V74A + E229K + V375I). **c**, Comparison of integration efficiencies when installing either *attP* or *attB* into *AAVS1*, *CCR5*, *ACTB* and *Rosa26* genomic loci using PASSIGE, evoPASSIGE and eePASSIGE. **d**, Fold change in integration efficiencies relative to PASTE for PASSIGE, evoPASSIGE and eePASSIGE across four loci. **e**, The effects of donor size on PASSIGE,

evoPASSIGE, eePASSIGE and PASTE. For PASSIGE and PASTE experiments, all components were delivered using single transfection and a 5.6-kb donor DNA plasmid was used (**a**–**d**). In **a**, **b**, **d** and **e**, dual pegRNAs were used to insert *attP* into *AAVS1* and *ACTB* or *attB* into *CCR5* and *Rosa26*. *Rosa26* is a genomic site in N2a cells; all other sites are in HEK293T cells (**a**–**e**). The bars reflect the mean of three independent replicates and dots show individual *n* = 3 replicate values. The integration efficiency (**a**–**e**) was assessed by ddPCR analysis as described in Supplementary Note 1.

Data Fig. 6a) to simultaneously assess attachment site installation, donor integration and indel frequencies. During sample preparation, unique molecular identifiers (UMIs) were applied to minimize potential PCR bias[24,57]. HTS integration efficiencies were consistent with those obtained from ddPCR, validating this assay as an orthogonal approach to assess per cent integration (Extended Data Fig. 6b).

HTS analysis of the first integration junction revealed that evo- and eePASSIGE did not yield higher indel frequencies compared to PASSIGE (Extended Data Fig. 6c). Indeed, most indels across samples originated from prime editing rather than recombination, consistent with the known mechanisms of both processes, with most indels arising due to incomplete attachment site installation (Supplementary Table 6).

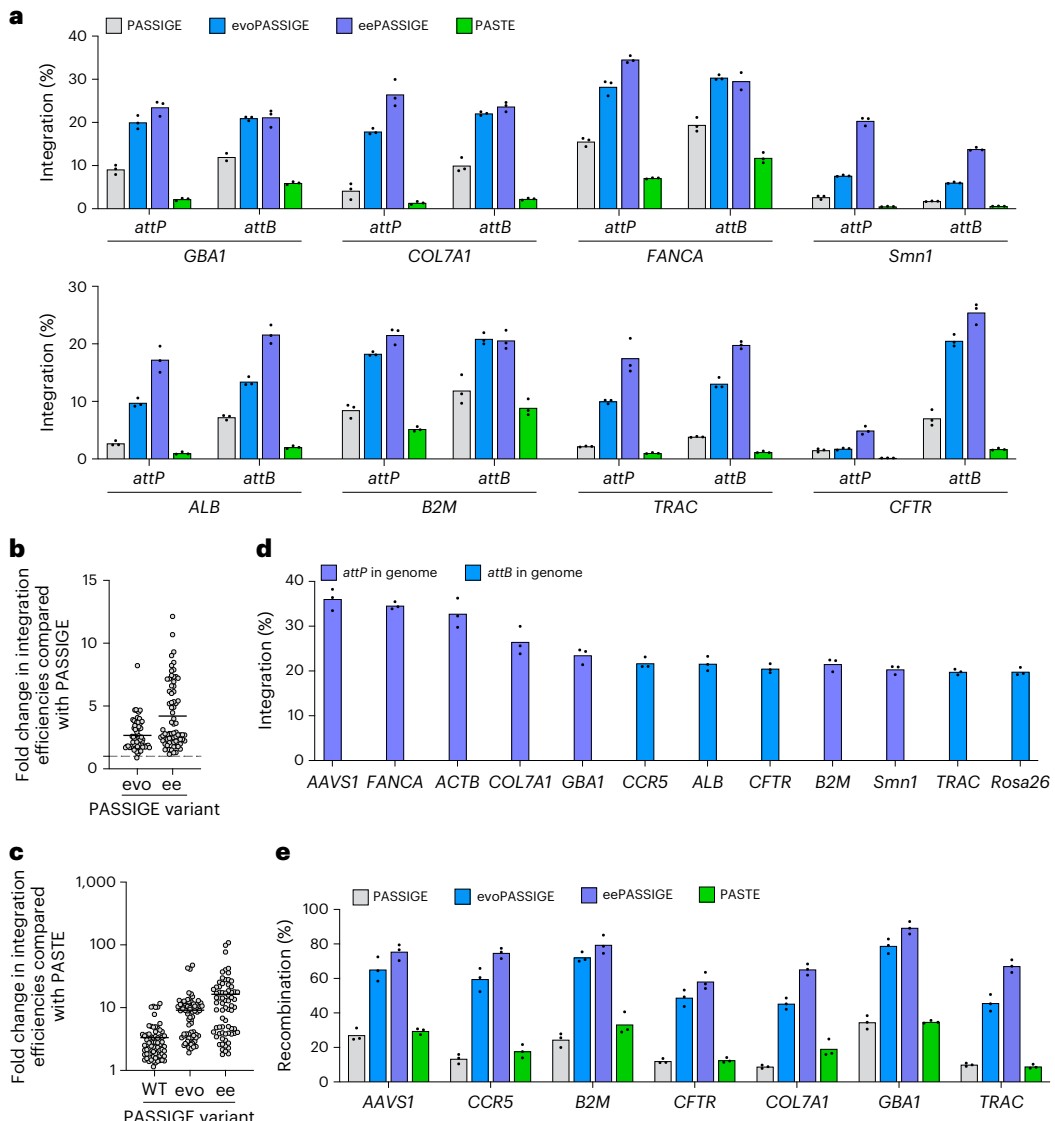

**Fig. 4 | Characterization of PASSIGE, evoPASSIGE, eePASSIGE and PASTE at additional loci. a**, The absolute integration efficiencies for PASSIGE, evoPASSIGE, eePASSIGE and PASTE at eight different therapeutically relevant genomic sites. Integration was assessed when installing both *attP* and *attB* into each locus separately. **b**, The fold change in integration efficiencies relative to PASSIGE for evoPASSIGE and eePASSIGE across all sites tested in this study. **c**, The fold change in integration efficiencies relative to PASTE for PASSIGE, evoPASSIGE and eePASSIGE across all sites tested in this study. **d**, The absolute integration efficiencies with eePASSIGE at 12 sites. Either *attP* or *attB* was installed into the genome, as indicated. **e**, The recombination efficiencies for PASSIGE, evoPASSIGE, eePASSIGE and PASTE across seven sites. For *CCR5*,

and *CFTR*, *attB* was installed into the genome; for all other genomic sites *attP* was installed. HTS with UMI analysis was used to quantify recombination efficiencies. For PASSIGE and PASTE experiments in **a**–**e**, all components were delivered using single transfection and a 5.6-kb donor DNA plasmid was used. *Smn1* and *Rosa26* are genomic sites in N2a cells, all other genomic sites are in HEK293T cells. For **b** and **c**, integration efficiencies were evaluated at 12 different loci when installing both *attP* and *attB* separately. The bars reflect the mean of three independent replicates (**a**,**d** and **e**), dots show individual *n* = 3 replicate values (**a**–**e**) and horizontal lines show the mean value (**b** and **c**). The integration efficiency (**a**–**d**) was assessed by ddPCR analysis as described in Supplementary Note 1.

To assess Bxb1 variant activity independently from the prime editing step, we quantified the total per cent recombination by multiplying the ratio of total integration reads to the sum of (attachment site installation reads + integration reads) by 100. Across seven genomic loci, PASSIGE achieved an average of 19% recombination, consistent with previous observations[24,27–30], compared with 59% with evoPASSIGE and 73% with eePASSIGE (Fig. 4e). Remarkably, eePASSIGE achieved up to 92% recombination at the *GBA1* locus and on average demonstrated a 3.9-fold improvement over PASSIGE. It is interesting to note that PASTE recombination efficiencies were similar to that of PASSIGE, suggesting that the substantially lower integration observed when using PASTE may primarily be due to the reduced activity of the prime editor. Indeed, across 11 sites, PASTE installed attachment sites with an average

of 1.9-fold lower efficiencies compared with PEmax (Extended Data Fig. 6d). Collectively, the HTS assay validated integration efficiencies quantified by ddPCR and provided strong evidence that the evolved and engineered Bxb1 variants substantially improve recombination efficiencies without increasing indel frequencies.

### Effects of chromatin accessibility on gene integration
The site-dependent variation in PASSIGE- and PASTE-mediated integration efficiencies led us to investigate the potential influences of chromatin accessibility on these systems. Since histone modifications are key indicators of chromatin accessibility,[58] we extracted HEK293T cell chromatin immunoprecipitation (ChIP)-sequencing scores reflecting the abundance of all available histone modifications within a 1-kb

window centred around each target site[59], and correlated them with integration efficiencies at ten genomic loci where we installed *attP* into the genome.

PASSIGE, evoPASSIGE, eePASSIGE and PASTE integration efficiencies were all positively correlated with active chromatin histone markers H3K27ac and H3K4me3 (Pearson coefficient $r > 0.64$, $P < 0.05$ for all conditions), and negatively correlated with the heterochromatin-associated histone marker H3K9me3 ($r < −0.60$ for all conditions, $P < 0.05$ for evoPASSIGE and eePASSIGE) (Extended Data Fig. 7). To assess the effects on Bxb1 variant-mediated recombination specifically, we performed a similar analysis using recombination efficiencies obtained from HTS. Although we observed similar trends, the correlations were weaker (for active chromatin markers H3K27ac and H3K4me3, $0.31 < r < 0.65$, $P$ value not significant; for heterochromatin-associated marker H3K9me3, $r < −0.46$, $P$ value not significant except for PASTE $P < 0.05$) (Supplementary Note 2).

These results indicate that higher integration efficiencies in active chromatin regions primarily originate from more efficient attachment site installation during the prime editing step, consistent with recent reports that experimentally demonstrate a positive correlation between prime editing and euchromatin markers[60–62].

## Off-target profiling of evolved Bxb1 variants

To assess off-target integration of Bxb1 variants, we first co-transfected HEK293T cells with either dead Bxb1, WT Bxb1, evoBxb1 or eeBxb1 along with an *attP*- or *attB*-containing DNA donor plasmid encoding mCherry. We passaged cells for 2 weeks to dilute the donor plasmid and then performed flow cytometry to assess the percentage of mCherry+ cells (Supplementary Note 3). We reasoned that any mCherry expression detected above the background signal from dead Bxb1 could be attributed to genomic integration of the mCherry cassette into an off-target site.

We observed very low percentage of mCherry+ cells for dead Bxb1 and no statistically significant integration above background for the WT Bxb1 or evoBxb1 when transfecting either donor plasmid ($P > 0.4$), indicating the absence of off-target activity (Fig. 5a). For eeBxb1, we observed a significant increase in mCherry expression above background when transfecting an *attP*-donor ($P < 0.001$), suggesting this highly active variant may recognize and integrate its cargo into *attB*-resembling sequences in the genome. No off-target activity was detected when transfecting eeBxb1 with an *attB*-donor ($P = 0.2$), possibly due to the minimal *attP* sequence required for recombination being 10-bp longer than that for *attB*, making its occurrence in random DNA approximately $4^{10}$ times rarer. The identities of these ten bases in *attP* are the most important for Bxb1-mediated recombination[63].

To identify which CTD mutations contribute to off-target integration when transfecting an *attP*-donor with eeBxb1, we tested variants V74A (evoBxb1), V74A + E229K, V74A + V375I and V74A + E229K + V375I (eeBxb1) alongside dead Bxb1. We identified E229K as the cause for off-target integration as adding this mutation significantly increased mCherry expression above background ($P < 0.0001$) (Extended Data Fig. 8a). Docking the DNA substrate from the *Listeria innocua* prophage serine recombinase[44] (PDB: 6DNW) onto the AlphaFold2 (ref. 42) predicted structure of the CTD-a domain of Bxb1 suggests that this negatively charged Glu side chain is located near the DNA substrate (Extended Data Fig. 8b). Mutating this residue to a positively charged lysine may increase the affinity of the recombinase to the negatively charged DNA backbone in a sequence-independent manner, resulting in more potent DNA engagement and increased integration efficiency but reduced sequence specificity.

We further profiled off-target integration for PASSIGE, evoPASSIGE, eePASSIGE and PASTE when delivering an *attP*-containing donor by using a modified version of UDiTaS, inspired by previous publications[26,64,65] (schematic shown in Supplementary Note 4). We transfected cells with PASSIGE components to integrate a 5.6-kb

puromycin-encoding donor plasmid, containing a UMI, into the *CCR5* locus via *attB* installation. To enrich for integration events, we selected cells with puromycin for 2 weeks before collecting them for analysis. We performed the assay for all PASSIGE variants and PASTE.

UDiTaS-nominated >100 off-target sites for eeBxb1 (Supplementary Table 7). In contrast to the fluorescence-based assay used above, UDiTaS also nominated 23 and 3 off-target sites for evoBxb1 and WT Bxb1, respectively (Extended Data Fig. 8c and Supplementary Table 7). To validate the authenticity of these hits, we performed ddPCR analysis on a subset of nominated sites, since designing customized ddPCR assays for >100 loci is impractical. We selected all off-target sites nominated for evoBxb1 that had multiple reads aligned post-deduplication and one highly enriched nominated site for eeBxb1 as a potential positive control, since we had already validated its ability to mediate off-target integration when delivering an *attP*-containing donor above. To validate the design of the ddPCR assay, we included a 1:1 mixture of a DNA sequence encoding the same off-target amplicon identified by UDiTaS and a reference *ACTB* sequence.

The ddPCR analysis confirmed off-target integration at 4/5 sites (OT1–4) tested (ddPCR assay for one evoBxb1-nominated off-target could not be optimized). All four off-target sites are homologous to the Bxb1 *attB* sequence (Supplementary Table 7), supporting our hypothesis that these recombinases recognize *attB* pseudosites in the genome. Consistent with UDiTaS, the highest off-target activity was observed at OT4, with WT Bxb1, evoBxb1 and eeBxb1 achieving integration efficiencies of 3.8%, 20.9% and 24%, respectively (Extended Data Fig. 8c,d). In contrast, despite being the second-most enriched off-target read for evoBxb1 in UDiTaS (Extended Data Fig. 8c), ddPCR analysis did not reveal any off-target integration at OT5 (0.04%, 0.01%, 0.08% and 0.03% with WT Bxb1, evoBxb1, eeBxb1 and dead Bxb1, respectively) (Extended Data Fig. 8d), indicating that while UDiTaS can nominate off-target sites for large-gene integration technologies, even the most enriched hits can be false positives. The nomination of off-target sites for the dead recombinase control further supports this (Extended Data Fig. 8c). See Supplementary Note 4 for a detailed discussion on the validity of UDiTaS-nominated off-target candidates.

We next performed a similar analysis when *attP* is installed into the genome by integrating a 5.6-kb puromycin-encoding donor plasmid via *attP* installation into the *AAVS1* locus. UDiTaS initially nominated ten off-target sites for eeBxb1 when delivering an *attB* donor. However, eight out of ten nominations contained a random DNA sequence between the genome and the integrated donor, inconsistent with the known mechanism of LSR-mediated recombination[66]. Indeed, ddPCR analysis revealed <0.08% integration at six of the eight sites (all sites with multiple sequences aligned post-deduplication were chosen along with two sites with only one read), strongly suggesting that they were not genuine off-targets (Supplementary Note 4). Further validation using ddPCR analysis of the two remaining nominated off-target sites, OT1 and OT2, whose sequences did not contain these artifacts (Fig. 5b) revealed minimal integration efficiencies: <0.08% and <0.16% at OT1 and OT2, respectively (Fig. 5c).

Collectively, these data demonstrate that in-depth validation of UDiTaS-nominated hits is essential to confirm their authenticity. We hypothesize that false positives may arise from template-switching during PCR, as the reverse primer used to generate UDiTaS amplicons binds to the donor plasmid, which is present in multiple different molecules in cells. Indeed on average, 43% and 18% of reads obtained from UDiTaS aligned to the donor plasmid and pegRNA-donor recombined products, respectively (Supplementary Note 4 and Extended Data Fig. 8e). Given these observations, we reason that the actual number of off-target sites for the Bxb1 variants, even when *attB* is installed into the genome, is substantially lower than the number of sites nominated by UDiTaS. Although four out of five off-target sites were validated using ddPCR analysis in this scenario, they were pre-selected based on having multiple reads aligned to them after deduplication. Over 80%

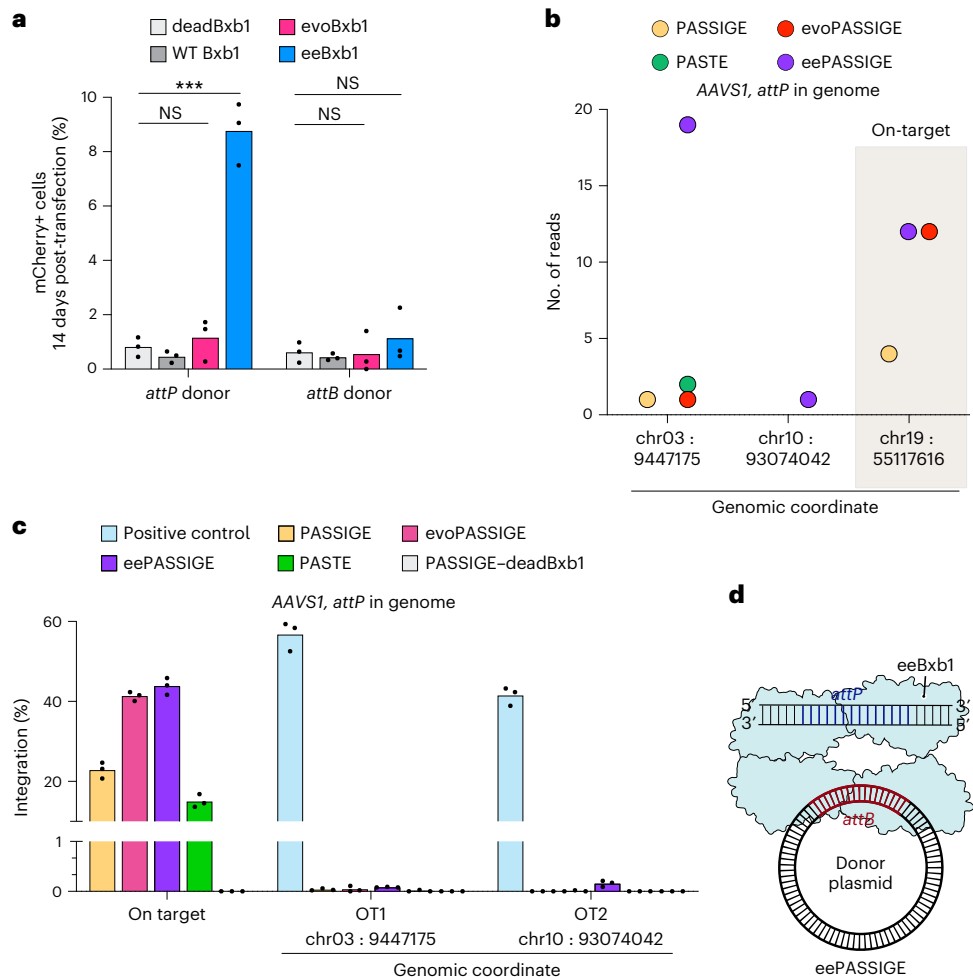

**Fig. 5 | Off-target profiling of PASSIGE, evoPASSIGE, eePASSIGE and PASTE.**
**a**, The percentage of mCherry-positive cells 14 days after transfecting a 3.2-kb donor DNA plasmid along with either dead Bxb1, WT Bxb1, evoBxb1 or eeBxb1. The donor plasmid either has an *attP* or *attB* site and encodes mCherry under the CMV promoter. Statistical significance was calculated using Student's unpaired two-tailed *t*-test, ***$P < 0.001$. **b**, The number of uniquely mapped reads and genomic coordinates (human GRCh38) for UDiTaS-nominated off-target sites when *attP* is installed into the genome. The on-target *AAVS1* locus is shaded. All PASSIGE variants and PASTE were used to integrate a puromycin-encoding donor plasmid. **c**, Absolute integration efficiencies at the on-target and UDiTaS-nominated off-target sites when *attP* is installed into the *AAVS1* locus for all PASSIGE variants and PASTE. For the negative control, dead Bxb1

was used instead of the WT recombinase in PASSIGE. For the positive control, a DNA sequence encoding the off-target sequence identified by UDiTaS was mixed with an *ACTB* reference sequence in a 1:1 ratio, so that roughly 50% of the total droplets would give a positive signal. Integration efficiency was assessed by ddPCR analysis as described in Supplementary Note 1. **d**, The recommended configuration for PASSIGE using eeBxb1. To minimize off-target integration, dual pegRNAs should be used to install *attP* into the genome, and the eeBxb1 variant should be used to integrate the DNA cargo. For PASSIGE and PASTE experiments, all components were delivered into cells using a single-transfection (**b** and **c**). The bars reflect the mean of three independent replicates and dots show individual $n = 3$ replicate values (**a** and **c**).

of nominated off-targets only had a single read aligned and several of them also had a random DNA sequence inserted between the genomic site and integrated donor, similar to the artifactual sequences observed in *attB* donor treated samples (Supplementary Table 7).

Overall, off-target profiling reveals that all Bxb1 variants, including WT Bxb1, exhibit infrequent but detectable levels of off-target integration when *attB* is installed into the genome. In contrast, minimal off-target integration for these recombinases was observed when *attP* is installed into the genome. To achieve the highest integration efficiencies while minimizing off-target events, we recommend installing *attP* into the genome and using the eeBxb1 variant (Fig. 5d).

**Integration of therapeutic DNA cargo using PASSIGE variants**
Next, we integrated therapeutic gene cargoes into multiple genomic loci optimized in Fig. 4a using PASSIGE, eePASSIGE and PASTE. Across all sites, we used either PEmax, or a PE6 variant to install the *attP* landing site (Fig. 6a). We integrated (1) a 6.1-kb plasmid encoding *GBA1* cDNA

(Δ exon 1) into intron 1 of *GBA1*, (2) an 8.8-kb plasmid encoding *FANCA* cDNA (Δ exon 1) into intron 1 of *FANCA*, (3) a 5.9-kb plasmid encoding a CD19-CAR cassette[67,68] into the 5′ UTRs of *TRAC* and *B2M*, (4) a 7.1-kb plasmid encoding the human *Factor IX* (*F9*) cDNA (Δ exon 1) into intron 1 of *ALB* and (5) a 5.3-kb plasmid encoding *Smn1* cDNA (Δ exon 1) into intron 1 of *Smn1*. In all cases, eePASSIGE resulted in the highest integration efficiencies (32% average integration and a minimum of 23% integration across all sites). At *GBA1*, *B2M* and *ALB*, therapeutic cargoes were integrated with >30% efficiencies, and at *FANCA*, cargo knock-in reached 46% (Fig. 6a). Consistent with observations in other genomic loci and human cell types, PASTE yielded the lowest integration efficiencies among all tested methods, averaging 4.4%.

To assess whether the knock-in of therapeutic cargoes led to protein production, we assessed the integration and expression of *F9* in hepatocyte-derived HuH7 cells where albumin is expressed. We installed *attB* into intron 1 of *ALB*, and integrated a minicircle DNA encoding an *attP* motif, a partial *F9* intron 1 splice acceptor-containing

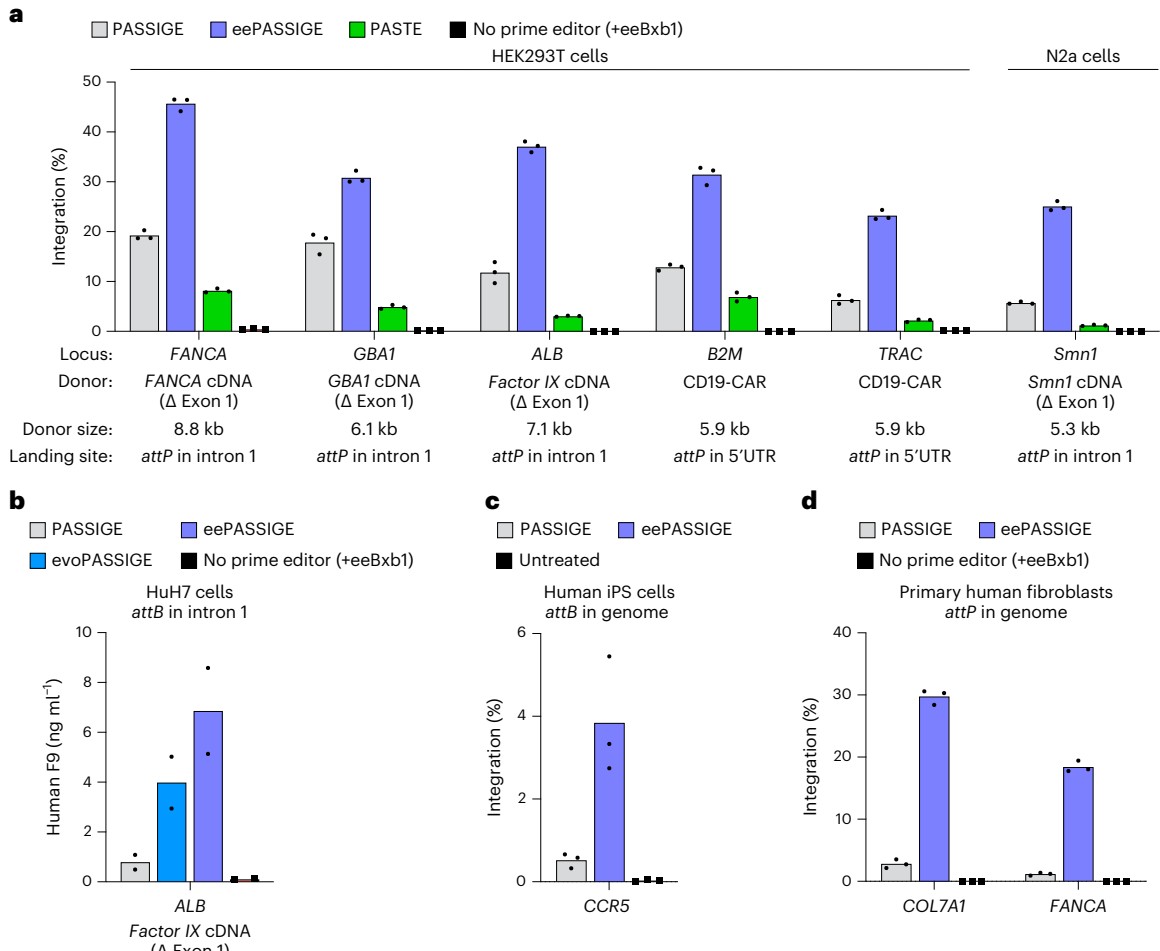

**Fig. 6 | Assessing the therapeutic potential of PASSIGE variants. a**, Absolute integration efficiencies for PASSIGE, eePASSIGE and PASTE when integrating therapeutically relevant cDNA cargoes into multiple loci in HEK293T and N2a cells. **b**, F9 protein measurement via ELISA assay. HuH7 cells were passaged 72 h after transfection with PASSIGE, evoPASSIGE and eePASSIGE. Day 9 post-transfection, media supernatants were collected from each condition and used for F9 ELISA assay. **c**, Absolute integration efficiencies for PASSIGE and eePASSIGE when integrating a 5.6-kb donor plasmid in human iPS cells with pre-installed *attB* sequence in the *CCR5* locus. The Bxb1 variant was delivered as an mRNA. **d**, Absolute integration efficiencies for PASSIGE and eePASSIGE when integrating a 5.8-kb DNA donor into the *COL7A1* and *FANCA* loci in primary human fibroblasts. The *attP* sequence was installed into intron 4 and intron 1 of the *COL7A1* and *FANCA* loci, respectively. PEmax mRNA, recombinase mRNA, synthetic pegRNAs and donor-encoding integrase-deficient lentivirus were all delivered via a single electroporation. For PASSIGE and PASTE experiments in **a** and **b**, all components were delivered using single transfection. For the negative control, either all components except the prime editor protein were delivered into cells with eeBxb1 recombinase (**a,b** and **d**) or an untreated sample was used (**c**). The bars reflect the mean of either three (**a,c** and **d**) or two (**b**) independent replicates and dots indicate individual replicate values of either *n* = 3 (**a,c** and **d**) or *n* = 2 (**b**). Integration efficiency (**a,c,d**) was assessed by ddPCR analysis as described in Supplementary Note 1.

sequence, the *F9* cDNA lacking exon 1 and a 3′ UTR sequence (Extended Data Fig. 9a). After cargo knock-in at intron 1, splicing between the secretion signal of *ALB* exon 1 and the integrated *F9* cDNA leads to F9 expression[24,50]. Enzyme-linked immunosorbent assay (ELISA) on conditioned media 9 days after transfection showed average F9 levels of 0.79, 4.0 and 6.9 ng ml[−1] following PASSIGE, evoPASSIGE and eePASSIGE treatment, respectively. EvoPASSIGE and eePASSIGE thus showed 5.0-fold and 8.7-fold higher *F9* expression, respectively, than PASSIGE (Fig. 6b).

Collectively, these results demonstrate that evoPASSIGE and eePASSIGE are robust, programmable large DNA integration technologies capable of mediating targeted gene integration at a wide variety of therapeutically relevant loci with efficiencies suitable for many therapeutic applications.

### Integration in primary human cells and in human iPS cells

Having determined the ability of PASSIGE variants to achieve efficient targeted gene integration in HEK293T, N2a, and HuH7 cells, we next evaluated their performance in more diverse and therapeutically relevant cell types such as primary human fibroblasts and human induced pluripotent stem (iPS) cells. In primary human fibroblasts, we delivered all the necessary components for PASSIGE, evoPASSIGE, eePASSIGE and PASTE as plasmids and assessed the integration of a 3.0-kb donor plasmid at the *AAVS1* site. Although integration efficiencies were substantially lower than those seen in previous cell types, we observed consistent trends in the performance of all four systems: eePASSIGE demonstrated the highest integration, followed by evoPASSIGE, PASSIGE and finally PASTE (Extended Data Fig. 9b). In human iPS cells homozygous for *attB* insertion at the *CCR5* locus, we delivered the recombinase variant as messenger RNA and a 5.6-kb donor DNA plasmid and observed a 7.3-fold improvement in integration efficiency with eeBxb1 (3.8%) compared with WT Bxb1 (0.52%) (Fig. 6c).

We reasoned that the low integration efficiencies observed above were due to DNA-mediated activation of intrinsic cellular defence mechanisms and cytotoxicity in primary cells[69]. To circumvent this issue, we employed a non-plasmid delivery approach in primary human

fibroblasts, where we delivered prime editor and recombinase mRNAs, synthetic pegRNAs, and an integrase-deficient lentiviral vector (IDLV) encoding the donor sequence. Strikingly, switching to this delivery modality substantially improved integration efficiencies: eePASSIGE exhibited 30% and 18% average integration in the *COL7A1* and *FANCA* loci, respectively, a 14-fold improvement over PASSIGE (Fig. 6d). Additionally, a PASSIGE and eePASSIGE exhibited off-target integration rates of 0.008% and 0.004%, respectively, across the two UDiTaS-nominated off-target sites OT1 and OT2 (Extended Data Fig. 9c). Taken together, these results underscore the potential of eePASSIGE for robust targeted gene integration in a variety of cell types, while also highlighting the importance of minimizing donor DNA-triggered cellular toxicity.

## Discussion

Targeted integration of large DNA payloads into the genome has been a long-standing challenge, with existing approaches such as PASSIGE[24], PASTE[26], CRISPR-associated transposases[19–23] and nuclease-mediated integration[11,12] suffering from modest efficiencies or high frequencies of undesired byproducts. Here, we used phage-assisted evolution to substantially enhance the activity of the Bxb1 recombinase for large DNA cargo integration in mammalian cells. In HEK293T cells with pre-installed recombinase attachment sites, evolved Bxb1 variants achieved up to 60% integration of a 5.6-kb plasmid compared with 18% observed with the WT enzyme. We predict that the evolved mutations may enhance integration by improving enzyme stability, catalysis or attachment site binding. However, we cannot exclude the possibility that they may also enhance recombination through alternative mechanisms, such as facilitating Bxb1 oligomerization.

When combined with prime-editing installation of recombinase landing sites in the PASSIGE system, evoBxb1 variant (V74A) demonstrates a 2.7-fold average improvement in donor integration from a single catalytic domain mutation, while the eeBxb1 variant (V74A, E229K and V375I) generated by rationally combining evolved mutations from distinct domains of the enzyme demonstrates a 4.2-fold average improvement over PASSIGE. Evo- and eePASSIGE show improvements in integration at all 12 genomic loci tested across three mammalian cell lines, can efficiently integrate cDNA cassettes into six therapeutically relevant endogenous genomic sites, can integrate gene cargoes that produce protein, and substantially outperform PASSIGE in diverse cell types of greater therapeutic relevance, such as primary human fibroblasts and iPS cells. In fibroblasts, eePASSIGE can achieve targeted gene integration efficiencies of 30% using integrase-deficient lentivirus as a donor. Consistent with previous studies, our results indicate that the delivery modality is a key determinant of editing efficiencies in primary cells, with plasmid DNA being poorly tolerated[69]. Exploring alternative delivery strategies, especially for the DNA donor, may further enhance PASSIGE performance in primary cells and will probably be crucial to maximize the therapeutic potential of any large-gene integration technology.

The PASSIGE variants developed in this study show large improvements over other programmable gene integration methods including PASTE, with evo- and eePASSIGE offering a 9.1-fold and 16.2-fold average improvement in integration across all 12 sites tested, respectively. PASTE did not outperform PASSIGE at any site tested in this study and exhibited on average 3.3-fold lower donor knock-in. This deficit primarily arises from a reduction in prime editing efficiencies, as PASTE installed recombinase landing sites 1.9-fold less efficiently than untethered PEmax, while maintaining similar recombination efficiencies as PASSIGE, which like PASTE, uses WT Bxb1 recombinase. Additionally, we note that one of the ddPCR probes used in the original report to quantify PASTE integration efficiency at multiple sites[26] does not exclusively report integrated DNA product formation and shows high background in negative controls lacking prime editor or using dead Bxb1 recombinase (Extended Data Fig. 10a,b). This background may partially explain the disparity of PASTE performance in our hands compared with the previous report.

Off-target integration assays of the evolved and engineered recombinases demonstrate that when installing *attB* into the genome, WT Bxb1, evoBxb1 and eeBxb1 can mediate off-target integration at genomic sites highly homologous to *attB*. In contrast, when installing *attP* into the genome, minimal off-target integration was observed for all variants using ddPCR. Based on these findings, we recommend installing *attP* into the genome via prime editing and using the eeBxb1 variant. At 7/12 sites tested in this study, installing *attP* into the genome led to higher PASSIGE-mediated integration efficiencies compared to installing *attB* (Fig. 4d) and at 11/12 sites, installing *attP* and using eeBxb1 resulted in integration efficiencies ranging from ~15% to 46% (Fig. 4a). At the 12th site in which donor integration was low, *attP* installation into the genome was substantially lower (33%) compared with *attB* installation (88%) (Supplementary Table 5), suggesting that additional pegRNA optimizations could potentially improve integration at this site. Additionally, we demonstrate that commonly used off-target nomination methods such as UDiTaS can nominate false off-target hits for large-gene integration technologies. The development of future integrase-based therapeutics would benefit from a genome-wide off-target nomination method with a lower false discovery rate.

To our knowledge, the evolved and engineered Bxb1 variants generated in this study enable the most effective programmable gene integration in mammalian cells reported so far, consistently achieving over 30% (and up to 46%) average donor gene integration at various safe harbour and therapeutically relevant loci in mammalian cells. In addition to PASSIGE strategies, we anticipate that these recombinase variants may also be applied to modify DNA for various applications in synthetic biology, biotechnology and cell and gene therapies. Finally, the rapid continuous evolution system developed in this study may also be used to improve the activity of other recently discovered LSRs to enhance their activity[26,64].

## Methods

### General methods and molecular cloning

Gibson assembly was used to clone all plasmids. Briefly, for Gibson cloning, fragments were obtained from PCR amplification, plasmid vector digestion or synthetic gene fragments and assembled using NEBuilder Hifi DNA assembly master mix (New England Biolabs). PCR was performed using Phusion U Hot Start II DNA polymerase (Thermo Fisher Scientific), Phusion U Green Multiplex PCR Master Mix (Thermo Fisher Scientific) or Q5 Hot Start High-Fidelity 2× Master Mix (New England Biolabs). DNA oligonucleotides were obtained from either Integrated DNA Technologies (IDT) or Eton-Biosciences. Synthetic gene fragments were obtained from either IDT or Genscript. Plasmids for mammalian expression of Bxb1 variants were cloned into the pCMV-Bxb1 vector backbone (Addgene, #182142). Plasmids expressing pegRNAs were cloned by assembling PCR-amplified pegRNA backbone (forward primer: 5′-GCTCGAGGTACCTCTCTA-3′, reverse primer: 5′-GAAATACTTTCAAGTTACGG-3′) or BsaI-digested pegRNA backbone (Addgene, #132777) with pegRNA-encoding eblocks ordered from IDT. DNA donor plasmids for mammalian cell experiments were cloned by assembling PCR-amplified fragments or synthetic gene fragments into either Factor IX donor vector backbone (Addgene, #182141) or *attB*-puro donor vector backbone (Addgene, #181923) digested by restriction enzymes. All prime editor variants used in this study (PEmax and PE6b-d) are available on Addgene (#174820, 207852–207854). Constructs for PASTE experiments were obtained from Addgene: PASTE v3 (#179105), PASTE DNA donor plasmid (#179115), ACTB atgRNA (#179108) and ACTB nicking single guide RNA (sgRNA) (#179109). All vectors for mammalian cell experiments were purified using Plasmid Plus Midiprep kits (Qiagen), QIAprep Spin Miniprep kits or Qiagen Plasmid Plus 96 Miniprep kit. Sequences of all pegRNA and sgRNA constructs and PE6 variants used in this work are listed in Supplementary Table 8.

## General mammalian cell culture conditions

HEK293T cells (American Type Culture Collection (ATCC) CRL-3216), N2A cells (ATCC, CCL-131), HuH7 cells (a gift from Erik Sontheimer's group, originated from ATCC) and HEK293T clonal cell lines with either pre-installed *attP* at *AAVS1* or *attB* at *CCR5* were cultured in Dulbecco's modified Eagle medium plus GlutaMAX (Thermo Fisher Scientific) supplemented with 10% (v/v) foetal bovine serum (Thermo Fisher Scientific). Clonal cell lines were generated using twin prime editing as previously described[24]. All cell lines were maintained and cultured at 37 °C with 5% $CO_2$, authenticated by their respective suppliers, and tested negative for mycoplasma.

## Phage plaquing

Plaque assays were performed to check for phage that cheat the selection (for example, by integrating gIII onto the SP), to measure phage titres, and for bacteriophage cloning. An overnight culture of host cells was diluted 50-fold in Davis rich medium (DRM) with carbenicillin and grown at 37 °C with shaking at 225 r.p.m. until $OD_{600}$ reached 0.3–0.8. Phage were serially diluted by a factor of ten in water, up to $10^6$-fold, and four different dilutions were then chosen for plaquing. Plates for plaquing were made by pipetting ~1 ml of molten 2× yeast extract tryptone (YT) agar mixed with 0.04% Bluo-gal (Gold Biotechnologies) into a 12-well plate (Corning). Top agar was made by combining 2× YT medium and agar (2:1 ratio) and stored at 55 °C until use. To plaque, 100 µl of host cells, 10 µl of serially diluted phage and 500 µl of top agar were mixed and quickly added onto the solid agar in the 12-well plate. After the top agar solidified, plates were incubated overnight at 37 °C.

## Preparation and transformation of chemically competent cells

Strain S2060 was used for all evolution experiments. To make competent cells, an overnight culture of bacteria was diluted 50-fold into 30 ml of 2× YT media with the appropriate antibiotics and grown at 37 °C with shaking at 225 r.p.m. until $OD_{600}$ reached 0.3–0.4. The cells were centrifuged for 10 min at 4,000$g$ at 4 °C and the pellet was resuspended in 3 ml of cold TSS media (Luria-Bertani (LB) medium supplemented with 5% v/v dimethylsulfoxide (DMSO), 10% w/v (polyethylene glycol) PEG 3350 and 20 mM $MgCl_2$) on ice. The resuspended cells were aliquoted into 100 µl volumes, frozen in dry ice, and stored at −80 °C. To transform cells with the appropriate plasmids, 1–5 µl of each plasmid, 20 µl of 5× KCM solution (500 mM KCl, 150 mM $CaCl_2$ and 250 mM $MgCl_2$), 100 µl of chemically competent cells and 80 µl of water were mixed and incubated on ice for 10 min. Cells were heat shocked at 42 °C for 90 s then 1 ml of SOC medium (New England Biolabs) was added for recovery. Cells were recovered at 37 °C with shaking at 225 r.p.m. for 1–2 h before plating.

## Bacteriophage cloning

Cloning of Bxb1 phage was performed using Gibson assembly of PCR fragments, as previously described[70]. Following assembly, the reaction was transformed into chemically competent S2060 *E. coli* host cells containing plasmid pJC175e, which encodes gIII under the phage-shock promoter and allows for activity-independent phage propagation[71]. After transformation, the cloned phage in *E. coli* was grown first for 15 min in DRM media without antibiotics at 37 °C, and then overnight in media with carbenicillin. Bacteria were centrifuged for 3 min at 8,000$g$ and plaqued in host strain S2060 transformed with pJC175e. The next day, individual plaques were picked and grown in DRM with carbenicillin. Once the culture reached late growth phase, bacteria were centrifuged for 10 min at 4,000$g$ and the supernatant containing phage was isolated. Colony PCR was performed using primers (5′-GCTGTCTTTCGCTGCTGAGG-3′ and 5′-GCAAGAAACAATGAAATAGCAATAGCTATCTTACCGAAGCCC-3′) and sent for Sanger sequencing (Quintara Biosciences).

## PANCE

Strain S2060 cells were transformed with the appropriate P1 and P2 plasmid (Fig. 1c) and made chemically competent. Chemically competent host cells were transformed with MP6 (ref. 72) as described above and plated on 2× YT agar with 100 mM glucose. The next day, several colonies were picked and grown overnight at 37 °C with shaking at 225 r.p.m. The overnight culture was then diluted by 50-fold in DRM with the appropriate antibiotics and grown at 37 °C with shaking at 225 r.p.m. until $OD_{600}$ reached 0.3–0.4. To induce MP6 expression, arabinose was added to reach a final concentration of 20 mM. Immediately, 1 ml of this culture was mixed with 10 µl of the SP in a 96-well plate (Avantor, VWR) and grown overnight for 12–18 h at 37 °C with shaking at 225 r.p.m. The plate was centrifuged for 10 min at 4,000$g$ and phage were isolated from the supernatant. Isolated phage were used to infect the next PANCE passage until a noticeable change in phage propagation was observed. After each PANCE passage, titres of isolated phage were determined by quantitative PCR (qPCR; described below) and this information was used to determine the selection strategy for the next passage. After evolution, phage were plaqued in (1) host strain 2060 to check for cheater phage that might have recombined with gIII and (2) host strain S2060 transformed with pJC175e to determine phage titres. Individual plaques were PCR amplified using the same primers noted in 'Bacteriophage cloning' and sent for sanger sequencing. Mutation tables were generated using Mutato.

## Assessment of PANCE titres using qPCR

To generate a standard curve for qPCR, a standard phage sample of high titre (~1 × $10^{10}$ plaque-forming unit (p.f.u.) per ml as determined by plaquing) was serially diluted by a factor of 10, up to $10^8$-fold, in water and carried forward along with the isolated phage from PANCE. First, 50 µl of phage was lysed for 30 min at 80 °C. To remove the genome of replication-incompetent polyphage, 5 µl of the lysed phage was mixed with 44.5 µl of 1× DNase buffer and 0.5 µl of DNase I enzyme (New England Biolabs). This mixture was incubated first for 20 min at 37 °C, then for 20 min at 95 °C. Finally, 1.5 µl of this reaction was combined with 14 µl of Q5 Hot Start High-Fidelity 2× Master Mix, SYBR Green (Invitrogen), 0.125 µl each of 100 µM M13 forward and reverse primers (5′-CACCGTTCATCTGTCCTCTTT-3′ and 5′-CGACCTGCTCCATGTTACTTAG-3′) and water to achieve a final volume of 28 µl. The qPCR was performed with the following conditions: 98 °C for 2 min, and then 45 cycles of 98 °C for 10 s, 60 °C for 20 s and finally 72 °C for 15 s. Standard curve was generated using Cq values and phage titres of PANCE pools were determined accordingly.

## PACE

PACE experiments were performed as previously described[40]. Briefly, as explained above for PANCE, S2060 host cells with the appropriate plasmids (P1, P2 and MP6) were grown until $OD_{600}$ reached 0.3–0.4. Next, the chemostat and all four lagoons were filled with 80 ml and 15 ml of this cell culture, respectively. To maintain an $OD_{600}$ between ~0.2 and 0.8 in the chemostat, an appropriate flow rate (~80 ml $h^{-1}$) was established to continuously dilute the cells with fresh media (59 g Harvard Custom Media C, 50 µl of 0.1 M $CaCl_2$, 120 µl of trace metal solution, 400 mg chloramphenicol pre-dissolved in 4 ml of ethanol, 500 ng carbenicillin, 1 g spectinomycin, 500 ml deionized water and 20 l Harvard Custom Media A solution). A flow rate of 7.5 ml $h^{-1}$ was set in the lagoons, and cells were induced with 10 mM arabinose. This set-up was allowed to equilibrate for at least an hour before SP infection. Next, all pumps were turned off and the lagoons were infected with ~$10^8$ p.f.u. $ml^{-1}$ phage. After 10 min, the pumps were turned back on to start the evolution and samples (~500 µl) were taken from the waste line of each lagoon for plaquing ($T = 0$ timepoint). As indicated in Extended Data Fig. 2b, samples from the lagoons were taken throughout evolution at different timepoints and flow rate in the lagoons was increased with time. After collection, samples were centrifuged for 3 min at 8,000$g$ and the

supernatant was collected and plaqued as described above in PANCE to check for cheater phage and determine phage titres. Individual plaques were picked, PCR amplified, sequenced and analysed as described in the 'PANCE' section.

### Structure analysis via AlphaFold2

All protein structures were predicted using AlphaFold via ColabFold v1.5.3 (refs. [42,73]). ChimeraX[74] was used to align structures.

### Mammalian cell culture transfection

All transfections were performed in 96-well poly-D-lysine coated plates (Corning). HEK293T, N2a and HuH7 cells were seeded at a density of 10K per well, 20K per well and 15K per well, respectively. After 16–24 h, cells were transfected at approximately 50–60% confluency with 0.5 μl of Lipofectamine 2000 (Thermo Fisher Scientific), according to the manufacturer's protocols. In HEK293T cells for PASSIGE, evoPASSIGE or eePASSIGE, 100 ng of prime editor plasmid, 10–20 ng of each pegRNA plasmid, 100 ng of Bxb1 plasmid and 150 ng of donor plasmid were transfected. For PASTE experiments, 100 ng of prime editor plasmid and 100 ng of Bxb1 plasmid were replaced with 200 ng of PASTE v3 construct (Addgene #179105). For PE3 experiments, 20 ng of pegRNA plasmid and 10 ng of nicking sgRNA plasmid were used. To assess attachment site installation using twinPE in Supplementary Table 5, 100 ng of prime editor or 200 ng of PASTE v3 along with 10 ng of each pegRNA were transfected. In N2a cells, 25 ng of each pegRNA was transfected, and the amount of all other components were kept the same as above. In HuH7 cells, 50 ng of Bxb1 plasmid, 50 ng of prime editor, 75 ng of F9 DNA donor plasmid or minicircle plasmid (see below for preparation) and 10 ng of each pegRNA plasmid were transfected.

### Genomic DNA preparation for HTS and ddPCR

For the extraction of genomic DNA, media were removed from cells cultured for 3 days after transfection. The cells were washed with 1× PBS solution (Thermo Fisher Scientific) before adding 50 μl of freshly prepared lysis buffer (10 mM Tris–HCl at pH 8.0, 0.05% SDS, 25 μg ml⁻¹ of proteinase K; Thermo Fisher Scientific) into each well. This mixture was incubated at 37 °C for 1–2 h, transferred into a 96-well PCR plate and then heated at 80 °C for 30 min to inactivate proteinase K. This genomic DNA mixture was directly used as a template for HTS (see below).

To prepare genomic DNA for ddPCR, the above mixture was further purified using the DNAdvance kit from Beckman Coulter (A48705), according to the manufacturer's protocol. Briefly, 60 μl of Pre-Bind PBBA buffer was mixed with 30 μl of cell lysate. Next, 60 μl of Bind BBE buffer with beads was added and thoroughly mixed. The beads were washed twice with 200 μl of freshly prepared 70% ethanol and air dried for 5 min before eluting in 20–30 μl of water or elution buffer. Final DNA concentrations were determined using a NanoDrop (Thermo Fisher Scientific).

### HTS and analysis of genomic DNA samples

Illumina Miseq was used to sequence PCR amplified genomic sites of interest, as previously described[24]. First, a 25 μl PCR reaction (PCR1) was performed using 0.5 μM of each forward and reverse primer containing Illumina adaptors (Supplementary Table 9), Phusion U Hot Start II DNA polymerase (Thermo Fisher Scientific) 1 μl of extracted genomic DNA (see above) and water. PCR1 was performed with the following conditions: 98 °C for 2 min, 30 cycles of 98 °C for 10 s, 61 °C for 20 s and 72 °C for 30 s, and finally a 72 °C extension for 2 min. Next, a second 25 μl PCR reaction (PCR2) was performed using 0.5 μM of unique forward and reverse Illumina barcoding primer pair, Phusion U Hot Start II DNA polymerase, 1 μl of PCR1 product and water. PCR2 was performed with the following conditions: 98 °C for 2 min, 10 cycles of 98 °C for 10 s, 61 °C for 20 s and 72 °C for 30 s, and finally a 72 °C extension for 2 min.

To reduce PCR bias when assessing allelic distribution post-integration in Fig. 4e and Extended Data Fig. 6b,c, UMIs were applied, using a modified protocol. First, linear amplification was performed in a 25 μl reaction using 1 μl of lysed genomic DNA, 0.1 μM of a 15-nt UMI containing forward primer (Supplementary Table 9) and Phusion U Hot Start II DNA polymerase (Thermo Fisher Scientific) with the following conditions: 11 cycles of 98 °C for 1 min, 61 °C for 25 s, and 72 °C for 1 min. The amplified products were bead purified using 1.6× AMPure beads (Beckman Coulter) and eluted in 10 μl water. PCR1 was performed as described above with two modifications: (1) 2 μl of the purified linearly amplified product was used as the template and (2) PCR was carried out for 31 cycles instead of 30. Similarly, PCR2 was carried out as described above, but with two additional cycles (12 cycles instead of 10).

Products from PCR2 were combined, electrophoresed on a 1.5% agarose gel and extracted using QIAquick Gel Extraction kit (Qiagen). DNA concentration of the resulting library was quantified using a Qubit dsDNA High Sensitivity Assay kit (Thermo Fisher Scientific). The library was then normalized and sequenced on an Illumina Miseq instrument according to the manufacturer's instructions. Individual sequencing reads were demultiplexed using Miseq Reporter (Illumina).

CRISPResso2 (ref. [75]) was used to analyse HTS reads, as previously described[24]. For UMI-barcoded samples, AmpUMI[57] was used to deduplicate the sequencing reads before further analysis. Briefly, for all experiments, CRISPResso2 was executed on homology-directed repair (HDR) mode with the following parameters for each edit: 'e' specified the amplicon expected after editing, 'qwc' specified the quantification window, which was set between 10-bp upstream of the first nick and 10-bp downstream of the second nick, 'discard_indel_reads' was set to TRUE and 'q' was set to 30. Per cent editing was quantified by multiplying the ratio of non-discarded HDR aligned reads and total reads aligned to all amplicons by 100. Per cent indels were quantified by multiplying the ratio of indel-containing discarded reads and total reads aligned to all amplicons by 100.

### ddPCR analysis to assess integration efficiency

ddPCR was used to determine the abundance of genomic DNA fragments containing the integrated donor at target loci in comparison to a reference gene. Approximately 50–200 ng of bead-purified DNA was added to a 25 μl reaction mixture containing (1) 2× ddPCR Supermix for Probes (no dUTP) (Bio-Rad, 1863025), (2) reference gene primer pair + probe master mix from Bio-Rad, (*ACTB* (unique assay ID: dHsaCNS141996500) or *GAPDH* (unique assay ID: dHsaCNS794216737) for human cells; *Tfrc* (unique assay ID: dMmuCNS420644255) for mouse cells) and (3) post-integration junction primer pair and probe (900 nM each primer, 250 nM probe). Droplet generation, PCR and droplet reading steps were all performed using the Bio-Rad QX ONE platform. PCR was performed with the following conditions: 95 °C for 10 min, 50 cycles of 94 °C for 30 s and 58 °C for 2 min, and finally 98 °C for 10 min. Data from ddPCR were analysed using the QX ONE software 1.3 Standard Edition. Supplementary Note 1 highlights an example of how thresholds for each channel were determined to avoid false positives observed from plasmid recombined products. To determine per cent integration, the ratio between the concentrations (copies per μl) of the genome–donor junction and reference gene was multiplied by 100. For off-target integration, a probe that binds to either the genome–donor junction or genomic locus was used and in all cases, a positive DNA sequence was included to validate the design of the primer pair and probe, and a dead recombinase control was used to determine the presence of false positives. When measuring on-target integration, a genome–donor junction probe was used in all cases. All primer pairs, probes and reference primer pair + probe master mixes used in this study are listed in Supplementary Table 10.

### Analysis of chromatin accessibility on gene integration efficiency

Normalized ChIP-sequencing signals (fold change of ChIP signals over input) of H3K27ac, H3K36me3, H3K4me1, H3K4me3 and H3K9me3 in

HEK293T cells were downloaded from the ENCODE Portal[59]. The average signal for each histone modification was calculated within a 1-kb window centred around each target site. Pearson correlation analysis was performed using PRISM (GraphPad) to assess the relationship between integration efficiency across ten genomic sites installed with *attP*, and the corresponding epigenetic marker signal. For the Pearson analysis with recombination efficiencies, data from seven genomic sites were used instead. Statistical significance was calculated using Student's unpaired two-tailed *t*-test.

## Flow cytometry to assess off-target integration

To assess off-target integration, 15,000 HEK293T cells were transfected with 100 ng of Bxb1 variant and 150 ng of a 3.2-kb donor DNA plasmid with either an *attP* or *attB* Bxb1 landing site encoding mCherry under the cytomegalovirus (CMV) promoter. Cells were passaged for 2 weeks to dilute the plasmid DNA. Transfected cells were trypsinized, resuspended in PBS solution and assessed for mCherry fluorescence using the CytoFLEX S Flow Cytometer (Beckman Coulter) software. Supplementary Note 3 highlights an example of how the cells were gated.

## UDiTaS sample preparation and sequencing

A total of 10,000 HEK293T cells were plated into 96-well plates and transfected with 100 ng of prime editor plasmid, 10 ng of each pegRNA plasmid, 100 ng of Bxb1 plasmid and 150 ng of puromycin-encoding donor plasmid. For PASTE treated samples, the prime editor and Bxb1 plasmids were replaced with 200 ng of the PASTE v3 construct. Each donor plasmid was uniquely barcoded with a 10-nt UMI sequence. After 4 days, cells from 6 wells were pooled, expanded, and selected for 2 weeks using 2 mg ml$^{-1}$ puromycin. The genomic DNA was collected using the Monarch HMW DNA Extraction kit (New England Biolabs) and eluted in water.

Samples were prepared as previously described[64] with a few modifications (schematic shown in Supplementary Note 4). Tn5 was purified by the Harvard Structural and Chemical Biology Center as previously described[76] and stored at −20 °C at a concentration of 3.1 mg ml$^{-1}$ in 50 mM HEPES–HCl, 100 mM NaCl, 0.1 mM ethylenediaminetetraacetic acid, 10% glycerol, 0.1% Triton X-100 and 1 mM dithiothreitol at a pH of 7.2. Adaptors were annealed by combining 50 µl of each top and bottom 100 µM oligos, heating the mixture to 95 °C and then gradually cooling it down to 12 °C over the course of ~12 h. The transposome was assembled by incubating a mixture containing 90 µl of Tn5 and 10 µl of annealed adaptors at 23 °C for 1 h with gentle shaking. The assembled transposome was stored at −20 °C until further use. Samples were tagmented in a 20 µl reaction volume by adding 100 ng of extracted genomic DNA (quantified using Qubit dsDNA High Sensitivity Assay kit, Thermo Fisher Scientific), 2 µl of the assembled transposome and 4 µl of 5× N-Tris(hydroxymethyl) methyl-3-aminopropanesulfonic acid–dimethylformamide (50 mM N-Tris(hydroxymethyl)methyl-3-aminopropanesulfonic acid NaOH, 25 mM MgCl$_2$ and 50% v/v dimethylformamide, pH 8.5). This mixture was incubated for 7–15 min at 55 °C in a thermocycler, immediately quenched with 0.2% SDS at room temperature for 5 min, bead purified using 0.9× SPRIselect beads (Beckman Coulter) and eluted in 22 µl water. Tagmented products were evaluated using the Agilent High Sensitivity D5000 ScreenTape kit to confirm size of ~1–2 kb. Next, PCR was performed using 9 µl of the tagmented product, 12.5 µl of Platinum Superfi Master Mix (Thermo Fisher Scientific), 1.25 µl of DMSO, 0.5 µl of 10 µM outer-donor-specific primer (outer_donor), 0.25 µl of 10 µM outer-tagmented adaptor primer (outer_P5) and water to a volume of 25 µl with the following conditions: 98 °C for 2 min, 13 cycles of 98 °C for 10 s, 65 °C for 10 s, 72 °C for 90 s, and finally a 72 °C extension for 2 min. Amplified products were purified using 0.9× SPRIselect beads and eluted in 11 µl water. Next, 10 µl of this purified product was mixed with 25 µl of Platinum Superfi Master Mix, 2.5 µl DMSO, 5 µl of 10 µM nested donor-specific primer (inner_i7_donor), 2.5 µl of 10 µM nested

tagmented adaptor specific primer (inner_P5) and water to a volume of 50 µl. This mixture was used to perform a second PCR with the following conditions: 98 °C for 2 min, 18 cycles of 98 °C for 10 s, 65 °C for 10 s, 72 °C for 90 s. Four separate reactions were carried out for each condition. PCR2 products were combined, electrophoresed on a 1.5% agarose gel and the 300–500 bp band was extracted using QIAquick Gel Extraction kit (Qiagen). DNA concentration of the resulting library was quantified using the Qubit dsDNA High Sensitivity Assay kit. The library was normalized and sequenced on an Illumina Miseq instrument according to the manufacturer's instructions. Individual sequencing reads were demultiplexed using Miseq Reporter (Illumina). All adaptor and primer sequences used are provided in Supplementary Table 9.

## Computational analysis to map off-target sites after UDiTaS

UDiTaS analysis was performed using a custom bioinformatics pipeline available on GitHub following a previously described protocol[64] with a few modifications. Briefly, quality filtering and adaptor trimming were performed using fastp[77]. The default settings were used to define qualified bases ≥Q15 and to remove reads with >40% unqualified bases. Reads were filtered for the presence of both the donor-specific primer sequence and the attachment half-site from the donor plasmid, which are incorporated into the genome after recombinase-mediated on-target and off-target integration. Bases beyond the donor-specific primer were trimmed, as these were all artifacts resulting from a lack of template to sequence during HTS. Deduplication was performed by binning reads with identical UMI present in the donor plasmid, and retaining the most frequent sequence per group. Bins containing reads that aligned to either the full attachment site present in the donor plasmid, or sequences present in the pegRNA-donor plasmid recombined product were discarded to remove plasmid-contaminated reads.

The filtered and deduplicated reads were next aligned to both the human genome GRCh38 and the genome-integrated donor sequence using BWA-MEM[78]. Genomic coordinates were extracted for reads that aligned to both sequences to identify the integration site. The number of reads that aligned to each locus was counted. To account for sequencing errors that led to variability in the exact coordinates being found, BEDtools[79] was used to merge loci within 500 bp of each other. For sites where the identified genomic sequence was immediately adjacent to the attachment half-site that gets integrated into the genome, a predicted genomic pre-integration attachment site sequence was inferred to be the site where integration occurred. Up to one mismatch between the read and the genome was allowed to account for sequencing error. Sequences post-analysis are all reported in Supplementary Table 7 and were used to plot Fig. 5b and Extended Data Fig. 8c. Reads where the genomic sequence was not adjacent to the attachment half-site are talked about it more detail in Supplementary Note 4.

## Minicircle donor production

Minicircle donor DNA was prepared using the MC-Easy minicircle DNA Production kit (System Biosciences, MA925A-1). Briefly, the Factor IX donor plasmid (Addgene, #182141) was transformed into ZYCY10P3S2T Minicircle Production cells. Transformed ZYCY10P3S2T cells were inoculated in 2 ml of LB medium with kanamycin and grown for 1h at 30 °C. Then, 1 ml of this culture was inoculated into 200 ml of LB medium without antibiotics and grown overnight at 30 °C with shaking at 225 r.p.m. The next day, 200 ml of induction medium was added before the OD$_{600}$ of the overnight culture reached 0.6. This culture was further grown for 3 h at 30 °C and 1 h at 37 °C with shaking at 225 r.p.m. The Qiagen Plasmid Maxi kit was used to extract the minicircle DNA. To check the quality of the minicircle, 1 µg of the maxi-prepped product was linearized using restriction enzymes and electrophoresed on a 1% agarose gel. The F9 minicircle donor was used to transfect HuH7 cells as described above.

## Factor IX expression and quantification in HuH7 cells

Three days after transfection, HuH7 cells were passaged into a 48-well plate with fresh media and allowed to grow to confluence. ELISA was used to measure Factor IX concentration from conditioned media 9 days after transfection according to the manufacturer's protocol (Innovative Research Human Total Factor IX ELISA Kit, IHUFIXKTT).

## IVT of prime editor and recombinases

In vitro transcription (IVT) was performed as previously described[80]. Briefly, PEmax and Bbx1 variant were cloned into pT7 expression constructs (Addgene, #132777), which were used as templates to amplify the linear DNA for IVT. Primers noted in Supplementary Table 9 and the NEBNext High-Fidelity 2X PCR Master Mix (New England Biolabs) were used for PCR. Amplified products were purified using the QIAquick PCR purification kit (Qiagen). IVT was performed using the T7 high-yield RNA synthesis kit (New England Biolabs), following the manufacturer's instructions with three exceptions: (1) uridine was replaced with $N1$-methyl-pseudouridine (TriLink BioTechnologies), (2) capping was performed using CleanCap AG (TriLink BioTechnologies) and (3) the reaction was carried out for 4 h. Post-reaction, samples were treated with DNase (New England Biolabs) according to the manufacturer's instructions. RNA was precipitated by mixing the resulting samples with 0.5 volumes of lithium chloride, followed by two ethanol washes. The precipitated RNA was air dried, resuspended in water, normalized to 2 mg ml$^{-1}$ and stored at $-80$ °C.

## IDLV donor production

The D64V mutation was cloned into the integrase domain of the delta8.2 vector (Addgene, #8455) to generate delta8.2-D64V. For virus production, $6 \times 10^6$ HE293T/17 cells were seeded into 15-cm dishes. Twenty-four hours after seeding, cells were transfected using the following reagents per plate: 16.6 µg of transfer plasmid (Addgene, #89360), 12.5 µg of delta8.2-D64V, 8.4 µg of pMD2.G (Addgene, #386119), 1,250 µl of JetPRIME buffer and 50 µl of JetPRIME transfection reagent (Polyplus). Forty-eight hours post-transfection, the supernatant was collected and filtered through a 0.45 ml filter. IDLV was concentrated with 5× lentivirus precipitation solution (Alstem) following the manufacturer's instructions. Finally, the virus was concentrated 200-fold using primary human fibroblasts culture medium (see below).

## Primary human fibroblasts culture conditions and electroporations

Following the Declaration of Helsinki Principles, primary dermal fibroblasts were obtained and expanded. Cells were cultured at 37 °C with 5% CO$_2$ using 20% foetal bovine serum (Atlas Biologicals) in Alpha minimal essential medium containing: 1× penicillin/streptomycin/fungizone, 1× GlutaMAX, 1× antioxidant non-essential amino acids (all from Thermo Fisher Scientific), 0.5 ng ml$^{-1}$ fibroblast growth factor and 10 ng ml$^{-1}$ epidermal growth factor (Sigma-Aldrich). When cells reached >90% confluency, they were collected using trypsin and 100,000 cells were electroporated using the Neon Transfection System (Thermo Fisher) with a 10 µl tip and the following settings: 1,700 volts, 20 ms pulse and 1 pulse. For delivery of DNA, the doses were 100 ng of each epegRNA, 250 ng of PEmax and 300 ng of recombinase plasmid. For delivery of RNA, the doses were 0.9 µl of 100 µM of each epegRNA (a gift from Agilent, sequences in Supplementary Table 8) and 2 µg each of editor and recombinase mRNA prepared via IVT as described above. The integrase-deficient lentiviral donor was added 2 h after electroporation at a volume of 250 µl. Media were changed 72 h post-electroporation, and on day 5 the cells were collected. For the no PEmax negative control in Fig. 6d, media were changed on day 4 instead and the cells were collected a few hours later. Genomic DNA was isolated using the Monarch Genomic DNA Purification kit (New England Biolabs).

## Human iPS cell culture conditions and electroporations

Human iPS cells were grown in StemFlex™ medium (Thermo Fisher) on Geltrex-coated plates (Gibco). The human iPS cell line UCSD142i-86-1 was obtained from WiCell[81] and maintained at 37 °C in a humidified incubator containing 5% CO$_2$. Media were changed every other day. Before reaching confluency, cells were passaged 1:10 using Accutase™ (Innovative Cell Technologies). Then, 10 µM ROCK Inhibitor was included in the media for 2 days post splitting.

Twin prime editing was used to install attB with a GA central dinucleotide into the CCR5 locus as previously described[24]. An iPS cell clonal line with homozygous installation was isolated, and the genotype was confirmed by HTS. For Bxb1-mediated integration, 200,000 iPS cells were resuspended in 10 µl buffer R and combined with 2 µg recombinase mRNA prepared via IVT and 480 ng donor plasmid. Cells were electroporated using the Neon™ NxT system (Thermo Fisher) with the following settings: 1,400 V, 10 ms and 2 pulses. Following electroporation, 400,000 cells were plated in a Geltrex-coated 24-well plate containing 0.5 ml of StemFlex medium. To help cell attachment and prevent microbial contamination, 10 µM ROCK inhibitor and antibiotic–antimycotic (Gibco) were added to the recovery media. Media were changed 1–2 h post-electroporation and again on days 1 and 3 post-electroporation. Genomic DNA was isolated after 5 days using the DNeasy Blood and Tissue kit (Qiagen).

## Reporting summary

Further information on research design is available in the Nature Portfolio Reporting Summary linked to this article.

## Data availability

The main data supporting the results in this study are available within the paper and its Supplementary Information. High-throughput DNA sequencing FASTQ files are available from the National Center of Biotechnology's Information Sequence Read Archive under BioProject (PRJNA1042817). ddPCR and flow cytometry data are available from figshare via the identifier https://doi.org/10.6084/m9.figshare.24589083. Plasmids encoding evoBxb1 and eeBxb1 and a subset of pegRNA-encoding plasmids are available through Addgene. Other data are available from the corresponding author on reasonable request. Source data are provided with this paper.

## Code availability

Code used for processing and analysing HTS data are available at https://github.com/pinellolab/CRISPResso2. Code used to deduplicate sequencing reads before CRISPResso2 analysis are available in http://github.com/pinellolab/AmpUMI. Mutato code used to generate mutation tables from sequencing reads of evolved phage are available at https://hub.docker.com/r/araguram/mutato. Code used to analyse UdiTaS are available at https://github.com/Nicholas-Krasnow/integrato.

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

## Acknowledgements

We thank X. Zhang for contributions to assess the impact of chromatin accessibility on integration and recombination efficiencies. This work was supported by the National Institutes of Health grants R01 HL160970, U01 AI142756, R01 EB031172, RM1 HG009490 and R35 GM118062; the Bill and Melinda Gates Foundation and the Howard Hughes Medical Institute (HHMI). N.A.K. is supported by the National Science Foundation Graduate Research Fellowship Program. M.J.O. receives funding from the Bill and Melinda Gates Foundation, the Saint Baldrick's Foundation and the Kidz1stFund. J.T. receives funding from National Institutes of Health grant 5R01AR063070-08. This article is subject to HHMI's Open Access to Publications policy. HHMI laboratory heads have previously granted a nonexclusive CC BY 4.0 license to the public and a sublicensable license to HHMI in their research articles. Pursuant to those licenses, the author-accepted manuscript of this article can be made freely available under a CC BY 4.0 license immediately upon publication.

## Author contributions

S.P. and X.D.G. contributed equally. S.P. and X.D.G. cloned plasmids, designed and conducted protein evolution, protein engineering and mammalian cell experiments, and analysed data. N.A.K. analysed the UDiTaS data. A.M., B.J.S. and M.J.O. performed fibroblast experiments. Y.A.T. produced integrase-deficient lentivirus for fibroblast experiments. M.J.O. and J.T. advised on therapeutic applications for PASSIGE and supervised primary fibroblast experiments. T.B.M., and E.L.C. supervised iPS cell experiments. J.E.D., J.M. and S.E.P., assisted with experiments. D.R.L. supervised the research. S.P., X.D.G. and D.R.L. drafted the manuscript with input from all authors. Correspondence to David R. Liu (drliu@fas.harvard.edu).

## Competing interests

X.D.G., S.P. and D.R.L. have filed patent applications on this work. M.J.O. receives compensation as a consultant for Agathos Biologics. D.R.L. is a co-founder and consultant for Prime Medicine, Beam Therapeutics, Pairwise Plants, Chroma Medicine and Nvelop Therapeutics, and owns equity in these companies.

## Additional information

**Extended data** is available for this paper at https://doi.org/10.1038/s41551-024-01227-1.

**Correspondence and requests for materials** should be addressed to David R. Liu.

Circuit 1.1, *attB* in P1, *attP* in P2,  GT central dinucleotide

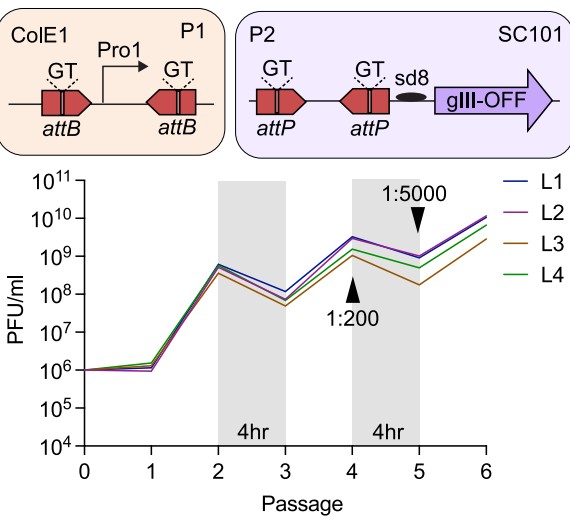

Circuit 1.2, *attB* in P1, *attP* in P2, GA central dinucleotide

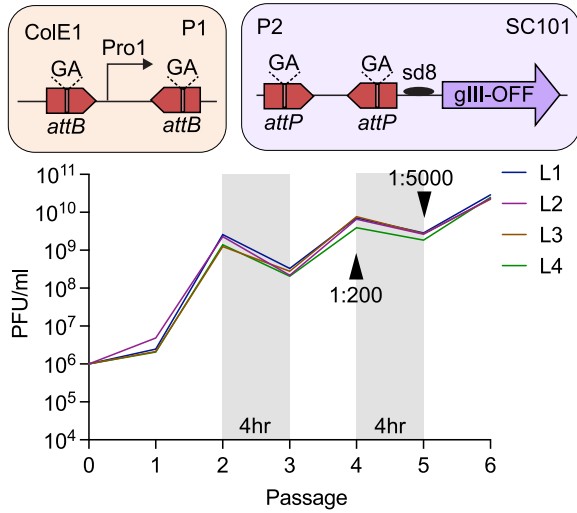

Circuit 1.3, *attP* in P1, *attB* in P2, GT central dinucleotide

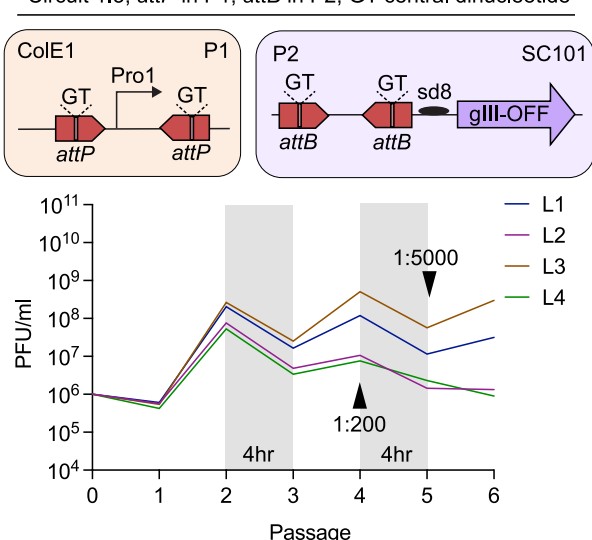

Circuit 1.4, *attP* in P1, *attP* in P2, GA central dinucleotide

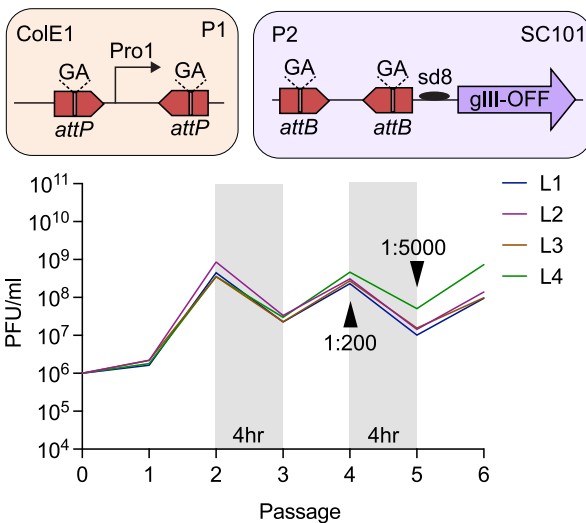

Circuit 2.1, *attB* in P1, *attP* in P2, GT central dinucleotide

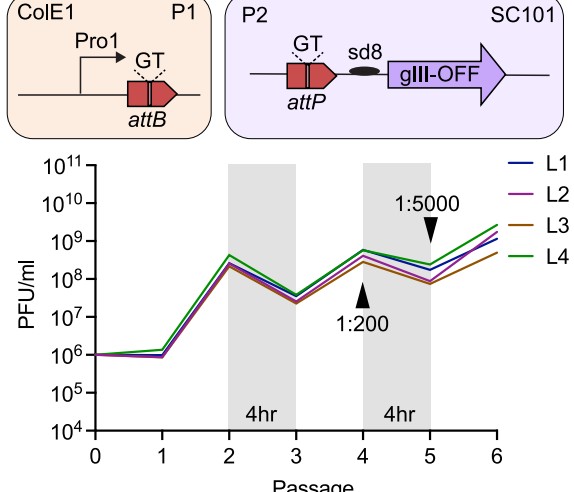

Circuit 2.2, *attB* in P1, *attP* in P2, GA central dinucleotide

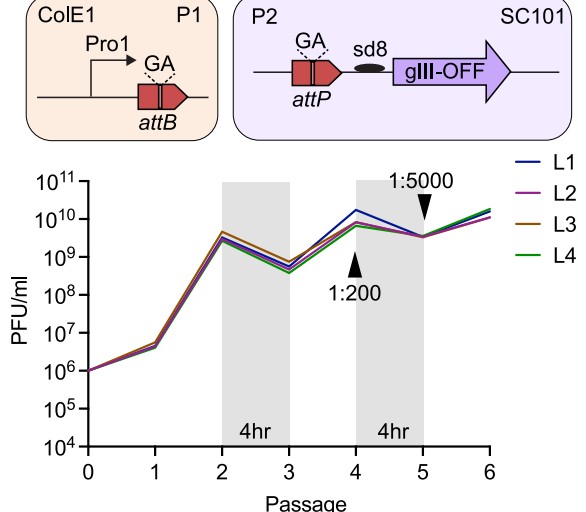

**Extended Data Fig. 1 | See next page for caption.**

**Extended Data Fig. 1 | Individual PANCE experiments for Bxb1 evolutions in Fig. 1d.** In circuits 1.1–1.4, two recombinase attachment sites are present in both plasmids, P1 and P2. Circuits 2.1, and 2.2 use one recombinase attachment site per plasmid. Circuits 1.1, 1.2, 2.1, and 2.2 have *attB* in P1 and *attP* in P2. Circuits 1.3, and 1.4 have *attP* in P1 and *attB* in P2. Circuits 1.2, 1.4, and 2.2 have a GA central dinucleotide in the attachment sites instead of GT, which is present in circuits 1.1, 1.3, and 2.1. PANCE traces for each lagoon (L1-L4) are shown. Selection stringency was modulated by decreasing the selection time and increasing dilution factor. Unless otherwise indicated, each PANCE passage was performed overnight, and phage were diluted 1:50 after each passage. PANCE titers were measured by qPCR as described in Methods.

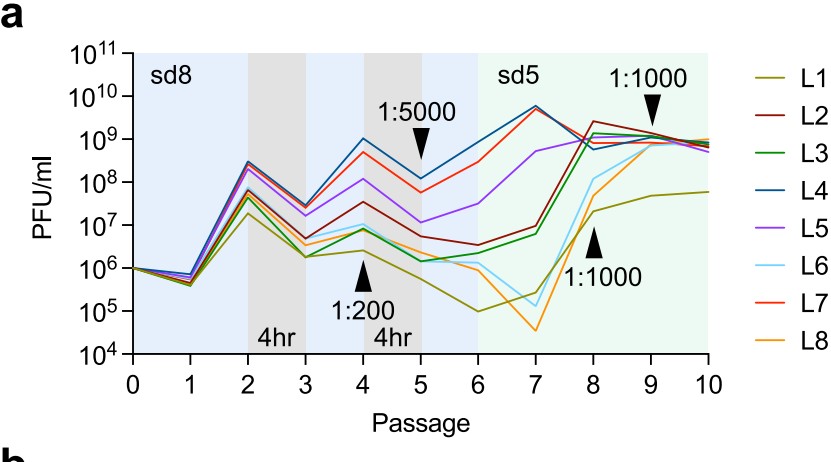

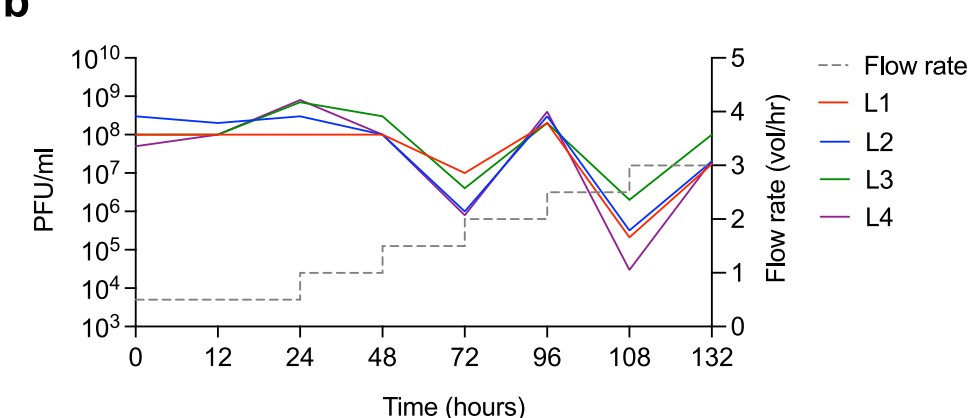

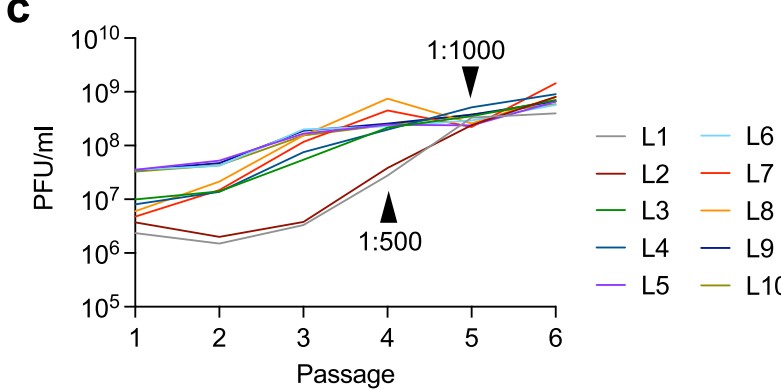

**Extended Data Fig. 2 | PANCE and PACE experiments for Bxb1 evolution in Fig. 2a. a**, PANCE traces in circuit 1.3. Traces for individual lagoons (L1-L8) are shown. Selection stringency was modulated by decreasing the strength of the ribosome binding site (RBS) from sd8 to sd5, decreasing selection time, and increasing dilution factor. **b**, PACE traces across four lagoons (L1-L4) using circuit 1.3. Phage pools obtained from PANCE in **a** were used to inoculate all lagoons. Selection stringency was modulated by increasing flow rate from 0.5 vol/hr to 3.0 vol/hr. **c**, PANCE traces for evolution where size of P1 was increased from 3.2-kB to 6.5-kB. Phage pools obtained from PACE in **b** were used to inoculate ten individual lagoons (L1-L10). Selection stringency was modulated by increasing dilution factor. For all PANCE experiments, unless otherwise indicated, each passage was performed overnight and phage were diluted 1:50 after each passage. PANCE titers were measured using qPCR and PACE titers were measured using plaquing, as described in Methods.

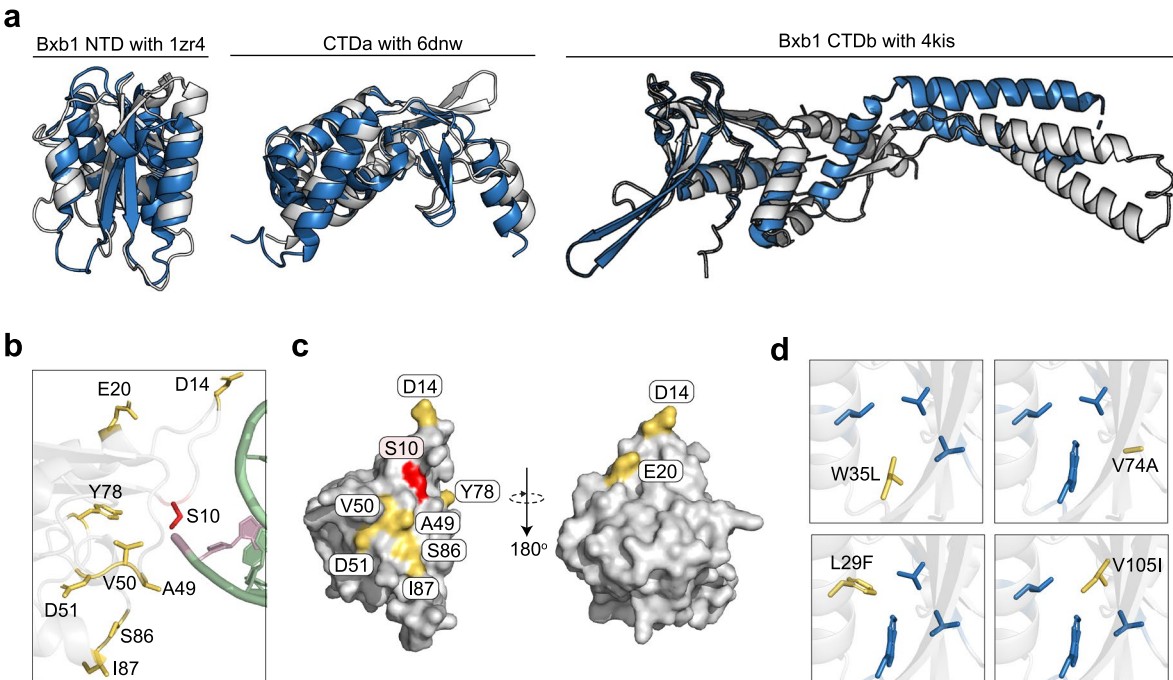

**Extended Data Fig. 3 | Mapping evolved mutations onto the predicted structure of Bxb1. a**, AlphaFold2-predicted structures of the NTD, CTD-a, and CTD- b of Bxb1 (grey). Each domain aligns well with solved structures of serine recombinases (blue) (PDB: 1ZR4, 6DNW, and 4KIS). **b**, Positions of beneficial evolved mutations (yellow) in the AlphaFold2-predicted structure of the NTD of Bxb1. The DNA substrate (green) from gammadelta resolvase tetramer

(PDB: 1ZR4) was docked onto the predicted structure. S10 (red) is the catalytic residue. **c**, Positions of beneficial mutations (yellow) on the surface of the AlphaFold2-predicted structure of the NTD of Bxb1 (grey). **d**, Predicted positions of the four mutated residues (yellow) in the core of the NTD (grey) that resulted in the highest integration efficiencies. Positions were predicted using AlphaFold2. The remaining three unmutated residues in each case are in blue.

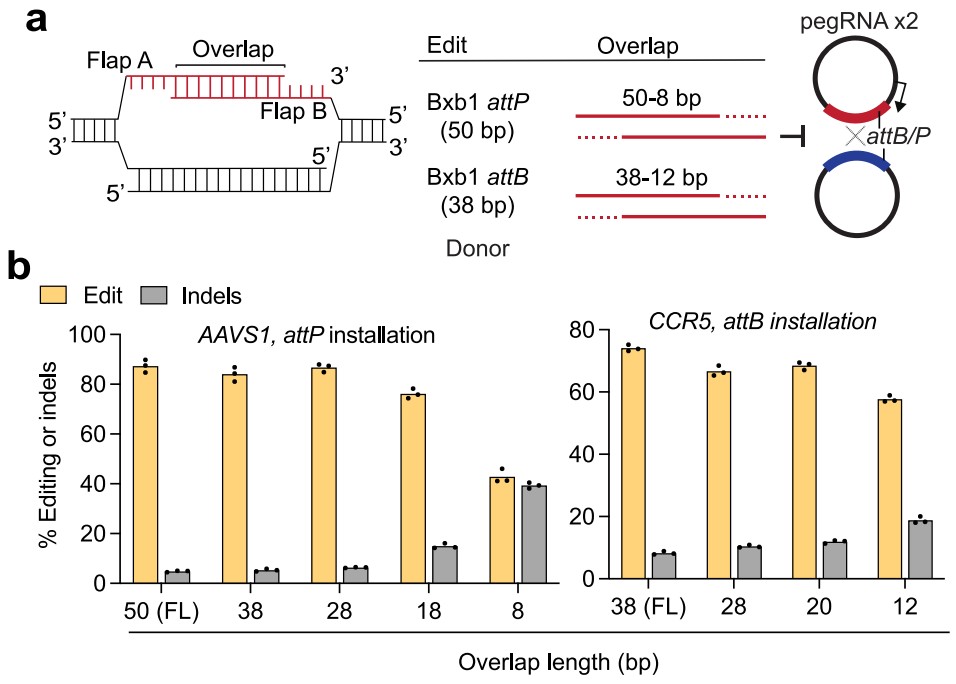

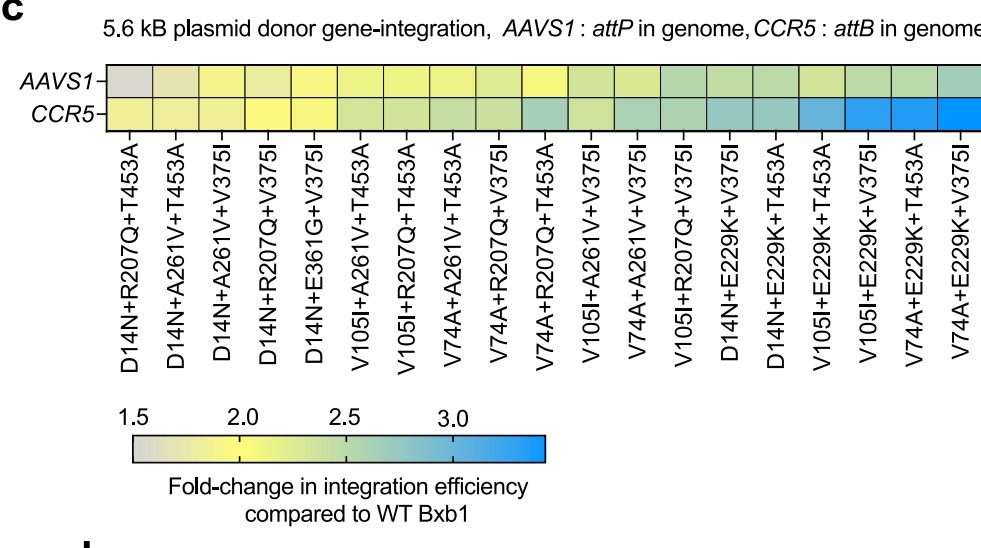

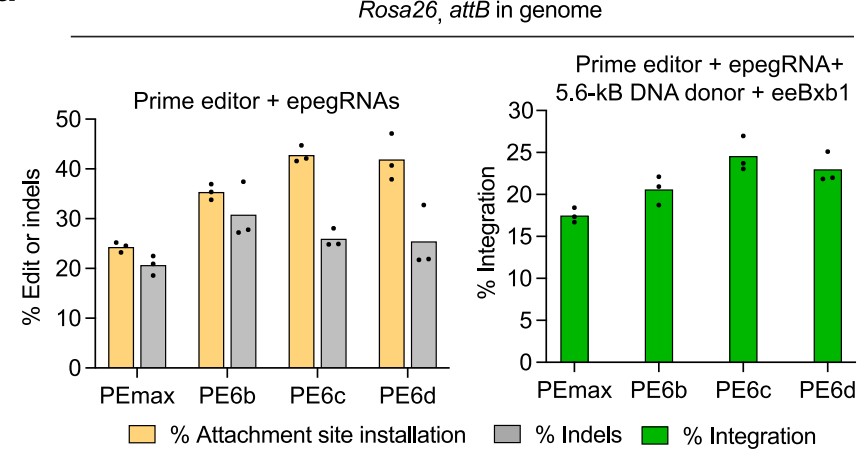

**Extended Data Fig. 4 | See next page for caption.**

**Extended Data Fig. 4 | Further optimization of the PASSIGE system.**
**a**, Schematic of trimmed pegRNA optimization for PASSIGE. When the overlap length between the two newly synthesized 3′ flaps is decreased, plasmid recombination is reduced. **b**, Attachment site installation efficiencies when using trimmed pegRNAs to install either *attP* or *attB* into the *AAVS1* and *CCR5* loci. Overlap lengths from 50 bp to 8 bp and 38 bp to 12 bp were tested to install *attP* and *attB*, respectively. **c**, Heat map of fold-change in integration efficiencies compared to wild-type (WT) Bxb1 for evolved and engineered (ee) Bxb1 variants that were generated by combining one mutation from each domain of Bxb1. A 5.6-kB donor plasmid along with either WT Bxb1, or an ee-variant were transfected into HEK293T cells with either pre-installed *attP* in *AAVS1*

or *attB* in *CCR5*. Each square reflects the mean value for three independent replicates. **d**, Attachment site installation efficiencies (left) and eePASSIGE-mediated integration efficiencies (right) when using PE6 variants to install *attB* into the *Rosa26* site in N2a cells. All components were delivered using a single-transfection. For attachment site installation (left), the prime editor and epegRNAs were delivered. For eePASSIGE-mediated integration (right), eeBxb1 and a 5.6-kB DNA donor were additionally delivered. The % edit or indels was assessed by high-throughput sequencing (**b** and **d**). Bars reflect the mean of three independent replicates and dots show the values of individual replicates (**b**-**d**). Integration efficiencies (**c** and **d**) were assessed by ddPCR analysis as described in Supplementary Note 1.

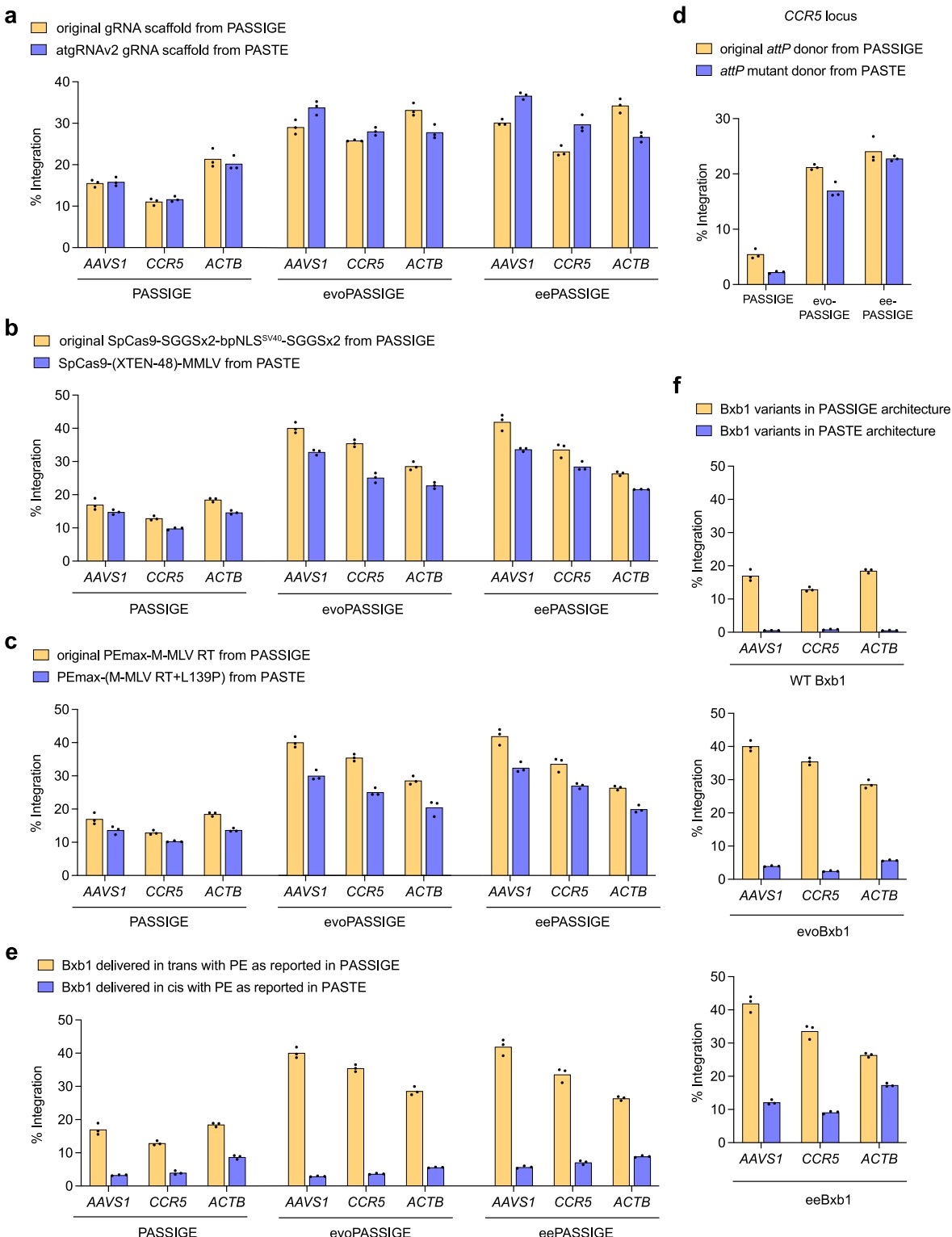

**Extended Data Fig. 5 | Comparison of PASTE with PASSIGE. a**, Comparing the optimized pegRNA scaffold (atgRNAv2) used in PASTE with the original pegRNA scaffold used in PASSIGE. **b**, Comparing the XTEN-48 linker between the Cas9 and M-MLV reverse transcriptase (RT) domain of the prime editor used in PASTE with the SGGSx2-bpNLS$^{SV40}$-SGGSx2 linker used in PASSIGE. **c**, Comparing the mutated M-MLV RT with the L139P mutation used in PASTE with the M-MLV RT used in PASSIGE. **d**, Comparing the mutated *attP* sequence used in PASTE with the original *attP* sequence used in PASSIGE. **e**, Comparing fusion of Bxb1 to the PEmax prime editor using the same linker specified in PASTE (in cis) with the unfused Bxb1 used in PASSIGE (in trans). **f**, Comparing PASSIGE architecture with the PASTE architecture using wild-type Bxb1, evoBxb1, and eeBxb1 recombinases.

In the PASSIGE architecture, Bxb1 variants and the PEmax prime editor are unfused. In the PASTE architecture, Bxb1 variants are fused to the PASTE prime editor using the same linker specified in **e**. For PASSIGE and PASTE experiments, all components were delivered using single transfection, a 5.6-kB donor DNA plasmid was used, and all experiments were performed in HEK293T cells. At the *AAVS1* locus, *attP* was installed and at the *CCR5*, and *ACTB* loci, *attB* was installed. In all cases, the WT Bxb1, evoBxb1, and eeBxb1 were used. Bars reflect the mean of three independent replicates and dots show individual replicate values. Integration efficiency (**a-f**) was assessed by ddPCR analysis as described in Supplementary Note 1.

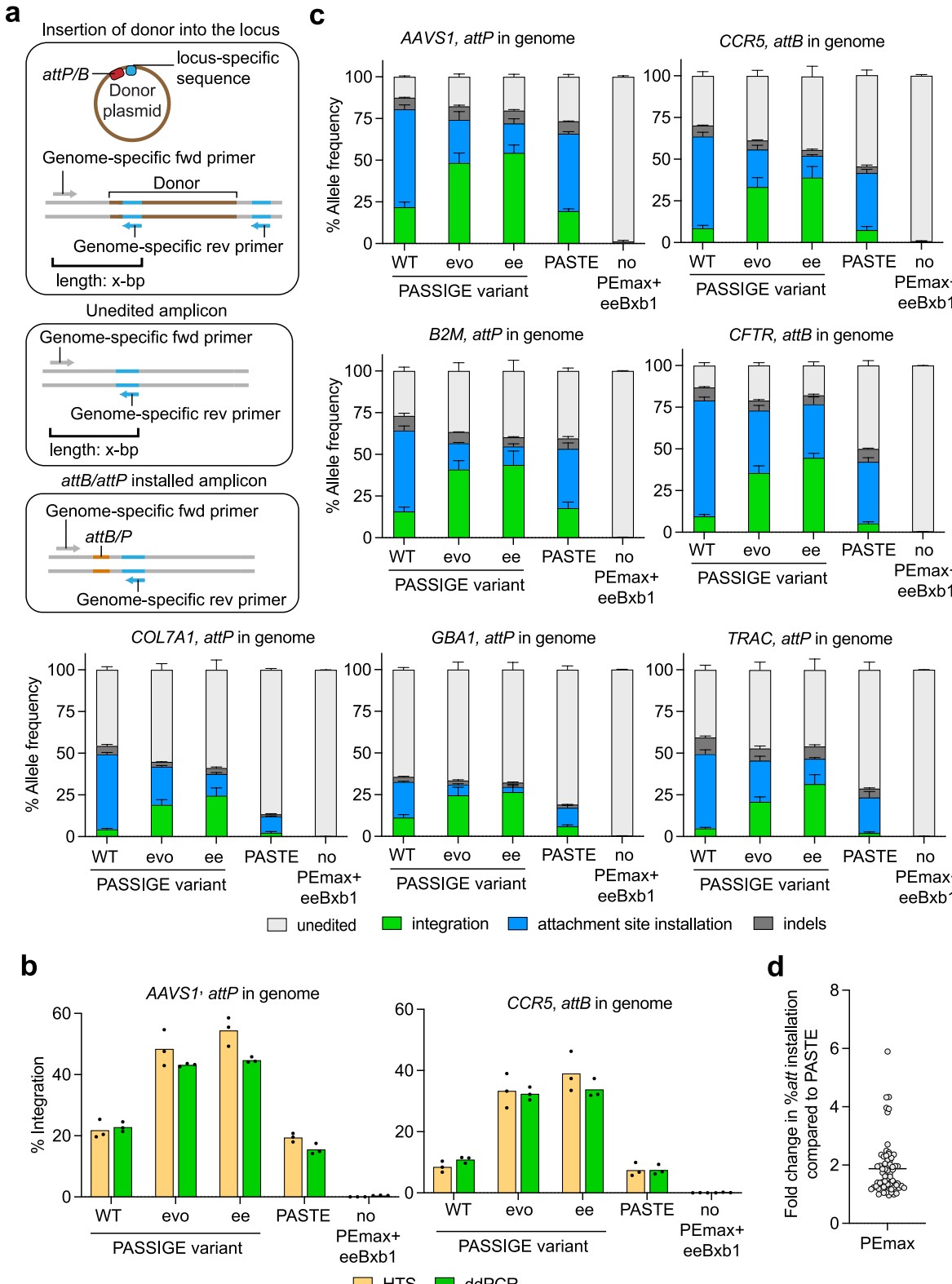

**Extended Data Fig. 6 | See next page for caption.**

**Extended Data Fig. 6 | High-throughput sequencing assay to evaluate allelic distribution after PASSIGE- and PASTE-mediated integration. a**, Schematic of the high-throughput sequencing assay (HTS) used to measure all genomic outcomes post-integration. The 5.6-kB DNA donor plasmid, which is integrated into the target locus encodes a genome-specific reverse primer sequence, enabling a single primer pair with an optimal PCR extension time to amplify unedited, attachment site installed, indel-containing, and donor-integrated amplicons. To reduce potential PCR bias, the integrated amplicon is designed to have the same length as the unedited amplicon (x-bp). **b**, Comparison of integration efficiencies obtained from the HTS assay and ddPCR. ddPCR analysis was performed as described in Supplementary Note 1. **c**, Frequencies of unedited, donor-integrated, attachment-site installed, and indel-containing amplicons

across seven genomic loci measured by HTS. Error bars represent mean ± s.e.m. of n = 3 replicates. **d**, Fold change in attachment (*att*) site installation efficiencies of PEmax relative to PASTE. Efficiencies were assessed across 11 different loci when installing both *attP* and *attB*. HTS was used to assess installation efficiencies, with absolute values and calculations reported in Supplementary Table 5. Horizontal line shows the mean value. For PASSIGE and PASTE experiments (**b** and **c**), all components were delivered using single transfection in HEK293T cells and a 5.6-kB donor DNA plasmid was used. Dual pegRNAs were used to install *attP* or *attB* into each locus as specified in the figure (**b** and **c**). Dots show individual n = 3 replicate values (**b** and **d**). Bars reflect the mean of three independent replicates (**b** and **c**). For the HTS assay, UMI analysis was performed to reduce PCR bias (**b** and **c**).

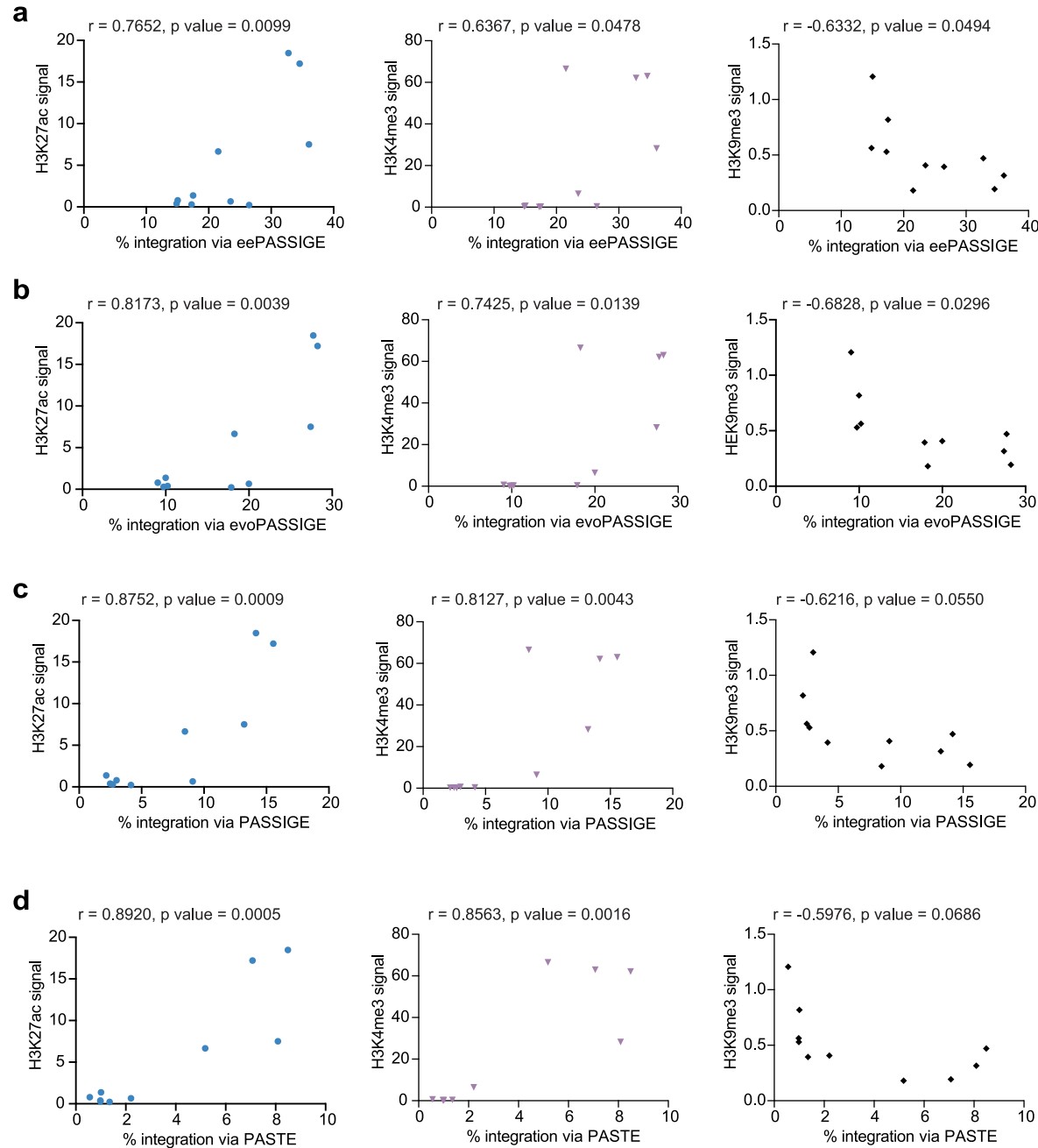

**Extended Data Fig. 7 | Correlation of PASSIGE- and PASTE- mediated integration efficiencies with histone modification markers. a**, Pearson Correlation analysis of eePASSIGE-mediated integration efficiencies across ten genomic loci and histone marker signals extracted and processed from ChIP-seq data deposited in ENCODE. Integration efficiencies were positively correlated with active histone markers, H3K27ac and H3K4me3 and negatively correlated with heterochromatin marker, H3K9me3. Identical analysis was also performed for **b**, evoPASSIGE, **c**, PASSIGE, and **d**, PASTE. Similar to eePASSIGE, the integration efficiencies from evoPASSIGE, PASSIGE and PASTE all positively correlated with HEK27ac and H3K4me3 and negatively correlated with H3K9me3. Pearson correlation coefficient and p-value for each analysis are labelled on the scatter plot. In all cases, p-value was < 0.05 except PASSIGE and PASTE with H3K9me3 marker. Integration efficiencies from three individual replicates, measured by ddPCR, were averaged and used for analysis. The ChIP-seq signal of each histone modification within a 1-kb window centred around the target site was extracted and processed from HEK293T datasets. Statistical significance was calculated using Student's unpaired two-tailed *t*-test.

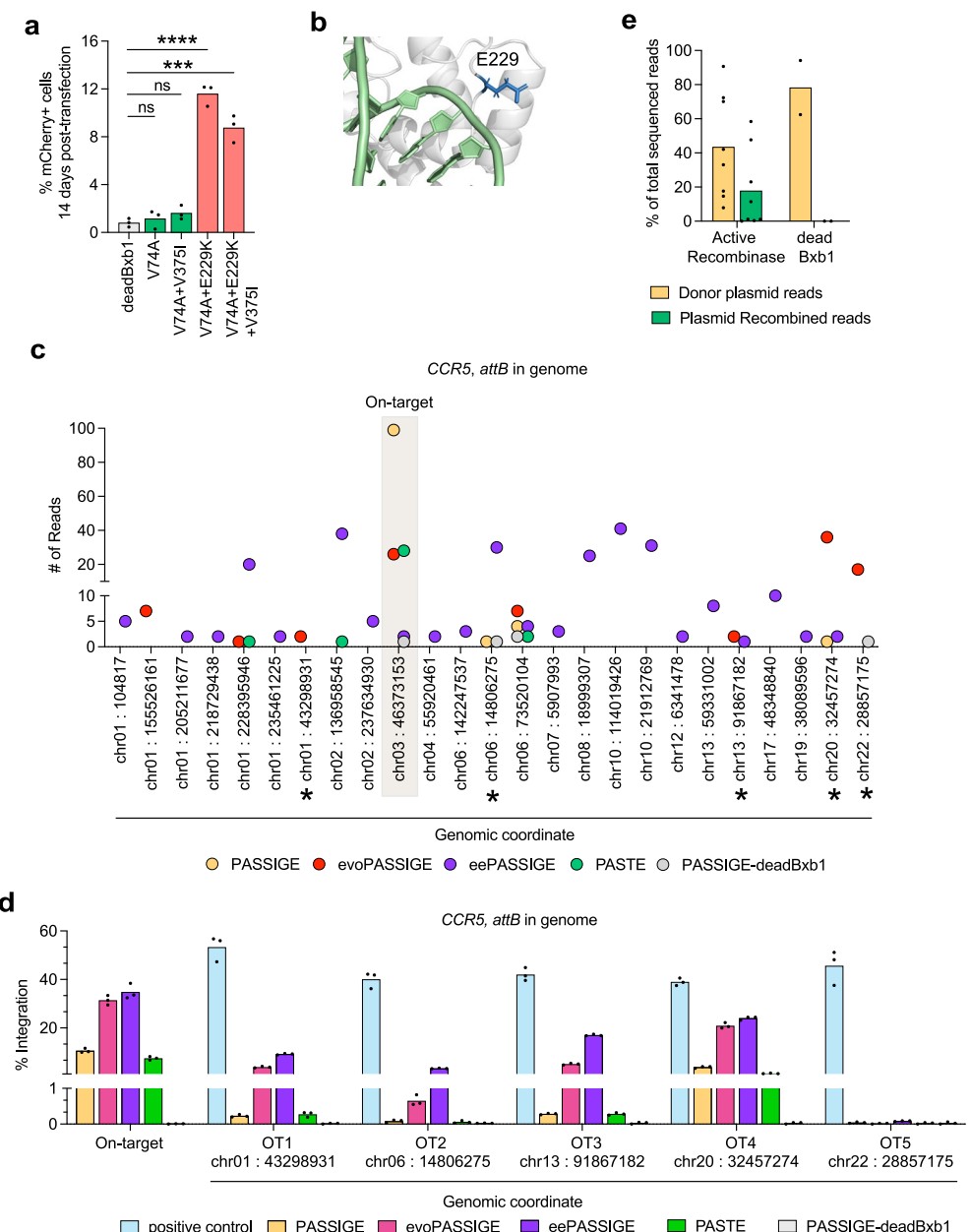

**Extended Data Fig. 8 | Additional off-target profiling of PASSSIGE, evoPASSIGE, eePASSIGE, and PASTE. a**, Percentage of mCherry-positive cells 14 days after transfecting a 3.2-kB donor DNA plasmid along with either dead Bxb1, evoBxb1, Bxb1-(V74A+V375I), Bxb1-(V74A+E229K), or eeBxb1. The donor plasmid has an *attP* site and encodes mCherry under the CMV promoter. Statistical significance was calculated using Student's unpaired two-tailed *t*-test, ***$P < 0.001$, ****$P < 0.0001$. **b**, Predicted position of the E229K (blue) mutation that resulted in off-target integration when delivering an *attP* containing donor. The DNA substrate (green) from *Listeria innocua* prophage serine recombinase (PDB: 6DNW) was docked onto the AlphaFold2-predicted structure of the CTD-a domain of Bxb1 (grey). **c**, Number of uniquely mapped reads and genomic coordinates (Human GRCh38) for UDiTaS nominated off-target sites, with multiple-alignments after deduplication when *attB* is installed into the genome. The on-target *CCR5* locus is shaded. All PASSIGE variants, and PASTE were used to integrate a puromycin-encoding plasmid and cells were passaged for 14 days before analysis. Sites with an asterick were further validated using ddPCR analysis in **d**. All nominated sequences are listed in Supplementary Table 7. **d**, Absolute integration efficiencies at the on-target and five UDiTaS nominated off-target sites when *attB* is installed into the *CCR5*

locus for all PASSIGE variants and PASTE. For the negative control, dead Bxb1 was used instead of the WT recombinase in PASSIGE. For the positive control, a DNA sequence encoding the off-target sequence identified by UDiTaS was mixed with an *ACTB* reference sequence in a 1:1 ratio, so that roughly 50% of the total droplets would give a positive signal. Integration efficiency was assessed by ddPCR analysis as described in Supplementary Note 1. **e**, Percent of reads that either align to the donor plasmid, or the plasmid-donor recombined product in UDiTaS samples. Bars reflect the average of n = 8 samples for active-recombinase treated samples (PASSIGE, evoPASSSIGE, eePASSIGE, and PASTE when an *attB* or *attP* donor is delivered) and n = 2 samples for the dead recombinase treated control. Reads from high-throughput sequencing that included the attachment half-site from the donor plasmid post-demultiplexing were directly used for analysis, as this sequence is expected to be present in all on-target and off-target integration events, pegRNA-donor recombined product, and donor plasmid. For PASSIGE and PASTE experiments, all components were delivered using a single transfection, and a 5.6 kB donor was used (**c**, **d**-**e**). Bars reflect the mean of three independent replicates (**a** and **d**) and dots show the values of individual replicates (**a**,**d**-**e**).

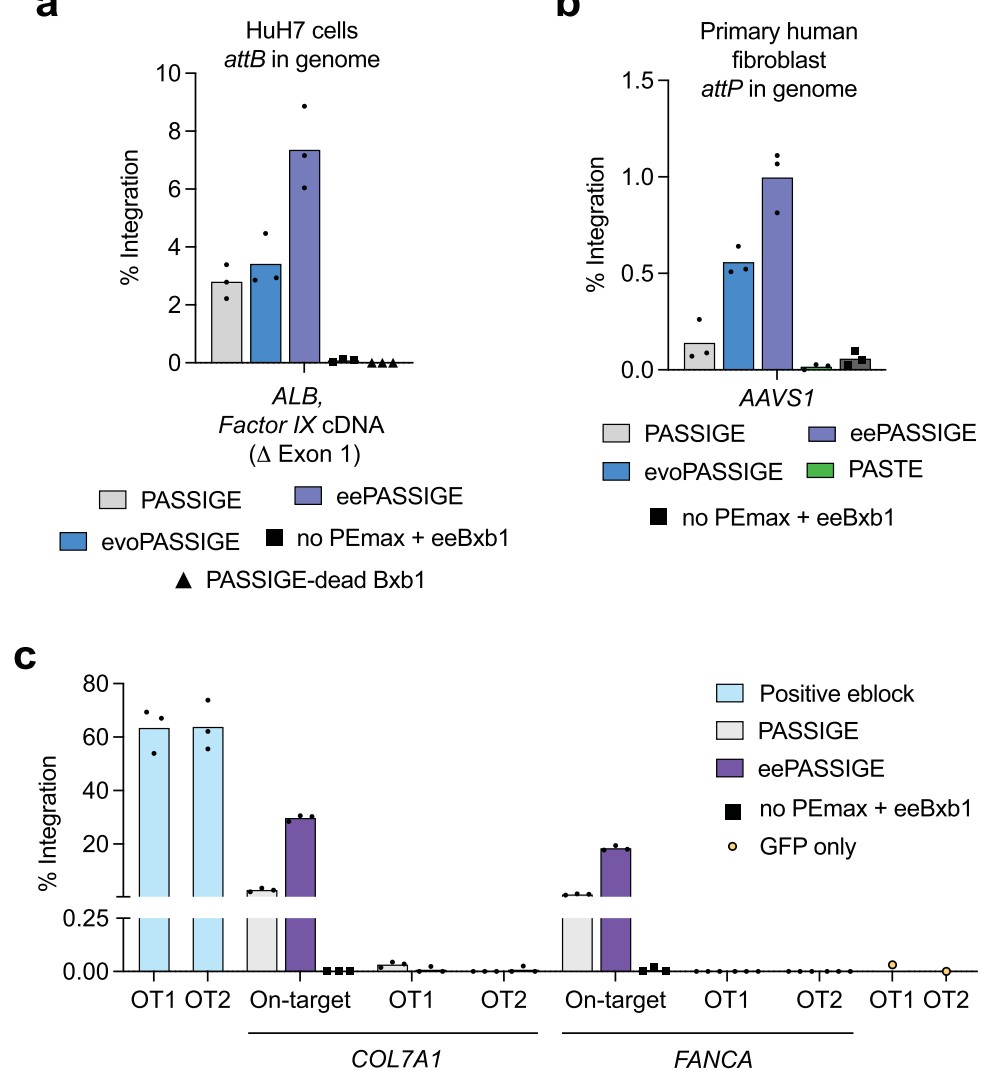

**Extended Data Fig. 9 | Characterization of PASSIGE variants in HuH7 and primary human fibroblast cells. a**, Absolute integration efficiencies of PASSIGE, evoPASSIGE, and eePASSIGE when integrating a *Factor IX* cDNA encoding DNA donor at the *ALB* locus in HuH7 cells. **b**, Absolute integration efficiency of PASSIGE, evoPASSIGE, eePASSIGE, and PASTE in primary human fibroblasts when integrating a 3.0‑kB DNA donor in the *AAVS1* locus. **c**, Absolute integration efficiencies at the on-target and the two UDiTaS nominated off‑target sites when *attP* is installed into intron 4 of *COL7A1* and intron 1 of *FANCA* for PASSIGE and eePASSIGE. For the positive control, a DNA sequence encoding the off-target sequence identified by UDiTaS was mixed with an *ACTB* reference sequence in a 1:1 ratio, so that roughly 50% of the total droplets would give a positive signal. PEmax mRNA, recombinase mRNA, synthetic pegRNAs, and donor-encoding integrase-deficient lentivirus were all delivered via a single electroporation. In (**a** and **b**), all components were delivered as plasmids using either a single transfection (**a**), or electroporation (**b**) and for the negative control, all components except the prime editor protein were delivered into cells with eeBxb1 recombinase. Dual pegRNAs were used to install *attB* into *ALB* (**a**), and *attP* into *AAVS1* (**b**). Bars reflect the mean of three independent replicates and dots indicate individual replicate values (**a-c**). Integration efficiency (**a-c**) was assessed by ddPCR analysis as described in Supplementary Note 1.

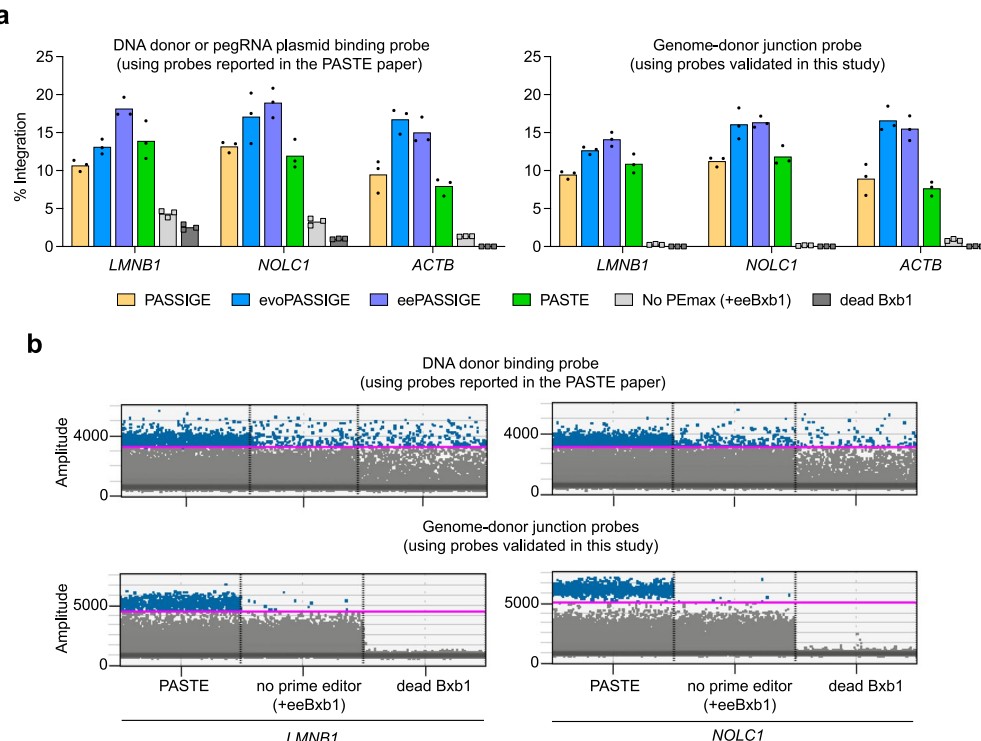

**Extended Data Fig. 10 | Performance of ddPCR probes at the *LMNB1*, *NOLC1*, and *ACTB* sites. a**, Integration efficiencies for PASSIGE, evoPASSIGE, eePASSIGE, and PASTE at the top three most common sites used to characterize PASTE. Initial PASTE publication uses ddPCR probes that bind either to the DNA donor or pegRNA plasmid[26] whereas the initial twin prime editing[24] publication and this work primarily employ probes that bind to the genome–donor junction. Bars reflect the mean of three independent replicates and dots show individual replicate values. **b**, ddPCR plots for PASTE, no PEmax (+eeBxb1) control, and dead Bxb1 control when using different ddPCR probes. The magenta line shows the threshold that was set to assess integration efficiencies in **a**. Details on threshold calculations are provided in Supplementary Note 1. In **a** and **b**, probes used in the original PASTE paper are compared side-by-side with probes used in this study. In the original PASTE report, probes bind to either the DNA donor

plasmid (*LMNB1*, and *NOLC1*) or the Bxb1 *attB* sequence, which is also present in the pegRNA plasmid (*ACTB*). In this study, we primarily used probes that bind to the Bxb1 *attL* or *attR* sites, which are only present in cells after recombination. When using the DNA donor plasmid-binding probe reported in the PASTE paper, high background was observed in the negative controls at *LMNB1*, and *NOLC1*: no PEmax (+eeBxb1) and dead Bxb1 both showed false positive signals. In contrast, minimal or no background was observed at these sites when using an *attL*- binding probe. For PASSIGE and PASTE experiments, all components were delivered using single transfection, and a 4.5- kB donor DNA plasmid reported in the PASTE paper was used. The primer pair used for ddPCR was that reported in the PASTE paper for all reactions. All experiments were performed in HEK293T cells. The genome–donor junction probe was used in all experiments except for a few off-target sites, as detailed in Methods.

# Reporting Summary

## Statistics

For all statistical analyses, confirm that the following items are present in the figure legend, table legend, main text, or Methods section.

| n/a | Confirmed | |
|---|---|---|
| ☐ | ☒ | The exact sample size (*n*) for each experimental group/condition, given as a discrete number and unit of measurement |
| ☐ | ☒ | A statement on whether measurements were taken from distinct samples or whether the same sample was measured repeatedly |
| ☐ | ☒ | The statistical test(s) used AND whether they are one- or two-sided *Only common tests should be described solely by name; describe more complex techniques in the Methods section.* |
| ☐ | ☒ | A description of all covariates tested |
| ☐ | ☒ | A description of any assumptions or corrections, such as tests of normality and adjustment for multiple comparisons |
| ☐ | ☒ | A full description of the statistical parameters including central tendency (e.g. means) or other basic estimates (e.g. regression coefficient) AND variation (e.g. standard deviation) or associated estimates of uncertainty (e.g. confidence intervals) |
| ☐ | ☒ | For null hypothesis testing, the test statistic (e.g. *F*, *t*, *r*) with confidence intervals, effect sizes, degrees of freedom and *P* value noted *Give P values as exact values whenever suitable.* |
| ☒ | ☐ | For Bayesian analysis, information on the choice of priors and Markov chain Monte Carlo settings |
| ☒ | ☐ | For hierarchical and complex designs, identification of the appropriate level for tests and full reporting of outcomes |
| ☐ | ☒ | Estimates of effect sizes (e.g. Cohen's *d*, Pearson's *r*), indicating how they were calculated |

*Our web collection on statistics for biologists contains articles on many of the points above.*

## Software and code

Policy information about availability of computer code

| Data collection | Illumina Miseq Control software (3.1) was used on the Illumina Miseq sequencers to collect the high-throughput sequencing data. Droplet Digital PCR data were collected using the Bio-Rad QX ONE platform. |
|---|---|
| Data analysis | CRISPResso2 was used to analyse Miseq data for quantifying %edit and %indels at genomic loci. Droplet digital PCR data were analysed using QX ONE software 1.3 Standard Edition. AmpUMI was used to deduplicate samples when noted. Code used for processing and analysing high-throughput sequencing data are available at https://github.com/pinellolab/CRISPResso2. Code used to deduplicate sequencing reads before CRISPResso2 analyis is available at http://github.com/pinellolab/AmpUMI. The Mutato code used to generate mutation tables from sequencing reads of evolved phage is available at https://hub.docker.com/r/araguram/mutato. Code used to analyse UdiTaS is available at https://github.com/Nicholas-Krasnow/integrato. |

For manuscripts utilizing custom algorithms or software that are central to the research but not yet described in published literature, software must be made available to editors and reviewers. We strongly encourage code deposition in a community repository (e.g. GitHub). See the Nature Portfolio guidelines for submitting code & software for further information.

## Data

Policy information about availability of data

All manuscripts must include a data availability statement. This statement should provide the following information, where applicable:

- Accession codes, unique identifiers, or web links for publicly available datasets
- A description of any restrictions on data availability
- For clinical datasets or third party data, please ensure that the statement adheres to our policy

The main data supporting the results in this study are available within the paper and its Supplementary Information. High-throughput DNA sequencing FASTQ files are available from the National Center of Biotechnology's Information Sequence Read Archive under BioProject (PRJNA1042817). Droplet digital polymerase chain reaction (ddPCR) and flow-cytometry data are available from Figshare via the identifier 10.6084/m9.figshare.24589083. Plasmids encoding evoBxb1 and eeBxb1 and a subset of pegRNA encoding plasmids are available through Addgene. Other data are available from the corresponding author on reasonable request.

## Research involving human participants, their data, or biological material

Policy information about studies with human participants or human data. See also policy information about sex, gender (identity/presentation), and sexual orientation and race, ethnicity and racism.

| | |
|---|---|
| Reporting on sex and gender | The study did not involve human research participants. |
| Reporting on race, ethnicity, or other socially relevant groupings | – |
| Population characteristics | – |
| Recruitment | – |
| Ethics oversight | – |

Note that full information on the approval of the study protocol must also be provided in the manuscript.

# Field-specific reporting

Please select the one below that is the best fit for your research. If you are not sure, read the appropriate sections before making your selection.

☒ Life sciences    ☐ Behavioural & social sciences    ☐ Ecological, evolutionary & environmental sciences

For a reference copy of the document with all sections, see nature.com/documents/nr-reporting-summary-flat.pdf

# Life sciences study design

All studies must disclose on these points even when the disclosure is negative.

| | |
|---|---|
| Sample size | Sample sizes were determined on the basis of precedents in genome-editing experiments (such as Anzalone et al., Nature 2019). |
| Data exclusions | No data were excluded. |
| Replication | The experiments described in the paper used three replicates (except for Fig. 6b, which used two replicates), and all attempts at replication were successful. |
| Randomization | The mammalian cells used in this study were grown under identical conditions; no randomization was used. |
| Blinding | The mammalian cells used in this study were grown under identical conditions; blinding was not used. |

# Reporting for specific materials, systems and methods

We require information from authors about some types of materials, experimental systems and methods used in many studies. Here, indicate whether each material, system or method listed is relevant to your study. If you are not sure if a list item applies to your research, read the appropriate section before selecting a response.

## Materials & experimental systems

| n/a | Involved in the study |
|---|---|
| ☒ | ☐ Antibodies |
| ☐ | ☒ Eukaryotic cell lines |
| ☒ | ☐ Palaeontology and archaeology |
| ☒ | ☐ Animals and other organisms |
| ☒ | ☐ Clinical data |
| ☒ | ☐ Dual use research of concern |
| ☒ | ☐ Plants |

## Methods

| n/a | Involved in the study |
|---|---|
| ☒ | ☐ ChIP-seq |
| ☐ | ☒ Flow cytometry |
| ☒ | ☐ MRI-based neuroimaging |

## Eukaryotic cell lines

Policy information about cell lines and Sex and Gender in Research

| | |
|---|---|
| Cell line source(s) | HEK293T (ATCC), N2A (ATCC), HuH7 (a gift from Erik Sontheimer Lab at UMMS via ATCC), Primary human fibroblasts (obtained and expanded following the declaration of Helsinki Principles at University of Minnesota Medical School), human induced pluripotent stem cells (WiCell). |
| Authentication | HEK293T and N2A were authenticated by the supplier using STR analysis. HuH7 cells were authenticated using ATCC's cell authentication. Primary human fibroblasts and human induced pluripotent stem cells were authenticated by the suppliers. |
| Mycoplasma contamination | All cell lines tested negative for mycoplasma. |
| Commonly misidentified lines (See ICLAC register) | No commonly misidentified cell lines were used. |

## Flow Cytometry

### Plots

Confirm that:

☒ The axis labels state the marker and fluorochrome used (e.g. CD4-FITC).

☒ The axis scales are clearly visible. Include numbers along axes only for bottom left plot of group (a 'group' is an analysis of identical markers).

☒ All plots are contour plots with outliers or pseudocolor plots.

☒ A numerical value for number of cells or percentage (with statistics) is provided.

### Methodology

| | |
|---|---|
| Sample preparation | 15,000 HEK293T cells were seeded into 96-well poly-D-lysine coated plates (Corning). 16-24 h post-seeding, cells were transfected with 100 ng of Bxb1 variant and 150 ng of an mCherry encoding donor plasmid using 0.5uL of Lipofectamine 2000 (Thermo Fisher Scientific). Cells were passaged for 2 weeks to dilute the plasmid DNA. Next, transfected cells were trypsinized, resuspended in PBS solution, and assessed for mCherry fluorescence. |
| Instrument | CytoFLEX S Flow Cytometer (Beckman Coulter). |
| Software | CytoFLEX S Flow Cytometer (Beckman Coulter) software. |
| Cell population abundance | The abundances of surviving single cells that were mCherry-positive were dependent on the Bxb1 variant that was transfected. These varied in the range of 0.28–12.15%. Abundances are noted in Fig. 5a and Extended data Fig. 8a. |
| Gating strategy | Cells were first gated (Gate A) based on forward (FSC-A) and side scattering (SSC-A) to remove dead cells and other debris. A second gate (Gate B) was used to select singlets based on FSC-H and FSC-A. Finally, mCherry-positive and mCherry-negative cells were gated (Gate C) and analysed via the ECD channel. Supplementary Note 2 highlights an example of how the cells were gated. |

☒ Tick this box to confirm that a figure exemplifying the gating strategy is provided in the Supplementary Information.

