## [Peer Review File · Nature Biomedical Engineering]

Efficient site-specific integration of large genes in mammalian cells via continuously evolved recombinases and prime editing

Corresponding author: David Liu

Editorial note

This document includes relevant written communications between the manuscript's corresponding author and the editor and reviewers of the manuscript during peer review. It includes decision letters relaying any editorial points and peer-review reports, and the authors' replies to these (under 'Rebuttal' headings). The editorial decisions are signed by the manuscript's handling editor, yet the editorial team and ultimately the journal's Chief Editor share responsibility for all decisions.

Any relevant documents attached to the decision letters are referred to as **Appendix #**, and can be found appended to this document. Any information deemed confidential has been redacted or removed. Earlier versions of the manuscript are not published, yet the originally submitted version may be available as a preprint. Because of editorial edits and changes during peer review, the published title of the paper and the title mentioned in below correspondence may differ.

Correspondence

Wed 10 Jan 2024

Decision on Article nBME-23-3328

Dear Prof Liu,

Thank you again for submitting to *Nature Biomedical Engineering* your manuscript, "Efficient and versatile programmable large-gene integration in mammalian cells enabled by continuously evolved recombinases and prime editing". The manuscript has been seen by four experts, whose reports you will find at the end of this message. You will see that the reviewers appreciate the work, and that they raise a number of technical criticisms that I am hoping you will be able to address. In particular, we would expect that a revised version of the manuscript provides:

- * Strengthened off-target-integration analyses, as per the questions and suggestions from Reviewers #1, #2 and #4.
- * Enhanced analysis of the outcomes of the evolution systems, to support your claim of mutational convergence, as per the relevant questions from Reviewers #2 and #4.
- * As per the requests from Reviewers #1 and #2, quantification of the method's gene-integration efficiency in primary cells (as also shown in these two related methods that this journal published).

When you are ready to resubmit your manuscript, please upload the revised files, a point-by-point rebuttal to the comments from all reviewers, the reporting summary, and a cover letter that explains the main improvements included in the revision and responds to any points highlighted in this decision.

Please follow the following recommendations:

- * Clearly highlight any amendments to the text and figures to help the reviewers and editors find andunderstand the changes (yet keep in mind that excessive marking can hinder readability).

* If you and your co-authors disagree with a criticism, provide the arguments to the reviewer (optionally, indicate the relevant points in the cover letter).

* If a criticism or suggestion is not addressed, please indicate so in the rebuttal to the reviewer comments and explain the reason(s).

* Consider including responses to any criticisms raised by more than one reviewer at the beginning of the rebuttal, in a section addressed to all reviewers.

* The rebuttal should include the reviewer comments in point-by-point format (please note that we provide all reviewers will the reports as they appear at the end of this message).

* Provide the rebuttal to the reviewer comments and the cover letter as separate files.

We hope that you will be able to resubmit the manuscript within 12 weeks from the receipt of this message. If this is the case, you will be protected against potential scooping. Otherwise, we will be happy to consider a revised manuscript as long as the significance of the work is not compromised by work published elsewhere or accepted for publication at *Nature Biomedical Engineering*.

We hope that you will find the referee reports helpful when revising the work, which we look forward to receive. Please do not hesitate to contact me should you have any questions.

Best wishes,

Pep

Pep Pàmies
Chief Editor, Nature Biomedical Engineering

Reviewer #1 (Report for the authors (Required)):

Pandey et al. used directed molecular evolution to generate improved versions of Bxb1 recombinase, a unidirectional large-serine recombinase that can deliver DNA pay-loads to the genome of mammalian cells. The authors subsequently assessed these refined recombinase variants in the context of prime editing-assisted site-specific integrase gene editing (PASSIGE), a technology previously established by the group. They report a 3-4-fold increase in integration of donor plasmids over wild-type Bxb1 and claim single-transfection targeting efficiencies of up to 46%. Finally, the authors compare their improved PASSIGE method to PASTE, a related method for integrating DNA payloads into mammalian genomes. The manuscript provides a well-structured presentation of their methodology and results, highlighting significant advancements over prior work. However, to enhance the manuscript's value before publication, certain aspects merit further consideration.

Major points:

1. The manuscript's primary weakness lies in the absence of PASSIGE validation within a more pertinent cell type. All experiments are conducted in established cancer cell lines with complex genotypes (e.g. HEK293 cells are hypotriploid). This makes it difficult to correctly calculate targeting efficiencies (see also point 2), but more importantly, it limits the proven utility of the approach. Targeting therapeutically relevant sites is most effectively achieved within cells that bear clinical relevance. Therefore, the manuscript would significantly benefit from conducting experiments in primary human cells (such as primary T cells or human iPSCs).

2. The ddPCR assay that is employed to measure integration efficiencies is well described and convenient to compare different variants to the WT enzyme. However, the assay might be misleading for the absolute

percentages of integration achieved when used in cell lines with complex genotypes. These cell lines frequently carry multiple copies of genomic regions due to gene amplifications or chromosome duplications/deletions. Hence, the reference probe would need to be selected carefully to adjust for these complications. Upon reviewing the numbers presented in the manuscript, it becomes challenging to reconcile the reported 46% integration efficiency from a single transfection experiment (co-transfection efficiency of 4 plasmids + prime editing efficiency + Bxb1 integration efficiency + indel frequency). Again, experiments in primary cells might be helpful to support the presented data.

3. To validate the ddPCR results it would be useful to employ an additional, independent assay (e.g. FACS). This should be relatively easy to achieve by adapting the donor plasmid. Employing a donor that carries a promoterless fluorescent marker gene could for instance be targeted to the ALB promoter described in the manuscript. A simple FACS assay would then reveal the successful integration events.

4. The authors optimized the PASSIGE system also to reduce indel formation. Nevertheless, even for the best combinations tested, indels were still observed. It would be useful to describe the observed indel sequences in more detail (additional Suppl. Table?) to understand better where they come from and whether they could cause unintended outcomes. This might also help to explain why much higher indel formation was observed for the attP installation at the CCR5 locus and for the Rosa26 and Smn1 loci for both, attP and attB installation. It would also be useful to understand how the observed indels influence overall PASSIGE efficiencies.

5. The authors report off-target integration when eeBxb1 was used in combination with an attP-donor. It would be useful to determine the integration site(s) (e.g. by Splinkerette PCR).

6. The rationale behind the authors' choice to employ distinct PE systems (PEmax or PE6) for integrating therapeutic DNA cargo via PASSIGE variants remains unclear. Is there a specific justification for utilizing different PE systems? Additionally, could employing the alternative PE system potentially yield different results, prompting a reevaluation of the findings?

7. For the integration of the F9 minicircle DNA donor into the Albumin locus in HuH7 cells only the improved expression levels are reported (unless these correspond to the results presented in Fig. 5a). It would be useful to also mention the integration efficiencies.

Minor points

1. FANCA targeting and indel results should be added to Extended Data Fig. 8.
2. In Figure 2d the mutations A449V and T435A should be swapped to maintain the order of mutations shown (ascending order).

Reviewer #2 (Report for the authors (Required)):

The manuscript by Liu and colleagues describes the utilization of a phage evolution system to improve the activity of Bxb1 recombinase for integration of a DNA cargo site specifically mediated by prime editing (PASSIGE). The authors use two different phage evolution approaches (PACE & PANCE) to isolate Bxb1 mutants that have improved recombinase activity in bacteria. Next, they screen a number of these isolated Bxb1 variants individually and in combination to define evoBxb1 and eeBxb1 variants that display improved recombination activity in mammalian cells (primarily HEK293T cells or N2A cells). These yield improved PASSIGE systems for site-specific DNA integration (typically 2 to 3 fold higher rates than the original system). They also assess the off-target activity of evoBxb1 and eeBxb1 with DNA donors containing each recombination site (attP and attB) and observe off-target activity only with the attP donor in the context of eeBxb1. Finally, they compare their improved PASSIGE systems (evoPASSIGE or eePASSIGE) with the previously described PASTE system that uses a similar approach with distinct components for site-specific DNA integration. The PASSIGE system outperforms PASTE in multiple contexts.

Overall this is an interesting manuscript that describes important improvements to the PASSIGE system for the site-specific integration of donor DNAs within the genome. The experiments are clearly presented within the manuscript, and the authors provide many extended data figures that provide important depth to their analysis of the Bxb1 variants and the PASSIGE system. The detailed comparison between the PASSIGE

and PASTE systems will be valuable to readers that are interested in choosing an approach for integrating large cargos within the genome.

The primary concern of this reviewer is the incremental nature of the content, as this study builds on the primary description of the PASSIGE system (Anzalone and Gao NBT 2022). The primary achievement is improved activity, and there is assessment of some additional therapeutically relevant targets, but the manuscript limits the analysis of PASSIGE to transformed cell lines. For example, the integration of Factor IX in the ALB locus was performed in the prior study.

Minor concerns/suggestions:

1) While the PANCE and PACE evolution systems identified Bxb1 mutants with improved activity, the selections did not appear to converge to specific mutants based on the mutations recovered from the clones in the supplementary tables (despite the authors claims of a “high degree of mutational convergence”). The recovered clones appear to have characteristics of both selection and jackpot effects (Table S4 – V74A is only found in a single clone in one lagoon and V105I is found in two related clones in a single lagoon). Do the authors have evidence of “a high degree of mutational convergence”? The modest depth that the selected pools were analyzed (four individual recovered clones were characterized from each lagoon) makes an assessment of convergence challenging. In addition, negative selective pressure (absence of mutations at amino acid positions that will negatively impact function) will cause the appearance of an enrichment of mutations at positions that are tolerant of change, such as linkers where many mutations were recovered.

2) For the off-target analysis, the authors propose that the integration is occurring at attB-resembling sequences within the genome by eeBxb1. Deep sequencing analysis of the integration sites (using a method similar to used for genome-wide mapping of lentiviral integration sites) should provide concrete evidence of the sequences where integration is occurring.

3) Epigenomic differences between different genomic loci can impact the efficiency of genome editing in some systems. Is there any evidence that transcriptionally active genomic loci are more amenable to modification using the PASSIGE system? In their prior 2022 study ALB was easier to target with PASSIGE in Huh7 cells than HEK293T cells.

4) In Figure 4 panel A – comparison of the activity of the different systems (PASSIGE, evoPASSIGE & eePASSIGE) at common target sites (e.g. ACTB) is made more challenging because neighboring graphs have different Y-axis scales. Showing the mean value above each bar would assist the reader in discerning that the improved PASSIGE systems have higher activity in general. Just going by bar height all three systems appear to have similar activity at ACTB.

Question: Since the authors are working with a system where the active species is a dimer of dimers and the N-terminal catalytic domain is at the interface of the four molecules in the active complex, have they considered the possibility that some of their mutants described as being in the “core” of the N-terminal domain (W35, L29, V74 & V105) might be involved in dimerization or tetramerization, and that they are selecting residues that are optimal in the context of a heterodimer between the WT sequence and the recovered mutant sequence? It is hard to tell from the structural models that are presented within the manuscript how buried these “core” residues are. There are multiple ways that more than one Bxb1 protein species being expressed in a bacterial cell could occur in the context of their selection system. Bxb1 heterodimers might have higher recombinase activity than the homodimers.

Reviewer #3 (Report for the authors (Required)):

The authors have substantially improved on the efficiency of programmable sequence insertion by using PANCE/PACE evolution techniques to greatly enhance Bxb1-mediated integration. By combining the newly derived versions of Bxb1 (evoBxb1 and eeBxb1) with sixth-generation prime editors (previously developed by the same lab), they are able to achieve insertion efficiency, in cells, manyfold higher than previous techniques (PASTE and PASSIGE). The authors characterize their new tools broadly in therapeutically relevant genes and show that the gains in efficiency are highly generalizable across loci.

From a technical perspective, the work is flawless. In reading through the manuscript, I came up with not a

single suggestion for improvement. This is a rare manuscript that I think could reasonably be published without any modification at all.

This is unquestionably an important advance for the genome editing field. Previous attempts at programmable sequence insertion (even PASSIGE and PASTE) have been rather disappointing. Yet programmable sequence insertion is widely regarded as the next frontier in therapeutic genome editing, since it uniquely would provide a mutation-agnostic way to treat genetic disorders. Tome Biosciences has been in the news a lot recently because of the potential perceived in their therapeutic efforts based on PASTE. It is clear that the new Bxb1 platform presented in this manuscript is far superior.

Reviewer #4 (Report for the authors (Required)):

In this study, the Liu lab presents another directed evolution study in which they utilize their common PACE system to evolve the Bxb1 large serine recombinase. They develop a clever selection system in which different selection strengths enable their use of an optimal selection. They result in two Bxb1 variants, namely evoBxb1 and eeBxb1, which exemplify enhanced recombination activity either alone with pre-installed attP/attB sites or with their PASSIGE system. Indeed the evolved variants seem to exhibit increased editing, which reflects their successful engineering effort, I have some questions regarding the evolution process and off-target analysis. I believe that the authors should address these concerns prior to publication.

1. I was curious about the evolution and looked carefully at the mutant table. Can the authors explain why they did not find any obvious trends in particular mutations being amplified throughout the evolution. Rather, it seems that the mutant table is very sporadic and that the mutants from one PANCE do not enrich in subsequent rounds (even some seem to disappear). Other studies by the Liu lab show much better enrichment so I am curious why there were no trends in this recombinase evolution? Furthermore, why did library members enrich (better phage propagation) but did not show better activity that enrich over time?
2. The authors show that the V105I mutation have enhanced activity alone when used with clonal cells. However, this did not show the best activity with the PASSIGE system. What do the authors believe is the reason for this discrepancy if the prime editing components are shared between both?
3. I hope the authors can directly compare the raw editing activity either in clonal cells or with the PASSIGE system between the single best mutants (V105I or V74A) and the best triple mutant (eeBxb1). The heat map does not exactly seem to enhance overall editing activity between the single and triple mutants.
4. The authors give a good hypothesis on how the mutation in the NTD increases activity. Could the authors do the same for the other two mutations in the CTD in their eeBxb1 mutant?
5. The authors give a very clever off-target analysis as analyzed by flow cytometry, which seems to also be used previously by other studies. I hope the authors can take their mCherry positive cells after flow and sequence them to indeed verify that the mCherry expression is due to integration and not plasmid leakage. This would be more powerful in supporting their claims. Furthermore, I wonder if the authors can take one of their samples with the attP donor and eeBxb1 and sequence to identify where the off target integration is located. This would also support their claim on sites being more similar located in the human genome.

Thu 02 May 2024

Decision on Article nBME-23-3328A

Dear Prof Liu,

Thank you for your revised manuscript, "Efficient and versatile programmable large-gene integration in mammalian cells enabled by continuously evolved recombinases and prime editing". Having consulted with the original reviewers (whose comments you will find at the end of this message), I am pleased to write that we shall be happy to publish the manuscript in *Nature Biomedical Engineering*, provided that the points specified in the attached instructions file are addressed.

When you are ready to submit the final version of your manuscript, please upload the files specified in the instructions file.

Also, please consider discussing in the manuscript the additional point made by Reviewer #4.

We encourage authors to take up transparent peer review. If you are eligible and opt in to transparent peer review, we will publish, as a single supplementary file, all the reviewer comments for all the versions of the manuscript, your rebuttal letters, and the editorial decision letters. **If you opt in to transparent peer review, in the attached file please tick the box 'I wish to participate in transparent peer review'; if you prefer not to, please tick 'I do NOT wish to participate in transparent peer review'**. In the interest of confidentiality, we allow redactions to the rebuttal letters and to the reviewer comments. If you are concerned about the release of confidential data, please indicate what specific information you would like to have removed; we cannot incorporate redactions for any other reasons. More information on transparent peer review is available.

Best wishes,

Pep

Pep Pàmies
Chief Editor, Nature Biomedical Engineering

P.S. Nature Portfolio journals encourage authors to share their step-by-step experimental protocols on a protocol-sharing platform of their choice. Nature Portfolio's Protocol Exchange is a free-to-use and open resource for protocols; protocols deposited in Protocol Exchange are citable and can be linked from the published article. More details can be found at www.nature.com/protocolexchange/about.

Reviewer #1 (Report for the authors (Required)):

The authors have largely addressed my comments/concerns and I would support publication of this manuscript in *Nature Biomedical Engineering*.

Reviewer #2 (Report for the authors (Required)):

The authors have addressed my primary concern with the original manuscript through the incorporation of data demonstrating activity of the PASSIGE system in primary cells. They have also incorporated additional off-target analysis (UDiTAS) that provides an alternate unbiased approach for the discovery of potential integration sites throughout the genome. This improved system will be of broad interest to the scientific

community for targeted integration of large DNA cargos.

Reviewer #4 (Report for the authors (Required)):

The authors addressed most of my concerns and I commend the authors on their thorough evaluation of the Bxb1 recombinase. Based on the newly included statements, it seems that the E229K is a critical mutation that greatly enhances editing activity. I wonder if the authors performed any experiments amongst mutants containing E229K and double or triple variants containing E229K? This would be interesting and a more thorough evaluation with possible mutations that also enhance editing as opposed to the originally identified starting mutations from the early evolution studies. Otherwise, the authors addressed all my previous concerns.

Nature Biomedical Engineering is a Transformative Journal. Authors may publish their research with us through the traditional subscription access route, or make their paper immediately open access through payment of an article-processing charge. More information about publication options is available.

You may need to take specific actions to comply with funder and institutional open-access mandates. If the work described in the accepted manuscript is supported by a funder that requires immediate open access (as outlined, for example, by Plan S) and your manuscript was originally submitted on or after January 1st 2021, then you will need to select the gold OA route. Authors selecting subscription publication will need to accept our standard licensing terms (including our self-archiving policies), and these will supersede any other terms that the author or any third party may assert apply to any version of the manuscript.

Rebuttal 1

Below is a point-by-point response (in blue) to each reviewer comment:

Reviewer #1:

Pandey et al. used directed molecular evolution to generate improved versions of Bxb1 recombinase, a unidirectional large-serine recombinase that can deliver DNA pay-loads to the genome of mammalian cells. The authors subsequently assessed these refined recombinase variants in the context of prime editing-assisted site-specific integrase gene editing (PASSIGE), a technology previously established by the group. They report a 3-4-fold increase in integration of donor plasmids over wild-type Bxb1 and claim single-transfection targeting efficiencies of up to 46%. Finally, the authors compare their improved PASSIGE method to PASTE, a related method for integrating DNA payloads into mammalian genomes. The manuscript provides a well-structured presentation of their methodology and results, highlighting significant advancements over prior work. However, to enhance the manuscript's value before publication, certain aspects merit further consideration.

Major points:

1. The manuscript's primary weakness lies in the absence of PASSIGE validation within a more pertinent cell type. All experiments are conducted in established cancer cell lines with complex genotypes (e.g. HEK293 cells are hypotriploid). This makes it difficult to correctly calculate targeting efficiencies (see also point 2), but more importantly, it limits the proven utility of the approach. Targeting therapeutically relevant sites is most effectively achieved within cells that bear clinical relevance. Therefore, the manuscript would significantly benefit from conducting experiments in primary human cells (such as primary T cells or human iPSCs).

We thank the reviewer for this feedback. We have now added new experiments in cells beyond transformed cell lines and reported targeted integration in therapeutically relevant cell types including iPSCs and primary human fibroblasts (Fig. 6c-d). By using a non-plasmid delivery modality, we demonstrate that eePASSIGE achieves up to 30% targeted integration of a 5.8-kB DNA cargo at therapeutically relevant sites in primary human cells following a single-electroporation (shown on the right), and exhibits a substantial 14-fold improvement over PASSIGE in primary human fibroblasts. We had to extensively optimize the delivery modality to achieve high levels of integration in these cell types, as further emphasized in the manuscript, and switching to a non-plasmid system was essential to avoid DNA-induced cytotoxicity. Additionally, in clonal human induced pluripotent stem cells with pre-installed attachment sites, we demonstrate that eeBxb1 improves integration of a 5.6-kB plasmid by 7.3-fold compared to the wild-type recombinase (Fig. 6c).

2. The ddPCR assay that is employed to measure integration efficiencies is well described and convenient to compare different variants to the WT enzyme. However, the assay might be misleading for the absolute percentages of integration achieved when used in cell lines with complex genotypes. These cell lines frequently carry multiple copies of genomic regions due to gene amplifications or chromosome duplications/deletions. Hence, the reference probe would need to be selected carefully to adjust for these complications. Upon reviewing the numbers presented in the manuscript, it becomes challenging to reconcile the reported 46% integration efficiency from a single transfection experiment

(co-transfection efficiency of 4 plasmids + prime editing efficiency + Bxb1 integration efficiency + indel frequency). Again, experiments in primary cells might be helpful to support the presented data.

The data presented in primary cells, as discussed in the previous comment, partially addresses this point. In addition, to further validate the ddPCR analysis, we performed a new UMI-de-duplicated high-throughput sequencing (HTS) assay to assess integration efficiencies in HEK293T cells at several loci. This assay sequences all alleles present at a given locus and thus avoids potential confounding factors that may arise from copy number variation. HTS of target loci confirmed the >40% integration efficiencies achieved by PASSIGE variants from a single transfection experiment (Extended Data Fig. 6b-c). An example of how the UMI HTS quantitation compares with the ddPCR quantitation is shown here.

3. To validate the ddPCR results it would be useful to employ an additional, independent assay (e.g. FACS). This should be relatively easy to achieve by adapting the donor plasmid. Employing a donor that carries a promoterless fluorescent marker gene could for instance be targeted to the ALB promoter described in the manuscript. A simple FACS assay would then reveal the successful integration events.

We agree with the reviewer that our study would benefit from an orthogonal approach to assess integration efficiencies. Although the FACS assay suggested by the reviewer is a viable option, the need for target gene expression, which is cell-type-dependent, limits the number of sites that can be tested. As an alternative, we instead employed a UMI-deduplicated high-throughput sequencing (HTS) assay, as described in the previous comment to assess integration efficiencies across seven sites (Extended Data Fig. 6c). Values obtained from the HTS assay aligned well with those obtained from ddPCR analysis (Extended Data Fig. 6d, an example shown in the figure above). Additionally, the HTS assay also enabled us to evaluate additional parameters post-integration, including indel distribution and frequencies, and recombination efficiencies across multiple sites (Figure 4e), which are discussed in more detail in the manuscript, and below.

4. The authors optimized the PASSIGE system also to reduce indel formation. Nevertheless, even for the best combinations tested, indels were still observed. It would be useful to describe the observed indel sequences in more detail (additional Suppl. Table?) to understand better where they come from and whether they could cause unintended outcomes. This might also help to explain why much higher indel formation was observed for the attP installation at the CCR5 locus and for the Rosa26 and Smn1 loci for both, attP and attB installation. It would also be useful to understand how the observed indels influence overall PASSIGE efficiencies.

The high indel frequencies observed at the *Rosa26* and *Smn1* loci are primarily a consequence of incomplete attachment site

installation (as shown in the right). Thus, for these edits, utilizing more processive PE6 variants, such as PE6c and PE6d described in Doman *et al.* Cell 2023, which are more capable of fully reverse transcribing the RTT in the pegRNA, improves installation efficiencies and reduces indels (data in Supplementary Table 7). In our initial submission, we had also reported high indel frequencies at the *CCR5* site when installing *attP*. However, upon further analysis, we discovered that these indels were actually a result of sample contamination, and that the true indel frequencies are considerably lower, as reported in Supplementary Table 7. We have corrected this data in the previous Extended Data Fig. S4b and appreciate the reviewer's assistance in identifying this issue.

In this revised submission we also report indel frequencies across seven different loci **following** PASSIGE-mediated integration (Extended Data Fig. 6b). As expected, the majority of indels originated from the prime editing step, many of which were a consequence of incomplete attachment site installation. As requested by the reviewer, we have reported these indels in Supplementary Table 9 (top 5 indels for PASSIGE and eePASSIGE at 6 loci) and have updated the manuscript to highlight these observations.

5. The authors report off-target integration when eeBxb1 was used in combination with an *attP*-donor. It would be useful to determine the integration site(s) (e.g. by Splinkerette PCR).

During the revision process, we substantially expanded our off-target characterization by adding new experiments: 1) We used a modified version of UdiTaS, a genome-wide assay recently used by multiple studies (e.g. Durrant *et al.*, NBT 2022) to nominate potential off-target sites of large-gene integration platforms. We performed this assay for all PASSIGE variants and PASTE when delivering either an *attP*- or *attB*- containing donor. **Supplementary Table 11** lists the potential off-target sites nominated by UdiTaS for all variants, as requested by the reviewer. 2) We then validated several of these nominated sites using ddPCR and targeted sequencing. Although several UdiTaS-nominated hits were confirmed to be genuine off-targets, most were found to be false positives, as demonstrated in Fig 5c, Extended Data Fig 5d, and explained in detail in Supplementary Note 3. Out of 12 nominated sites tested, only 4 demonstrated off-target integration >0.2%. The revised manuscript provides a comprehensive discussion on the validity of these nominated off-target sites.

Additionally, consistent with our earlier findings from the mCherry assay results, many of the off-target sites nominated by UdiTaS, and all four off-target sites validated by ddPCR resembled the Bxb1 *attB* sequence in samples treated with eeBxb1 and an *attP*-containing donor (Supplementary Table 11). Importantly, through these extensive characterizations, we observed that all recombinase variants exhibit minimal off-target integration (<0.16%) when delivering an *attB* donor into cells. These new data confirm that installing an *attP* site into the genome and using an *attB*-containing donor minimizes off-target integration.

6. The rationale behind the authors' choice to employ distinct PE systems (PEmax or PE6) for integrating therapeutic DNA cargo via PASSIGE variants remains unclear. Is there a specific justification for utilizing different PE systems? Additionally, could employing the alternative PE system potentially yield different results, prompting a reevaluation of the findings?

We were not sufficiently clear in the in the initial submission. Our recent publication (Doman *et al.*, Cell 2023) demonstrates that PE6 variants can enhance prime edits requiring longer, more structured reverse transcriptase templates including recombinase attachment site installation. Consequently, in this current study, we evaluated the ability of these variants to install the Bxb1 *attB/P* sequence at each target site to identify the most suitable variant to use in each case. As illustrated in Supplementary Table 7, PE6 variants can improve attachment site installation compared to PEmax at several sites. To address the reviewer's concern regarding whether alternative PE systems could potentially yield different results, we have provided evidence below demonstrating that they can indeed lead to modest improvements. However, we believe that these findings do not necessitate a reevaluation of our conclusions, as 1) the observed improvements align with our initial hypothesis that

the enhancements are correlated with those seen for attachment site installation (see below), and 2) all comparisons between PASSIGE, evoPASSIGE, and eePASSIGE across the manuscript were performed using the same prime editor variant. To improve clarity of our rationale, we have added the following sentences and data (Extended Data Fig. 4d) to the manuscript:

“For the *Rosa26* site, we used the PE6d prime editor variant, as it has been recently reported to outperform PEmax for *attB* installation at this site. Improvements in attachment site installation led to modest enhancements in PASSIGE-mediated integration (Extended Data Fig. 4d), prompting us to evaluate PE6 variants for each target locus throughout the remainder of this study.”

7. For the integration of the F9 minicircle DNA donor into the Albumin locus in HuH7 cells only the improved expression levels are reported (unless these correspond to the results presented in Fig. 5a). It would be useful to also mention the integration efficiencies.

We thank the reviewer for pointing out our omission. We have added this data to the manuscript (see Extended Data Fig. 9a).

Minor points

1. FANCA targeting and indel results should be added to Extended Data Fig. 8.

Despite testing different polymerases, primer pairs, and PCR strategies, we were unable to amplify the *FANCA* site tested for high throughput sequencing due to its unusually high GC content (81%). We have added text to Supplementary Table 7 to reflect this.

2. In Figure 2d the mutations A449V and T435A should be swapped to maintain the order of mutations shown (ascending order).

We thank the reviewer for noting this mistake, which has now been corrected.

Reviewer #2:

The manuscript by Liu and colleagues describes the utilization of a phage evolution system to improve the activity of Bxb1 recombinase for integration of a DNA cargo site specifically mediated by prime editing (PASSIGE). The authors use two different phage evolution approaches (PACE & PANCE) to isolate Bxb1 mutants that have improved recombinase activity in bacteria. Next, they screen a number of these isolated Bxb1 variants individually and in combination to define evoBxb1 and eeBxb1 variants that display improved recombination activity in mammalian cells (primarily HEK293T cells or N2A cells). These yield improved PASSIGE systems for site-specific DNA integration (typically 2 to 3 fold higher rates than the original system). They also assess the off-target activity of evoBxb1 and eeBxb1 with DNA donors containing each recombination site (*attP* and *attB*) and observe off-target activity only with the *attP* donor in the context of eeBxb1. Finally, they compare their improved PASSIGE systems (evoPASSIGE or eePASSIGE) with the previously described PASTE system that uses a similar approach with distinct components for site-specific DNA integration. The PASSIGE

system outperforms PASTE in multiple contexts.

Overall this is an interesting manuscript that describes important improvements to the PASSIGE system for the site-specific integration of donor DNAs within the genome. The experiments are clearly presented within the manuscript, and the authors provide many extended data figures that provide important depth to their analysis of the Bxb1 variants and the PASSIGE system. The detailed comparison between the PASSIGE and PASTE systems will be valuable to readers that are interested in choosing an approach for integrating large cargos within the genome.

The primary concern of this reviewer is the incremental nature of the content, as this study builds on the primary description of the PASSIGE system (Anzalone and Gao NBT 2022). The primary achievement is improved activity, and there is assessment of some additional therapeutically relevant targets, but the manuscript limits the analysis of PASSIGE to transformed cell lines. For example, the integration of Factor IX in the ALB locus was performed in the prior study.

We thank the reviewer for this feedback. As noted above, we have now added data in cells beyond transformed cell lines and reported targeted integration in therapeutically relevant cells including iPSCs and **primary human fibroblasts** to the manuscript (Figure 6c-d). By using a non-plasmid delivery modality, we demonstrate that eePASSIGE achieves up to **30%** targeted integration of a **5.8-kB** DNA cargo at therapeutically relevant sites, following a single-electroporation, (shown on the right) and exhibits a substantial **14-fold** improvement over PASSIGE in primary human fibroblasts.

Additionally, in clonal **human induced pluripotent stem cells** with pre-installed attachment sites, we demonstrate that eeBxb1 improves integration of a **5.6-kB** plasmid by **7.3-fold** compared to the wild-type recombinase (Fig. 6c).

Minor concerns/suggestions:

1) While the PANCE and PACE evolution systems identified Bxb1 mutants with improved activity, the selections did not appear to converge to specific mutants based on the mutations recovered from the clones in the supplementary tables (despite the authors claims of a “high degree of mutational convergence”). The recovered clones appear to have characteristics of both selection and jackpot effects (Table S4 – V74A is only found in a single clone in one lagoon and V105I is found in two related clones in a single lagoon). Do the authors have evidence of “a high degree of mutational convergence”? The modest depth that the selected pools were analyzed (four individual recovered clones were characterized from each lagoon) makes an assessment of convergence challenging. In addition, negative selective pressure (absence of mutations at amino acid positions that will negatively impact function) will cause the appearance of an enrichment of mutations at positions that are tolerant of change, such as linkers where many mutations were recovered.

We agree. Although we do see some convergence of mutations, we acknowledge that we did not have sufficient data to characterize the degree of convergence as “high”. To address this concern, we have removed any claims suggesting a high degree of mutational convergence from the manuscript.

2) For the off-target analysis, the authors propose that the integration is occurring at attB-resembling sequences within the genome by eeBxb1. Deep sequencing analysis of the integration sites (using a method similar to used for genome-wide mapping of lentiviral integration sites) should provide concrete evidence of the sequences where integration is occurring.

As mentioned in our response to reviewer 1's Major Point #5, during the revision process, we substantially expanded our off-target characterization by adding two new experiments using a modified UdiTaS method to perform genome-wide off-target site nomination and subsequent ddPCR and sequencing analysis in depth.

Additionally, consistent with our earlier findings from the mCherry assay results, many of the off-target sites nominated by UdiTaS, and all four off-target sites validated by ddPCR resembled the Bxb1 *attB* sequence in samples treated with eeBxb1 and an *attP*-containing donor (Supplementary Table 11). Importantly, through these extensive characterizations, we observed that all recombinase variants exhibit minimal off-target integration (<0.16%) when delivering an *attB* donor into cells. These new data confirm that installing an *attP* site into the genome and using an *attB*-containing donor minimizes off-target integration.

3) Epigenomic differences between different genomic loci can impact the efficiency of genome editing in some systems. Is there any evidence that transcriptionally active genomic loci are more amenable to modification using the PASSIGE system? In their prior 2022 study ALB was easier to target with PASSIGE in Huh7 cells than HEK293T cells.

To address this comment, we performed Pearson correlation analyses between PASSIGE-mediated integration efficiencies, and several histone modification signals extracted from ENCODE for each locus in HEK293T cells. Interestingly, the analysis revealed that PASSIGE-mediated integration is indeed positively correlated with chromatin accessibility. In contrast, Bxb1-mediated recombination showed no such correlation, indicating that this dependency primarily comes from the prime editing step. We have added text to the manuscript describing these observations and the data for these analyses are presented in Extended Data Fig. 7 and Supplementary Table 10.

4) In Figure 4 panel A – comparison of the activity of the different systems (PASSIGE, evoPASSIGE & eePASSIGE) at common target sites (e.g. ACTB) is made more challenging because neighboring graphs have different Y-axis scales. Showing the mean value above each bar would assist the reader in discerning that the improved PASSIGE systems have higher activity in general. Just going by bar height all three systems appear to have similar activity at ACTB.

We thank the reviewer for noting this. We have changed the Y-axis scale appropriately to improve clarity (now, Fig. 3c).

Question: Since the authors are working with a system where the active species is a dimer of dimers and the N-terminal catalytic domain is at the interface of the four molecules in the active complex, have they considered the possibility that some of their mutants described as being in the “core” of the N-terminal domain (W35, L29, V74 & V105) might be involved in dimerization or tetramerization, and that they are selecting residues that are optimal in the context of a heterodimer between the WT sequence and the recovered mutant sequence? It is hard to tell from the structural models that are presented within the manuscript how buried these “core” residues are. There are multiple ways that more than one Bxb1 protein species being expressed in a bacterial cell could occur in the context of their selection system. Bxb1 heterodimers might have higher recombinase activity than the homodimers.

The reviewer raises an interesting hypothesis that the presence of multiple Bxb1 variants may enhance dimerization and/or tetramerization of the recombinase complex, leading to increased integration. Although the αE helix, which connects the N-terminal domain (NTD), and C-terminal domain (CTD) is thought to primarily contribute to Bxb1 oligomerization (Rutherford, et al., 2014, Current Opinion in Structural Biology) we agree with the reviewer that the NTD mutations could also be playing a role, given that it is located at the interface of the four molecules in the tetrameric state. To reflect this, we have incorporated the following sentence into the discussion section:

“However, given the dynamic nature of Bxb1-mediated, we cannot exclude the possibility that these mutations may enhance recombination through alternative mechanisms, such as facilitating Bxb1 oligomerization.”

From a practical standpoint, delivering multiple Bxb1 species may present additional challenges in several contexts. However, elucidating the precise mechanism behind the enhanced integration remains valuable from a scientific perspective and could inform future engineering efforts.

Reviewer #3:

The authors have substantially improved on the efficiency of programmable sequence insertion by using PANCE/PACE evolution techniques to greatly enhance Bxb1-mediated integration. By combining the newly derived versions of Bxb1 (evoBxb1 and eeBxb1) with sixth-generation prime editors (previously developed by the same lab), they are able to achieve insertion efficiency, in cells, manyfold higher than previous techniques (PASTE and PASSIGE). The authors characterize their new tools broadly in therapeutically relevant genes and show that the gains in efficiency are highly generalizable across loci.

From a technical perspective, the work is flawless. In reading through the manuscript, I came up with not a single suggestion for improvement. This is a rare manuscript that I think could reasonably be published without any modification at all.

This is unquestionably an important advance for the genome editing field. Previous attempts at programmable sequence insertion (even PASSIGE and PASTE) have been rather disappointing. Yet programmable sequence insertion is widely regarded as the next frontier in therapeutic genome editing, since it uniquely would provide a mutation-agnostic way to treat genetic disorders. Tome Biosciences has been in the news a lot recently because of the potential perceived in their therapeutic efforts based on PASTE. It is clear that the new Bxb1 platform presented in this manuscript is far superior.

We thank the reviewer for this positive feedback.

Reviewer #4:

In this study, the Liu lab presents another directed evolution study in which they utilize their common PACE system to evolve the Bxb1 large serine recombinase. They develop a clever selection system in which different selection strengths enable their use of an optimal selection. They result in two Bxb1 variants, namely evoBxb1 and eeBxb1, which exemplify enhanced recombination activity either alone with pre-installed attP/attB sites or with their PASSIGE system. Indeed the evolved variants seem to exhibit increased editing, which reflects their successful engineering effort, I have some questions regarding the evolution process and off-target analysis. I believe that the authors should address these concerns prior to publication.

1. I was curious about the evolution and looked carefully at the mutant table. Can the authors explain why they did not find any obvious trends in particular mutations being amplified throughout the evolution. Rather, it seems that the mutant table is very sporadic and that the mutants from one PANCE do not enrich in subsequent rounds (even some seem to disappear). Other studies by the Liu lab show much better enrichment so I am curious why there were no trends in this recombinase evolution? Furthermore, why did library members enrich (better phage propagation) but did not show better activity that enrich over time?

We thank the reviewer for their feedback. This is an excellent observation that initially puzzled us as well, but our experiments in mammalian cells provided further clarity. Given that 39/40 variants tested from the evolution demonstrated improved recombination efficiencies, with 25 variants showing

> 2-fold improvement (Figure 2b-c), it was evident that the recombinase had found an incredibly high number of unique solutions to enhance integration, explaining the sporadic emergence of many mutations rather than the enrichment of a single genotype. We have added the following text to the manuscript to reflect this:

“This data indicates that the Bxb1-encoding selection phage found a remarkably diverse set of unique solutions to enhance integration, explaining the absence of a single dominant genotype throughout evolution (Supplementary Tables 1-4).”

Additionally, although the mutation tables may imply that several variants disappeared over time, it is important to note that sequencing 4-6 plaques only provides a limited snapshot of the vast number of genotypes present in a given phage pool. Therefore, it is very likely that these sequences were still present in the phage population.

Finally, the most likely explanation for why library members that enriched overtime did not additionally enhance integration is that the mechanisms by which these variants further improve recombination in bacteria may not translate to mammalian cells, which have a vastly different cellular environment, a much larger sequence space, and a chromatinized genome.

2. The authors show that the V105I mutation have enhanced activity alone when used with clonal cells. However, this did not show the best activity with the PASSIGE system. What do the authors believe is the reason for this discrepancy if the prime editing components are shared between both?

In clonal cell lines with pre-installed attachment sites, we observed only minor differences in activity when comparing the top 15 variants (Figure 2c), suggesting that we were approaching saturation in terms of integration percentages. Notably, V105I and V74A (evoBxb1) did not show statistically significant differences in integration at this setting (bar graph on the left). However, in the PASSIGE system, V74A demonstrated slightly higher integration at the *CCR5* site (bar graph on the right). Although this difference is minimal, we opted to proceed with V74A for simplicity. We have updated the manuscript to remove the statement that V105I is the highest activity evolved recombinase.

3. I hope the authors can directly compare the raw editing activity either in clonal cells or with the PASSIGE system between the single best mutants (V105I or V74A) and the best triple mutant (eeBxb1). The heat map does not exactly seem to enhance overall editing activity between the single and triple mutants.

We believe that the small difference we observed in activity when comparing the single and triple mutant variants in stable cell lines with pre-installed attachment sites was because we were approaching saturation in terms of integration percentages, as talked about in the previous comment. To better assess the differences between variants, we reevaluated a subset of the triple-mutant variants in the PASSIGE system (data shown on the right). Our findings confirmed two key points: 1) the trends observed in stable cell lines were consistent with those observed with PASSIGE, and 2) the V74A+E229K+V375I variant performed the best. To

address the reviewer's comment, we have added the raw editing data for the single point mutant variants (V105I and V74A) to Supplementary Table 6. Additionally, data presented in Figure 3b further emphasizes the substantial improvements in activity between the single and triple mutant variants.

4. The authors give a good hypothesis on how the mutation in the NTD increases activity. Could the authors do the same for the other two mutations in the CTD in their eeBxbI mutant?

As noted in Extended Data Fig. 8b, through structural modelling, we hypothesize that the E229K mutation located in the CTD-a domain of the Bxb1 recombinase likely enhances binding to the negatively charged DNA phosphate backbone. Conversely, the role of the V375I mutation, located in the second DNA-binding domain, CTD-b, remains elusive. Mapping the mutation onto an AlphaFold-predicted Bxb1 structure docked with a DNA substrate did not reveal any apparent interactions that could explain its contributions to enhanced integration (shown on the right). Notably, V375 is located in the CC motif of the recombinase which has been implicated in stabilizing the tetrameric synaptic complex of large serine recombinases (Rutherford et al., 2013, NAR). Given this, it is tempting to speculate that this residue could be enhancing Bxb1 oligomerization. Nevertheless, without experimental evidence, it is difficult to identify its exact role.

5. The authors give a very clever off-target analysis as analyzed by flow cytometry, which seems to also be used previously by other studies. I hope the authors can take their mCherry positive cells after flow and sequence them to indeed verify that the mCherry expression is due to integration and not plasmid leakage. This would be more powerful in supporting their claims. Furthermore, I wonder if the authors can take one of their samples with the attP donor and eeBxbI and sequence to identify where the off target integration is located. This would also support their claim on sites being more similar located in the human genome.

As mentioned in our detailed response to reviewer 1 and 2, during the revision process, we added new experiments using a modified UdiTaS method to perform genome-wide off-target site nomination and subsequent ddPCR and sequencing analysis in depth.

Additionally, consistent with our earlier findings from the mCherry assay results, many of the off-target sites nominated by UdiTaS, and all four off-target sites validated by ddPCR resembled the Bxb1 *attB* sequence in samples treated with eeBxb1 and an *attP*-containing donor (Supplementary Table 11). Importantly, through these extensive characterizations, we observed that all recombinase variants exhibit minimal off-target integration (<0.16%) when delivering an *attB* donor into cells. These new data confirm that installing an *attP* site into the genome and using an *attB*-containing donor minimizes off-target integration.

The revised manuscript has been substantially strengthened based on the reviewers' collective feedback. We are grateful for the reviewers' helpful comments and hope these new experiments and revisions address their concerns.

Rebuttal 2

Reviewer 4

The authors addressed most of my concerns and I commend the authors on their thorough evaluation of the Bxb1 recombinase. Based on the newly included statements, it seems that the E229K is a critical mutation that greatly enhances editing activity. I wonder if the authors performed any experiments amongst mutants containing E229K and double or triple variants containing E229K? This would be interesting and a more thorough evaluation with possible mutations that also enhance editing as opposed to the originally identified starting mutations from the early evolution studies. Otherwise, the authors addressed all my previous concerns.

We thank reviewer 4 for their comment. We have indeed tested several variants containing the E229K mutation in stable cell lines pre-installed with either *attB* or *attP*. These results are already reported in the manuscript:

1) In Figure 2b, we tested the single point mutant E229K along with other variants. The source data for Fig 2b (previously Supplementary Table S5) contains the absolute integration efficiencies. Here, E229K substantially improves integration efficiencies by 2-fold compared to WT Bxb1.

2) In Extended Data Fig. 4c, we evaluated six triple mutant variants with the E229K mutation. These variants were engineered by rationally combining individual mutations from evolution. The source data for Extended Fig. 4c (previously Supplementary Table S6) contains the absolute integration efficiencies. Here, the V74A + E229K + V375I (eeBxb1) outperforms all other variants tested.

In addition to the data presented in the manuscript we have also tested double and triple mutant variants with the E229K mutation for PASSIGE (shown on the right). This data indicates that the addition of E229K enhances integration efficiencies, and the eeBxb1 variant performs the best, consistent with our findings in Extended Fig. 4c. Overall, we agree with the reviewer that the E229K is a critical mutation that enhances integration. Thus, we have added the following phrase to the manuscript to highlight this: “Interestingly, the best-performing variants all contained the E229K mutation....”